# NANOG governs cell metabolism and redox homeostasis in human naïve embryonic stem cells

Min Shao [1,2,3,7], Han Wang [1,7], Yujie Liu [1], Yongqiang Wang[3,4,6], Hanzhi Zhao[2], Junjie Gu[1], Ning Zhong[1], Yifan Zhou[2], Huiyong Yin[4,5], Ying Jin [1,2✉] & Bing Liao [1✉]

## Abstract

**Naïve human embryonic stem cells (hESCs) possess some advantages over their primed counterparts, displaying distinctive metabolic and epigenetic properties. However, the master regulator governing these features remains unrecognized. Here, we systematically investigate functions of the core transcription factor NANOG in naïve hESCs. Acting as an upstream key regulator, NANOG directly activates genes associated with naïve pluripotency, acetyl-CoA synthesis and anti-oxidation in a naïve pluripotency state-dependent manner, and represses the expression of extraembryonic lineage genes in naïve hESCs. NANOG modulates transcription of multiple genes in various pathways of acetyl-CoA synthesis, maintains the intracellular acetyl-CoA level and characteristic epigenetic landscapes, particularly the high level of histone acetylation, in naïve hESCs. NANOG is indispensable for the high activity of both OXPHOS and glycolysis, a bivalent metabolic state typical in naïve hESCs. Furthermore, we identify GPX2 as a mediator of NANOG in sustaining redox balance and survival of naïve hESCs. Together, this study reveals previously unrecognized roles of NANOG in orchestrating transcriptional, metabolic and epigenetic signatures to secure human naïve pluripotency.**

**Keywords** Human Naïve Pluripotency; NANOG; Histone Acetylation; Acetyl-CoA Synthesis; GPX2
**Subject Categories** Chromatin, Transcription & Genomics; Metabolism; Stem Cells & Regenerative Medicine

## Introduction

Pluripotency is defined as a regulatory potential of single cells to generate all mature cell lineages of an organism, manifesting as a developmental continuum in the embryo and defined culture (Smith, 2017). Mouse embryonic stem cells (mESCs) are considered to display preimplantation epiblast features in a naïve pluripotency

state, displaying global DNA hypomethylation, activation of specific transposable elements, and X chromosome reactivation in female cell lines (Takashima et al, 2014; Theunissen et al, 2014). In contrast, conventional human embryonic stem cells (hESCs), derived from human preimplantation blastocysts (Thomson et al, 1998), represent post-implantation epiblasts in a primed pluripotency state (Brons et al, 2007). Over the past decade, advances in culture techniques have enabled the conversion of primed hESCs to a naïve state and direct establishment of naïve hESCs from the human blastocyst (Guo et al, 2021), akin to mESCs (Duggal et al, 2015; Takashima et al, 2014; Theunissen et al, 2014; Ware et al, 2014), offering valuable models for studying early human developmental mechanisms. However, challenges in maintaining genomic stability, epigenetic integrity, and conversion efficiency in naïve hESC cultures persist, underscoring the need for further research into transcriptional, epigenetic, and metabolic mechanisms underlying naïve pluripotency.

Emerging evidence has highlighted the importance of metabolic regulation in determining the cell fate of pluripotent stem cells (PSCs) (Folmes et al, 2012; Shyh-Chang and Ng, 2017; Sperber et al, 2015; Zhang et al, 2018). Interestingly, primed human PSCs (hPSCs) almost exclusively rely on glycolysis (Zhou et al, 2012), whereas naïve hPSCs have high levels of both glycolysis and oxidative phosphorylation (OXPHOS) (Sperber et al, 2015; Takashima et al, 2014; Theunissen et al, 2014). It was also shown that naïve hESCs exhibited higher glycolysis flux than primed hESCs (Gu et al, 2016). This bivalent metabolic program was considered to enable naïve hPSCs to efficiently adapt to different culture conditions and might contribute to their enhanced clonogenic feature (Mlody and Prigione, 2016). Despite the functional advantage, the upstream determinant of the bivalent metabolic feature in naïve hESCs remains elusive. On the other hand, active oxidative phosphorylation observed in naïve hPSCs can generate high levels of reactive oxygen species (ROS), which could damage intracellular macromolecules and DNA when their levels are beyond physiological ranges (Ufer and Wang, 2011). Thus, ROS scavenging systems, including antioxidant compounds and enzymes, are vital for the maintenance of genomic integrity and cellular functions. However, how the ROS homeostasis system acts and is modulated in naïve hPSCs remains largely an open question.

[1]Department of Histoembryology, Genetics and Developmental Biology, Shanghai Key Laboratory of Reproductive Medicine, Shanghai Jiao Tong University School of Medicine, Shanghai 200025, China. [2]The Key Laboratory of Tissue Microenvironment and Tumor, Shanghai Institute of Nutrition and Health (SINH), Chinese Academy of Sciences (CAS), Shanghai, China. [3]University of CAS, Beijing, China. [4]The Key Laboratory of Nutrition, Metabolism and Food Safety Research, SINH, CAS, Shanghai, China. [5]Department of Biomedical Sciences, City University of Hong Kong, Hong Kong SAR, China. [6]Present address: Frontier Medical Center, Tianfu Jincheng Laboratory, Chengdu, Sichuan, China. [7]These authors contributed equally: Min Shao, Han Wang. ✉E-mail: yjin@sibs.ac.cn; liaobing@shsmu.edu.cn

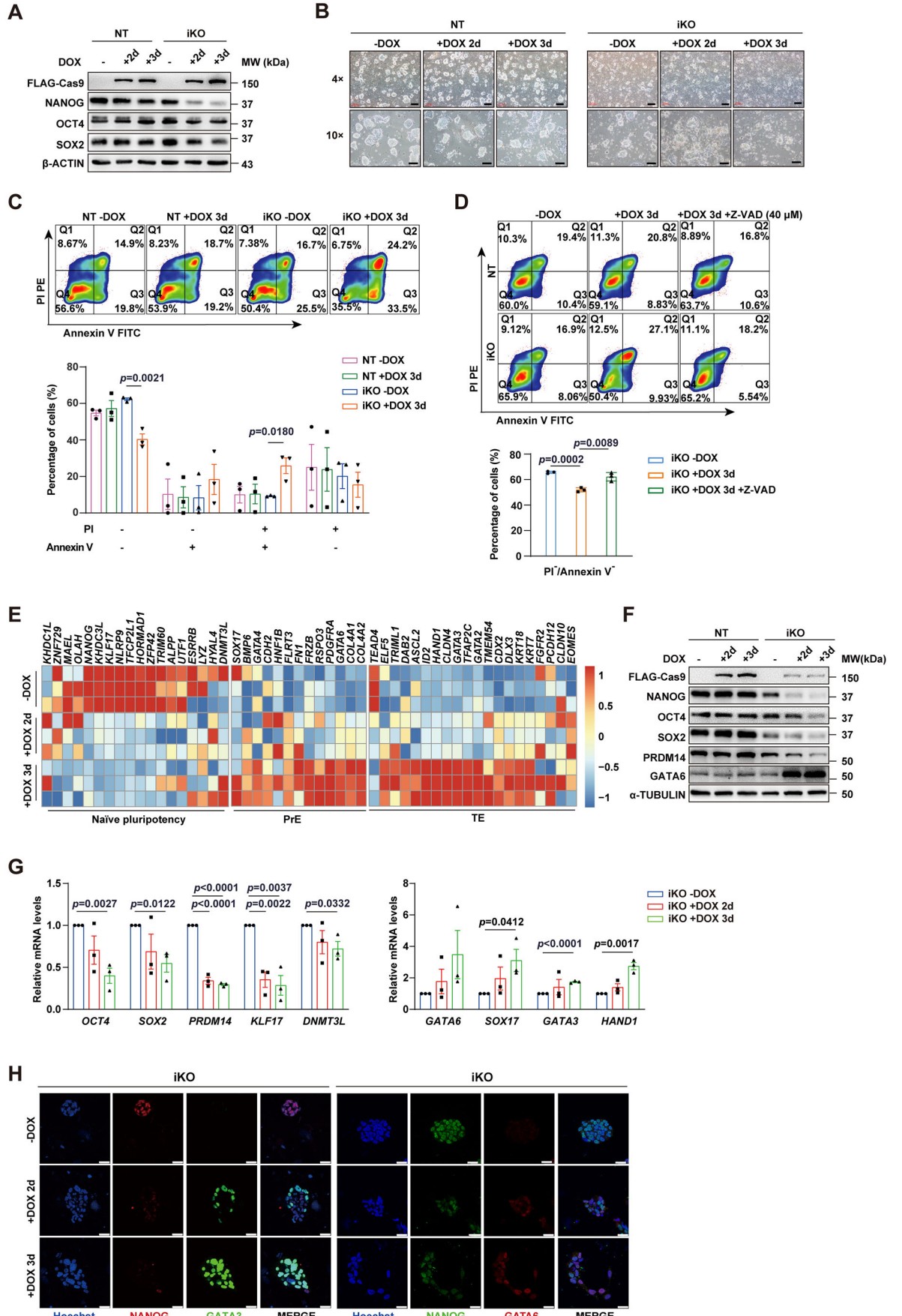

**Figure 1. NANOG deficiency leads to cell death and altered naïve transcriptional program.**

(A) A representative western blot analysis result showing the steady-state levels of proteins examined in control naïve NT (non-target) or *NANOG* iKO SHhES8 hESCs, without or with DOX treatment for two or three days. Beta-ACTIN served as a loading control. (B) Phase contrast images of cell colonies in naïve NT and *NANOG* iKO SHhES8 hESCs, either untreated or treated with DOX for two or three days. Scale bars, 200 μm (4×), 100 μm (10×). (C) A representative result of Annexin V/PI flow cytometry analysis (top) and statistical analysis results (bottom) from three independent experiments in naïve NT and *NANOG* iKO SHhES8 hESCs, either untreated or treated with DOX for three days. Data are presented as mean ± SEM ($n = 3$). (D) A representative Annexin V/PI flow cytometry analysis result (top) and statistical analysis of results (bottom) from three independent experiments in naïve NT and *NANOG* iKO SHhES8 hESCs, either untreated or treated with DOX or DOX and Z-VAD (40 μM) for three days. Data are presented as mean ± SD ($n = 3$). (E) Heatmaps showing normalized expression levels of selected naïve pluripotency, PrE (primitive endoderm), and TE (trophectoderm) marker genes measured by RNA-seq in naïve *NANOG* iKO SHhES8 hESCs, either untreated or treated with DOX for two or three days. The color represents Z-Scores. Three biological replicates were used for each condition ($n = 3$). (F) A representative western blot analysis result for steady-state levels of indicated proteins in naïve NT or *NANOG* iKO SHhES8 hESCs, either untreated or treated with DOX for two or three days. Alpha-TUBULIN served as a loading control. (G) RT-qPCR analysis results for relative mRNA levels of pluripotency genes (left) and lineage marker genes (right) in naïve *NANOG* iKO hESCs, either untreated or treated with DOX for two or three days. Data are presented as mean ± SEM ($n = 3$). (H) Representative immunofluorescence staining images of NANOG, GATA3, and GATA6 in naïve *NANOG* iKO hESCs, either untreated or treated with DOX for two or three days. Scale bar, 25 μm. The unpaired two-tailed student's *t*-test was used for statistical analysis in (C, D, G). Source data are available online for this figure.

In addition, metabolites can serve regulatory roles through modulating chromatin modifications and signaling pathways to ultimately control cell fate. For example, the availability of metabolite acetyl-coenzyme A (acetyl-CoA) has been shown to be closely associated with gene expression programs and cell fate determination (Moussaieff et al, 2015; Park et al, 2022). As an acetyl donor of protein acetylation modification, acetyl-CoA is generated by numerous pathways, including mitochondrial conversion of pyruvate to acetyl-CoA, cytosolic synthesis of acetyl-CoA from citrate via ATP-citrate lyase (ACLY) and from acetate via acetyl-CoA synthetase 2 (ACSS2) (He et al, 2023). The mitochondrial carnitine shuttle and peroxisomal metabolism also contribute to cellular acetyl-CoA generation (Kuna et al, 2023). Although the level of acetyl-CoA is closely linked to cell fate determination, the question of how its synthesis is controlled in naïve hESCs is unsolved.

Cellular metabolism and redox homeostasis in cultured hPSCs are modulated by intrinsic factors and environmental cues. The core transcription factor NANOG is expressed at a higher level in naïve hESCs than in primed hESCs (Szczerbinska et al, 2019). Particularly, inducible overexpression of *NANOG* and *KLF2* facilitates the primed-to-naïve transition in hESCs (Takashima et al, 2014; Theunissen et al, 2014), suggesting a critical role of NANOG in human naïve pluripotency resetting. Therefore, we were interested in addressing questions of what the function of NANOG in naïve hESCs is and how it contributes to the naïve features of pluripotency. To address these questions, we have established a doxycycline (DOX)- inducible *NANOG* knockout (KO) model in naïve hESCs. Unexpectedly, we find that, in addition to differentiation, *NANOG* deficiency in naïve hESCs leads to the disruption of metabolic processes, including mitochondrial respiration and glycolysis, as well as the reduced levels of histone acetylation, highlighting a pivotal role of NANOG in governing energy metabolism and epigenetic landscape of naïve hESCs. Our further investigation indicates that NANOG secures human naïve pluripotency by directly activating naïve pluripotency- associated genes and repressing certain differentiation- associated genes. Particularly, NANOG is required for the expression of a set of genes important for acetyl-CoA production. Moreover, NANOG also controls the naïve unique gene expression network at an epigenetic level. Furthermore, this study identifies *GPX2* (selenium-dependent glutathione peroxidase 2), a gene encoding an antioxidant enzyme, as a new target gene of NANOG to protect naïve hESCs against excessive ROS and maintain their survival. Collectively, we reveal the key role of the core transcription factor NANOG in modulating metabolic processes and redox homeostasis to safeguard human naïve pluripotency, shedding new insights into how the transcription factor, metabolic processes, and epigenetic signatures interplay to determine cell fates of naïve hPSCs.

## Results

### NANOG deficiency in naïve hESCs induces cell death and activation of extraembryonic lineage genes

To characterize the function of NANOG in naïve and primed hESCs, we generated DOX- inducible *NANOG* knockout (iKO) primed hESC lines (SHhES8 and H9 hESC lines) using the CRISPRn system (Mandegar et al, 2016), and then converted the edited primed *NANOG* iKO hESCs to the naïve state using a previously published 5iLA strategy (Theunissen et al, 2014). The successful establishment of the naïve pluripotency state was indicated by more than 95% of CD75 and SUSD2 double-positive cells (Fig. EV1A), activation and repression of naïve and primed pluripotency markers (Fig. EV1B), respectively, as well as high levels of nuclear DNMT3L, KLF17, REX1, and NANOG with negative staining of SSEA4 in these converted naïve hESCs (Fig. EV1C). The DOX treatment of converted naïve hESCs efficiently depleted *NANOG*, leading to decreases in OCT4 and SOX2 protein levels (Fig. 1A). Moreover, DOX- induced *NANOG* depletion triggered extensive cell death, as evidenced by reduced colony numbers (Fig. 1B), increased percentages of Annexin V and PI double-positive cells (Fig. 1C) and elevated levels of cleaved Caspase-3 (Fig. EV1D, top). Similar cell death was also observed in *NANOG* depleted primed hESCs (Fig. EV1D bottom - F). In contrast, these alterations were not observed in DOX-treated control cells (NT hESCs), suggesting that the cell death detected in *NANOG* iKO cells was specifically caused by *NANOG* deficiency. Notably, the reduction in percentages of Annexin V and PI double-negative cells induced by *NANOG* depletion in naïve hESCs was rescued by the treatment of Z-VAD, a pan-caspase inhibitor (Fig. 1D). These results clearly show an indispensable role of NANOG in maintaining the survival of primed and naïve hESCs.

To gain a global picture of transcriptional changes induced by *NANOG* deficiency in both primed and naïve hESCs, we carried out

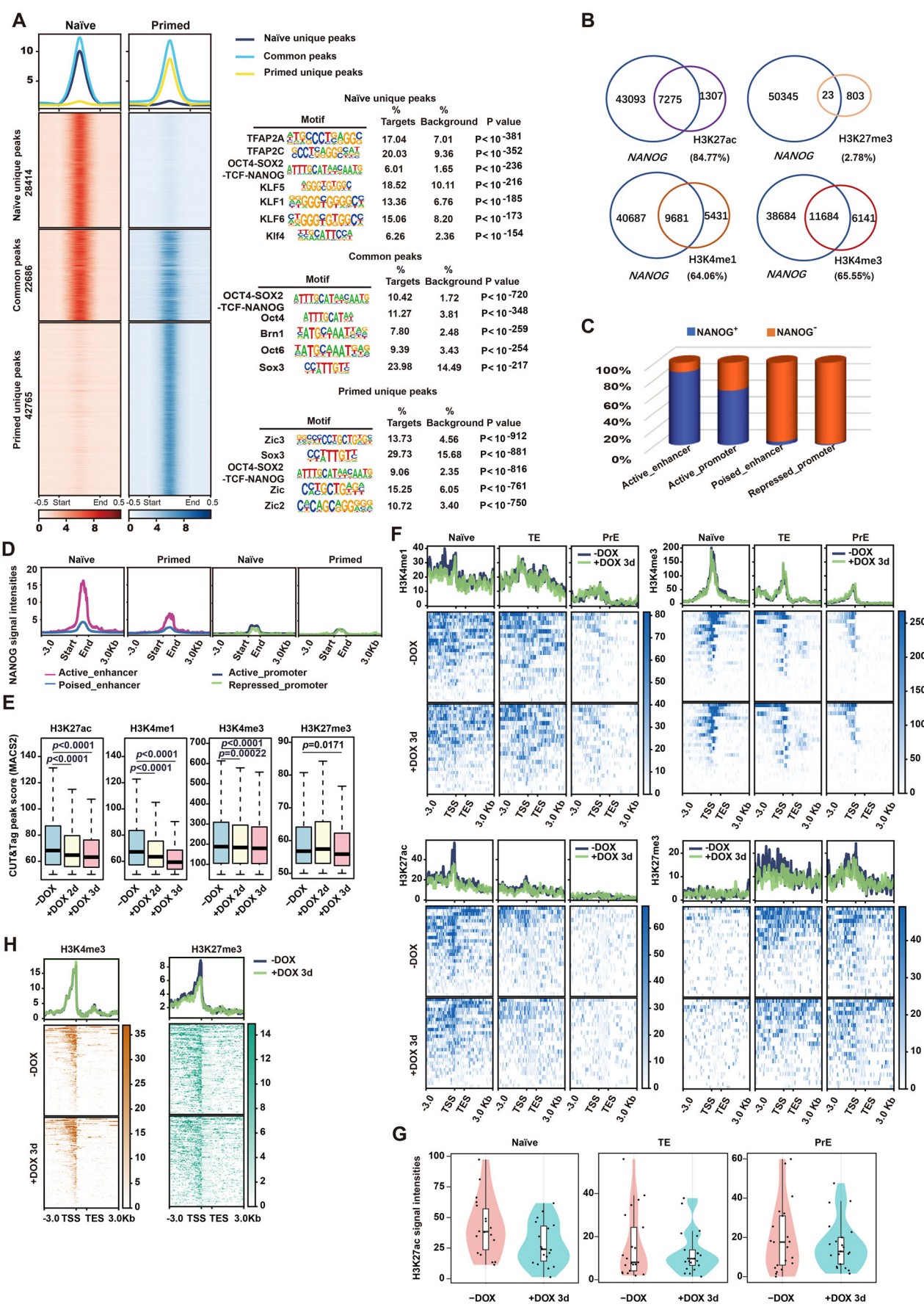

**Figure 2. NANOG maintains epigenetic signatures in distinct genomic regions of naïve hESCs.**

(A) The comparison of NANOG binding sites between naïve and primed hESCs. The heatmap and corresponding average peak map show unique and common NANOG binding peaks between naïve and primed hESCs (left). The motif enrichment analysis of NANOG binding regions by HOMER is shown on the right side. The hypergeometric test was used for statistical analysis. (B) Venn diagrams exhibiting the overlapping between NANOG binding sites and H3K27ac, H3K27me3, H3K4me1, or H3K4me3 modification sites, respectively, in naïve hESCs. The percentage of overlapping sites between NANOG and each histone modification in the total sites of each modification is shown in the bracket. (C) The graph shows the percentage of enhancers and promoters with or without NANOG occupancy in the total sites of each kind of regions in naïve hESCs. (D) The graph shows signal intensities normalized by the total reads of NANOG binding on enhancers and promoters in primed and naïve hESCs, respectively. (E) Box plots of peak scores from CUT&Tag assays for H3K27ac, H3K27me3, H3K4me1 and H3K4me3 in naïve *NANOG* iKO SHhES8 hESCs, either untreated or treated with DOX for two or three days. H3K27ac (-DOX, +DOX 2d, +DOX 3d) Minimum value: 50.00, 50.01, 50.000; Maxim value: 131.35, 114.94, 107.440; Median value: 68.22, 64.75, 63.210; 25th percentile: 57.41, 56.02, 55.470; 75th percentile: 87.03, 79.59, 76.295; H3K4me1 (-DOX, +DOX 2d, +DOX 3d) Minimum value: 50.00, 50.00, 50,00; Maxim value: 122.780, 105.03, 90.35; Median value: 67.225, 63.49, 59.37; 25th percentile: 57.320, 55.73, 54.01; 75th percentile: 83.520, 75.45, 68.58; H3K4me3 (-DOX, +DOX 2d, +DOX 3d) Minimum value: 50.00, 50.00, 50.010; Maxim value: 613.39, 578.71, 557.550; Median value: 187.75, 183.61, 179.610; 25th percentile: 105.33, 105.11, 103.985; 75th percentile: 308.76, 294.88, 285.505; H3K27me3 (-DOX, +DOX 2d, +DOX 3d) Minimum value: 50.01, 50.01, 50.010; Maxim value: 80.75, 84.24, 76.600; Median value: 56.77, 57.41, 55.865; 25th percentile: 53.05, 53.05, 52.370; 75th percentile: 64.13, 65.75, 62.280. Data are presented as mean ± SD ($n$ peak numbers). H3K27ac (-DOX, +DOX 2d, +DOX 3d): $n = 8582$, $n = 5433$, $n = 4055$; H3K4me1 (-DOX, +DOX 2d, +DOX 3d): $n = 15112$, $n = 11069$, $n = 6507$; H3K4me3 (-DOX, +DOX 2d, +DOX 3d): $n = 17825$, $n = 17469$, $n = 17243$; H3K27me3 (-DOX, +DOX 2d, +DOX 3d): $n = 826$, $n = 889$, $n = 646$. The Wilcoxon signed-rank test was used for this statistical analysis. H3K27ac, $p < 2.2 \times 10^{-16}$ (Row 1), $p < 2.2 \times 10^{-16}$ (Row 2); H3K4me1, $p < 2.2 \times 10^{-16}$ (Row 1), $p < 2.2 \times 10^{-16}$ (Row 2); H3K4me3, $p < 2.211 \times 10^{-11}$ (Row 1). (F) The deepTools was used to calculate the signal intensities normalized by the total reads of histone modifications (H3K27me3, H3K27ac, H3K4me3, and H3K4me1) within the 3 kb range upstream and downstream of marker genes associated with naïve pluripotency, TE, or PrE in naïve *NANOG* iKO SHhES8 hESCs, either untreated or treated with DOX for three days. (G) Violin plots showing the mean signal intensities of H3K27ac modification within the 2 kb range upstream and downstream of marker genes associated with naïve pluripotency, TE, or PrE in naïve *NANOG* iKO SHhES8 hESCs, either untreated or treated with DOX for three days. Naïve (-DOX, +DOX 3d) Minimum value: 11.40227, 1.335114; Maxim value: 97.35516, 61.694472; Median value: 38.43097, 24.038136; 25th percentile: 21.00130, 13.096263; 75th percentile: 59.74109, 43.652194; TE (-DOX, +DOX 3d) Minimum value: 1.962644, 1.456488; Maxim value: 39.183159, 22.666608; Median value: 8.079718, 9.782750; 25th percentile: 4.134504, 6.675573; 75th percentile: 24.318866, 13.885198; PrE (-DOX, +DOX 3d) Minimum value: 0.05056987, 1.590838; Maxim value: 59.91265543, 38.511498; Median value: 17.63624857, 12.859273; 25th percentile: 5.86610445, 6.396497; 75th percentile: 30.99934844, 20.004778. Data are presented as mean ± SD. (Naïve, $n = 18$; TE, $n = 21$; PrE, $n = 21$). The Wilcoxon signed-rank test was used for the statistical analysis. Naïve, $p = 0.07898$; TE, $p = 1$; PrE, $p = 0.7462$. (H) Heatmaps depicting changes in the intensity (normalized by total reads) of histone modifications (H3K4me3 and H3K27me3) on the bivalent genes in naïve *NANOG* iKO SHhES8 hESCs, either untreated or treated with DOX for three days, analyzed by the deepTools.

RNA-seq analyses. In naïve hESCs, on day 2 of DOX treatment, there were 61 upregulated and 62 downregulated differentially expressed genes (DEGs) (Fig. EV1G, left). Upregulated DEGs included trophectoderm (TE) marker gene *GATA3* and primitive endoderm (PrE) marker gene *GATA6*. Naïve marker genes (*KLF17* and *NLRP9*) and epiblast marker *PRDM14* were downregulated (Fig. EV1G, left). On day 3, more DEGs were identified, including 602 upregulated and 418 downregulated genes (Fig. EV1G, right). Particularly, transcriptional levels of multiple naïve markers were reduced, which was accompanied by elevated levels of several TE and PrE marker genes (Fig. 1E). Changes in expression levels of some marker genes in naïve hESCs were validated (Fig. 1F–H). Different from naïve hESCs, *NANOG* deficiency in primed hESCs upregulated certain marker genes of the three major embryonic germ layers, especially for the ectoderm and endoderm, with concomitant downregulation of a subset of primed pluripotency marker genes (Fig. EV1H,I). Taken together, our results indicate that *NANOG* deficiency results in distinct transcriptional changes between primed and naïve hESCs. NANOG activates naïve pluripotency genes and prevents extraembryonic lineage differentiation in naïve hESCs, while it sustains expression of primed pluripotency genes and suppresses differentiation into embryonic lineages in primed hESCs. Despite of these differences, NANOG is essential for maintaining cell survival and the undifferentiated state for both naïve and primed hESCs.

## NANOG specifically modulates epigenetic landscapes to sustain the naïve transcriptional program

To understand how NANOG would modulate transcriptional programs, we conducted ChIP-seq assays using a specific NANOG antibody, obtaining a genome-wide view of its occupancy on chromatin in both primed and naïve hESCs. Data analysis identified three groups of NANOG binding sites: naïve unique peaks (28,414), primed unique peaks (42,765), and common peaks (22,686) (Figs. 2A, left and EV2A). NANOG binding sites in all three groups enriched the typical binding motif of OCT4-SOX2-TCF-NANOG, reflecting the reliability of our data analysis. Motifs of AP2 family members (TFAP2A and TFAP2C) and KLF family members were detected in naïve unique NANOG peaks, while motifs of ZIC family members were enriched in primed unique NANOG peaks (Fig. 2A, right). In line with this finding, TFAP2C, KLF5, and KLF4 were previously shown to play critical roles in controlling human naïve pluripotency (Ai et al, 2022; Pastor et al, 2018; Takashima et al, 2014), and Zic3 was reported to activate the expression of Nanog and a set of transcription factors, contributing to the establishment and maintenance of mouse primed pluripotency (Lim et al, 2010; Lim et al, 2007; Yang et al, 2019).

To assist dissecting the mechanism by which NANOG would regulate gene expression programs, we carried out CUT&Tag experiments in naïve hESCs, picturing binding profiles of various well-studied histone modifications and their relationships with NANOG. Strikingly, a high co-distribution was detected between NANOG binding sites and active transcription modification (H3K4me1, H3K4me3, and H3K27ac) sites, in contrast to the minimal overlap of distribution between NANOG and repressive transcription modification (H3K27me3) (Fig. 2B). Of note, H3K27ac exhibited the highest percentage of co-distribution with NANOG (84.77%) (Fig. 2B). Moreover, the majority of NANOG binding sites localized in the regions of H3K4me1+/H3K27ac+ active enhancers and H3K4me3+/H3K27ac+ active promoters, with a very small proportion in the regions of H3K4me1+/H3K27me3+ poised enhancers or H3K4me3-/H3K27me3+ repressed promoters (Fig. 2C). Among these four classes of defined genomic regions, the

binding intensity of NANOG was highest in active enhancers, being markedly higher in naïve hESCs than in primed hESCs (Fig. 2D). These findings highlight a close association of NANOG binding with epigenetic marks of active transcription, especially with H3K27ac.

We next examined whether *NANOG* deficiency would disturb the aforementioned histone modifications in naïve hESCs. Compared to control cells (without DOX), DOX- induced *NANOG* depletion significantly reduced the global histone modification levels of H3K27ac, H3K4me1, and H3K4me3, with only minor changes in the H3K27me3 level (Figs. 2E and EV2B). Particularly, signal intensities of H3K4me1 and H3K27ac near human naïve pluripotency genes were reduced, while those near PrE and TE genes were relatively low and were not altered by *NANOG* depletion (Fig. 2F,G). Conversely, H3K27me3 modification was rarely enriched near naïve pluripotency genes, but had higher levels near PrE and TE genes. Our detailed analysis detected reductions in the H3K27me3 level at specific regions near the transcription start sites (TSS) of PrE and TE genes after *NANOG* depletion (Fig. 2F, the right side of the bottom row). Therefore, reductions in the H3K27me3 level at PrE and TE genes as well as decreases in H3K4me1 and H3K27ac levels near human naïve pluripotency genes after *NANOG* depletion aligned well with upregulation of extraembryonic lineage genes and downregulation of naïve pluripotency genes, respectively. Furthermore, our ATAC-seq data analysis revealed higher chromatin accessibility at naïve pluripotency genes, compared to PrE or TE genes, in naïve hESCs (Fig. EV2C). Interestingly, *NANOG* depletion did not provoke evident effects on the open state of these genes and whole genome (Fig. EV2C,D).

To obtain further evidence that *NANOG* depletion distinctly impacted on epigenetic signatures of naïve pluripotency and lineage markers, we examined changes in histone modifications near these markers. For example, a set of naïve pluripotency markers (*KLF17, ZFP42, UTF1,* and *TFCP2L1*) were highly occupied by NANOG and marked by high levels of H3K27ac without H3K27me3 deposition in control naïve hESCs, consistent with their robust transcription. However, depletion of *NANOG* reduced the enrichment of H3K27ac evidently (Fig. EV2E). Differently, PrE markers (*GATA6, GATA4,* and *PDGFRA*) and TE markers (*GATA3, CDX2,* and *HAND1*) were marked with H3K4me3 and H3K27me3 simultaneously, a bivalent state, at their promoter regions in control naïve hESCs. *NANOG* depletion led to decreases in H3K27me3 enrichments in the promoter regions of these extra-embryonic lineage genes, with H3K4me3 enrichment being unchanged (Fig. EV2E). These alterations were in line with the upregulation of these lineage genes. The decrease in the H3K27me3 modification without changes in H3K4me3 enrichments was observed for all bivalent genes in *NANOG*- depleted cells (Fig. 2H). Of note, NANOG was enriched at the regulatory regions of *GATA6* and *GATA3*, suggesting a specific and direct regulatory role of NANOG in the expression of these two members of the GATA family. As for primed pluripotency markers, such as *CYTL1* and *CER1*, H3K27me3 exhibited a broad distribution extending across their coding region, whereas H3K4me3 was rarely deposited around their TSS regions (Fig. EV2E), in line with their silence in naïve hESCs. In fact, the binding of NANOG to these genes was weaker in naïve hESCs compared to their primed counterparts (Fig. EV2E), and *NANOG* depletion in naïve hESCs did not produce notable changes in their transcript levels. Taken together, our findings favor the notion that NANOG controls the transcriptional program of naïve hESCs through both direct binding to its target genes and specific modulation of epigenetic signatures.

## NANOG governs acetyl-CoA production and histone acetylation in naïve hESCs

As presented above, NANOG occupancy was positively correlated with H3K27ac modification, and *NANOG* deficiency led to a marked reduction in H3K27ac signals in naïve pluripotency genes. To learn the role of NANOG for histone acetylation in naïve hESCs further, we conducted western blot analysis and found substantial decreases in acetylation levels of histones H3 and H4, as well as H2BK5 and H3K27 after *NANOG* depletion in naïve *NANOG* iKO hESCs (Fig. 3A). To validate that the decrease in histone acetylation was indeed caused by *NANOG* depletion, we generated hESCs overexpressing *NANOG-mut* (NANOG coding sequence containing synonymous mutations to avoid gRNA targeting) by DOX induction in the background of *NANOG* iKO naïve hESCs, referred to as *iNANOG-mut_NANOG* iKO. In this hESC line, DOX treatment induced depletion of endogenous *NANOG* and expression of exogenous *NANOG-mut* simultaneously (Fig. 3B). Our results showed that *NANOG-mut* expression fully blocked the decline in histone acetylation caused by *NANOG* depletion (Fig. 3B), demonstrating the specific connection between NANOG and global histone acetylation in naïve hESCs. In contrast, *NANOG* deficiency elicited little change in the histone acetylation levels in primed hESCs (Fig. EV3A). Moreover, *NANOG* depletion did not alter methylation levels on specific lysine residues of H3 in naïve hESCs (Fig. EV3B). Thus, NANOG was specifically required for the appropriate histone acetylation in naïve hESCs. We then asked the question of how NANOG would execute this function. Given the centrality of acetyl-CoA as a substrate for histone acetylation, and undetectable changes in transcript levels of enzymes associated with histone acetylation or deacetylation after *NANOG* depletion (Fig. EV3C), we measured intracellular acetyl-CoA concentrations using the LC-MS analysis and found significant declines in acetyl-CoA concentrations after *NANOG* depletion in naïve hESCs (Fig. 3C), indicating an important role of NANOG in maintaining normal acetyl-CoA levels in naïve hESCs. Remarkably, supplementation with sodium acetate (NaAc), commonly utilized to boost intracellular acetylation, efficiently rescued the reduced histone acetylation levels caused by *NANOG* depletion (Fig. 3D). Therefore, the reduced acetyl-CoA level is primarily responsible for *NANOG* depletion-induced decline in global histone acetylation levels.

Next, to address the issue of whether NANOG- controlled intracellular acetyl-CoA/histone acetylation levels would contribute to its function in controlling transcriptional program, we treated *NANOG* iKO naïve hESCs with either DOX or DOX with NaAc supplementation for 1 or 2 days, and collected cells for RNA-seq. The transcriptome analysis showed that the addition of NaAc to DOX- treated cells (for both 1 and 2 days) enhanced the expression levels of certain naïve pluripotency genes, whereas it exerted little impact on *NANOG* depletion- induced upregulation of PrE and TE marker genes (Fig. EV3D). NaAc- induced significant increases in transcript levels of naïve markers (*ALPP* and *MAEL*) are shown (Fig. EV3E). Thus, the control of intracellular acetyl-CoA abundance presents one of the important mechanisms for NANOG to sustain naïve pluripotency.

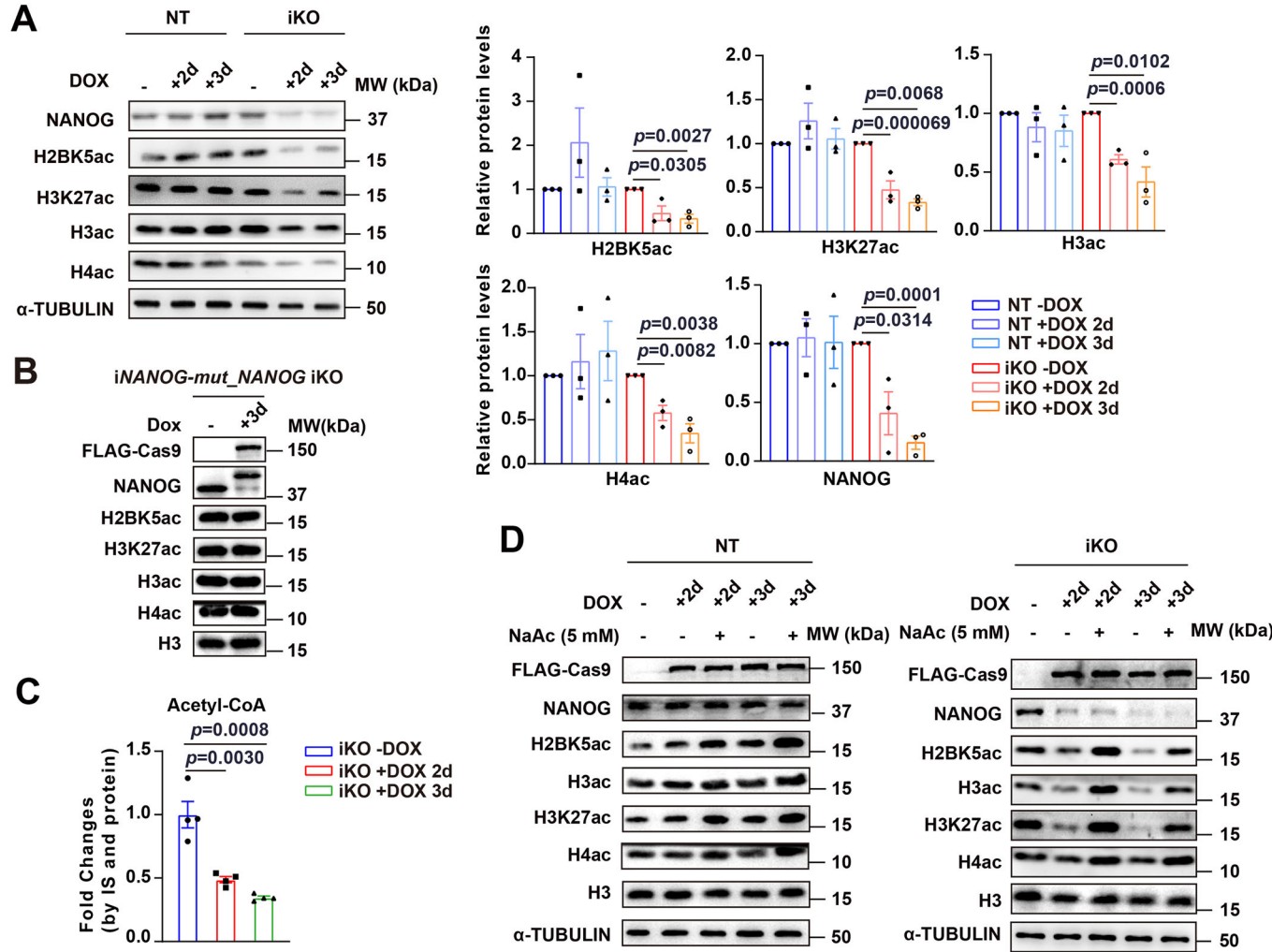

**Figure 3. NANOG- modulated acetyl-CoA production contributes to global histone acetylation in naïve hESCs.**

(A) A representative western blot analysis result for steady-state protein levels of NANOG and histone acetylation modifications in naïve NT and *NANOG* iKO SHhES8 hESCs, either untreated or treated with DOX for two or three days. Alpha-TUBULIN served as a loading control (left). The quantitative analysis results of relative protein levels from three independent experiments are shown on the right side. Data are presented as mean ± SEM ($n = 3$). (B) A representative western blot analysis result for protein levels of NANOG and histone acetylation modifications in naïve i*NANOG-mut_NANOG* iKO SHhES8 hESCs, either untreated or treated with DOX for three days. H3 served as a loading control. (C) The relative abundance of intracellular acetyl-CoA in naïve *NANOG* iKO hESCs, either untreated or treated with DOX for two or three days, as measured by the LC-MS analysis. Data are presented as mean ± SEM ($n = 3$). IS, internal standard. (D) A representative western blot analysis result for protein levels of NANOG and histone acetylation modifications on H3, H4, H2BK5, and H3K27 in naïve NT and *NANOG* iKO SHhES8 hESCs, either untreated or treated with DOX or DOX and NaAc. Alpha-TUBULIN and H3 served as loading controls. The unpaired two-tailed Student's *t*-test was used for the statistical analysis (A, C). Source data are available online for this figure.

We then asked how NANOG would maintain normal intracellular acetyl-CoA levels in naïve hESCs. To address this question, we carefully analyzed DEGs resulted from *NANOG* depletion and aligned them with genes occupied by NANOG in naïve hESCs, leading to the identification of 473 up- and 362 down- regulated putative target genes of NANOG (Fig. 4A). The upregulated targets were enriched for Gene Ontology (GO) terms associated with developmental processes, particularly tissue morphogenesis and cell mobility (Fig. EV4A, top), whereas GO terms enriched by downregulated targets included transcription regulation of pluripotent stem cells and signaling pathways regulating pluripotency of stem cells and metabolic processes (Fig. EV4A, bottom). Further analysis of putative target genes downregulated by *NANOG*

deficiency and involved in metabolism regulation (Figs. 4B and EV4B) identified six NANOG target genes, which are known to be associated with acetyl-CoA synthesis. This set of genes included pyruvate metabolism factors (*MPC2* and *DLAT*), acetate metabolism enzymes (*ALDH2* and *ACSS2*), and fatty acid oxidation regulators (*ACADS* and *CPT1A*) (Fig. EV4C). Our NANOG ChIP-seq data revealed a robust and specific NANOG association with the regulatory regions of these six genes in naïve hESCs (Fig. 4C), and ChIP-qPCR verified NANOG occupancy on *DLAT* and *MPC2* (Fig. 4D). These six genes were all highly expressed in control naïve hESCs, and evidently downregulated by *NANOG* depletion (Fig. 4E). The presence of H3K27ac and the absence of H3K27me3 marks in the regulatory regions of these six genes should underpin

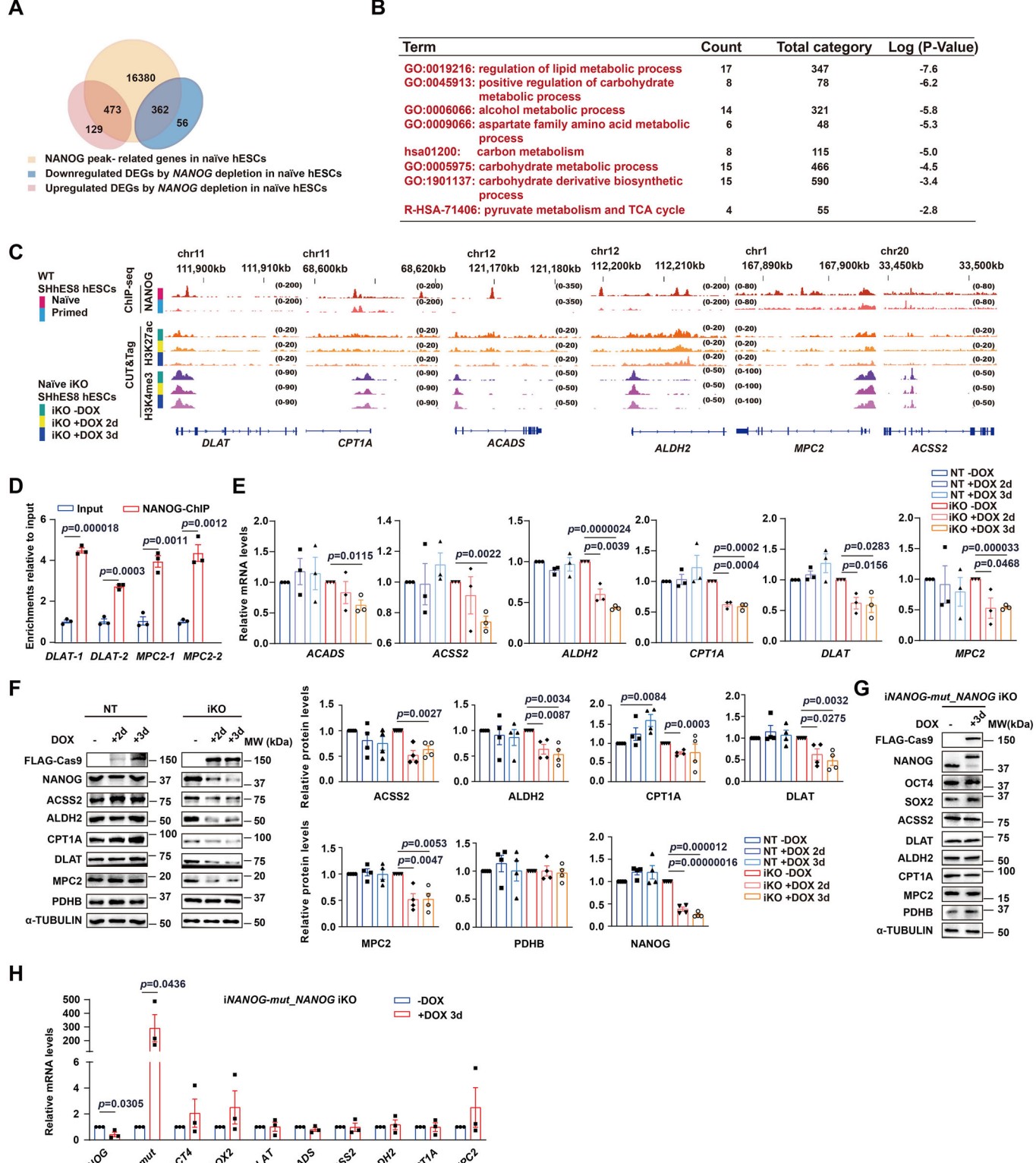

**Figure 4. NANOG activates multiple genes involved in acetyl-CoA synthesis in naïve hESCs.**

(A) A Venn diagram illustrating the overlap between NANOG- occupied genes and DEGs induced by *NANOG* deletion for three days in naïve hESCs. (B) The list of multiple metabolic pathways enriched by the GO analysis of 362 NANOG- occupied and *NANOG* depletion- downregulated DEGs in naïve hESCs. (C) Genome browser snapshots showing levels of NANOG occupancy near *DLAT, CPT1A, ACADS, ALDH2, MPC2*, and *ACSS2* in primed and naïve SHhES8 hESCs, as well as histone modifications and chromatin openness in naïve *NANOG* iKO SHhES8 hESCs, either untreated or treated with DOX for two or three days. (D) The results for ChIP-qPCR validation of NANOG occupancy on the promoter region of *DLAT* and *MPC2*. Two pairs of primers were used for each gene. Data are presented as mean ± SEM ($n = 3$). (E) RT-qPCR analysis results for relative mRNA levels of six NANOG target genes in naïve *NANOG* iKO hESCs, either untreated or treated with DOX for two or three days. Data are presented as mean ± SEM ($n = 3$). (F) The representative western blot analysis result for steady-state levels of proteins examined in naïve NT or *NANOG* iKO hESCs, either untreated or treated with DOX for two or three days. Alpha-TUBULIN served as a loading control (left). The quantitative analysis results of relative protein levels from four independent experiments are shown on the right side. Data are presented as mean ± SEM ($n = 4$). (G) The representative western blot analysis result for levels of proteins examined in naïve i*NANOG-mut_NANOG* iKO SHhES8 hESCs, either untreated or treated with DOX for three days. Alpha-TUBULIN served as a loading control. (H) RT-qPCR analysis results for relative mRNA levels of indicated genes in naïve i*NANOG-mut_NANOG* iKO SHhES8 hESCs, either untreated or treated with DOX for three days. Data are presented as mean ± SEM ($n = 3$). The unpaired two-tailed Student's *t*-test was used for the statistical analysis in (D–F, H). Source data are available online for this figure.

their active transcription in naïve hESCs, as *NANOG* depletion gave rise to simultaneous decreases in their transcription and H3K27ac enrichment (Fig. 4C,E). *NANOG* depletion- induced reduction in protein levels of ACSS2, ALDH2, DLAT, and MPC2 was consistently observed in different naïve hESC lines (SHhES8 and H9) under different culture conditions (5iLA and PXGL media), with concomitant decreases in global histone acetylation levels (Figs. 4F and EV4D,E). The decrease in the CPT1A protein level was variable between hESC lines (Figs. 4F and EV4D,E), although its downregulation by *NANOG* depletion was invariably detected at the transcriptional level (Fig. 4E). Thus, the contribution of CPT1A to NANOG-mediated regulation of cellular acetyl-CoA levels in different hESC lines needs further investigation. Unfortunately, we failed to examine the protein level of ACADS due to the poor quality of its antibody. As a negative control, the protein level of PDHB was not altered by *NANOG* depletion, although PDHB and DLAT exist in the same protein complex and are both important for the conversion of pyruvate to acetyl-CoA (Bhandary and Aguan, 2015), indicating the selectivity of NANOG in regulating the expression of genes important for acetyl-CoA synthesis. The close connection between NANOG and histone acetylation, as well as expression of six genes related to acetyl-CoA synthesis was further proven by our results obtained from our newly generated DOX- inducible *NANOG* knockdown SHhES8 hESC lines, referred to as *NANOG* iKD. In this naïve hESC line, DOX treatment induced a drastic reduction of *NANOG* expression levels (Fig. EV4F,G), with simultaneous decreases in the transcriptional levels of the six NANOG targets (*ACADS, ACSS2, ALDH2, CPT1A, DLAT*, and *MPC2*) (Fig. EV4F). Meanwhile, global histone acetylation levels decreased upon DOX treatment for three days in naïve *NANOG* iKD hESCs, in parallel with the reduction in the protein levels of ACSS2, ALDH2, CPT1A, DLAT, and MPC2 (Fig. EV4G). Specifically, DOX- induced overexpression of *NANOG-mut* prevented *NANOG* depletion- provoked reduction in protein levels of OCT4 and SOX2 as well as five NANOG targets (ACSS2, ALDH2, CPT1A, DLAT, and MPC2) (Fig. 4G). The overexpression of *NANOG-mut* also abolished *NANOG* depletion-induced reduction in transcript levels of *OCT4, SOX2*, and all six NANOG target genes related to acetyl-CoA production (Fig. 4H). Together, we conclude that NANOG specifically activates the transcription of important target genes to sustain intracellular acetyl-CoA homeostasis and epigenetic landscapes.

## NANOG is indispensable for the normal activity of mitochondrial respiration and glycolysis in naïve hESCs

On the one hand, our RNA-seq data and RT-qPCR results revealed that many tricarboxylic acid (TCA) cycle and OXPHOS genes were expressed at significantly higher levels in naïve hESCs than in primed ones (Fig. EV5A,B). On the other hand, *NANOG* depletion resulted in downregulation of key metabolic genes, including *CPT1A, ACADS, MPC2, DLAT, G6PD*, and *ACSS2* (Fig. EV4B). Therefore, we wanted to know whether NANOG would participate in the control of energy metabolism in naïve hESCs. Our LC-MS analysis showed that *NANOG* depletion markedly attenuated the abundance of glycolytic metabolites, including F6P, DHAP, GAP, PEP, pyruvate and lactate, as well as TCA cycle metabolites, such as succinate and citric acid (Fig. 5A,B). In contrast, levels of aspartate and glutamine were elevated by *NANOG* depletion (Fig. 5C). To determine whether *NANOG* depletion would affect energy metabolism in naïve hESCs, we measured the oxygen consumption rate (OCR) and extracellular acidification rate (ECAR) to evaluate mitochondrial respiration and glycolytic activities, respectively. Clearly, basal respiration, maximal respiration and spare respiration capacity were all significantly impaired by *NANOG* depletion (Fig. 5D,E). Similarly, glycolytic function was evidently disturbed in *NANOG*- depleted cells (Fig. 5F). In addition, *NANOG* depletion also diminished the level of reduced NADH significantly (Fig. EV5C). Collectively, these findings reveal, for the first time, a role of NANOG acting as an indispensable regulator of the bivalent metabolic state in naïve hESCs.

## *GPX2* is identified as a key target of NANOG in protecting naïve hESCs against oxidative stress

As shown in the preceding section, depletion of *NANOG* in naïve hESCs provoked massive cell death, particularly being evident on day 3 of DOX addition, highlighting an essential role of NANOG in the survival of naïve hESCs. To identify the direct target of NANOG that mediated this protective function, we overlapped genes bound by NANOG at their promoter regions uniquely in naïve hESCs with genes upregulated or downregulated by *NANOG* depletion in naïve hESCs. This analysis generated a group of 163 NANOG target genes unique in naïve hESCs (Fig. 6A,B). They were divided into four clusters based on their relative expression profiles

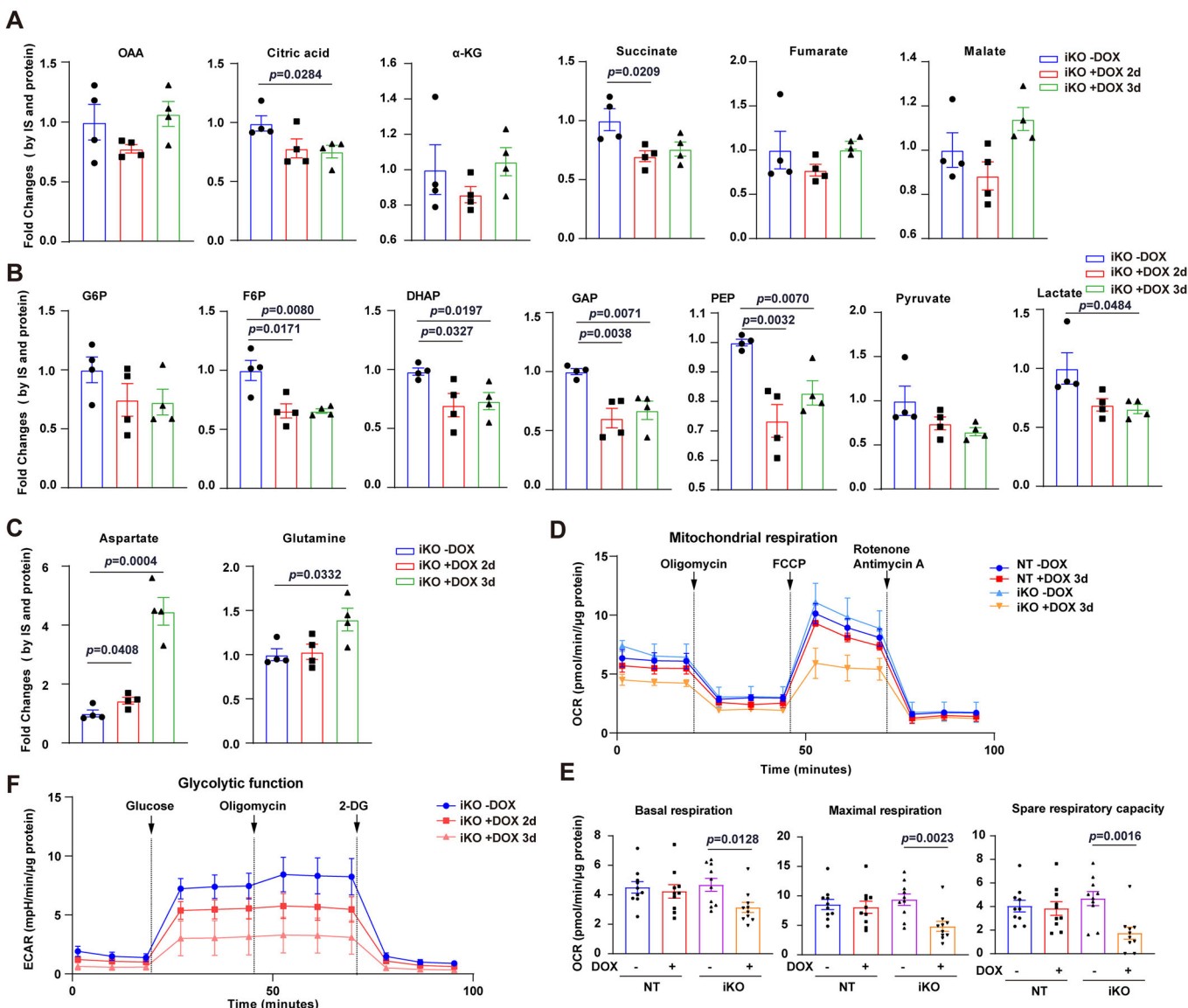

**Figure 5. NANOG is required for the normal activity of glycolysis and OXPHOS in naïve hESCs.**

(A–C) Column charts showing the relative abundance of different metabolites from the TCA cycle (A) and glycolysis (B), as well as aspartate and glutamine (C) in naïve *NANOG* iKO SHhES8 hESCs, either untreated or treated with DOX for two or three days, determined by the LC-MS analysis. Data are presented as mean ± SEM (*n* = 4). IS, internal standard. (D, E) Representative kinetic profiles of the OCR (D) and quantification analysis of basal respiration, maximal respiration, and spare respiratory capacity (E) in naïve NT and *NANOG* iKO hESCs, either untreated or treated with DOX for three days. Data are shown as mean ± SEM (*n* = 10 from two biological replicates). (F) Representative kinetic profiles of the ECAR in naïve *NANOG* iKO hESCs, either untreated or treated with DOX for two or three days. Data are presented as mean ± SEM (*n* = 11 from three biological replicates). The unpaired two-tailed Student's *t*-test was used for the statistical analysis in (A–C, E). Source data are available online for this figure.

in primed and naïve *NANOG* iKO SHhES8 hESCs, with the presence or absence of DOX for three days (Fig. 6C). According to the functional annotation of these 4 clusters (Figs. 6D and EV6A), our attention was particularly drawn to cluster 4, where genes exhibited higher expression levels in naïve hESCs than in primed hESCs and were downregulated by *NANOG* depletion. Importantly, genes of the cluster 4 enriched the term of response to oxidative stress (Fig. 6D). Particularly, five genes (*SLC23A2, G6PD, GPX2, HYAL1,* and *GPNMB*) known to play roles in the maintenance of intracellular redox homeostasis (Ho et al, 2014; Linowiecka et al,

2020; Pei et al, 2023; Wang et al, 2021) were included in this cluster. Our ChIP-seq data clearly displayed specific NANOG occupancy in their regulatory regions in naïve hESCs but not in primed ones (Fig. EV6B). NANOG association with their promoter regions was further validated by ChIP-qPCR assays (Fig. 6E). Meanwhile, decreases in their transcript levels and H3K27ac deposition at these genes were concomitantly observed after *NANOG* depletion (Figs. 6F and EV6B). Furthermore, with *NANOG* iKD naïve hESCs, we were able to validate that *NANOG* knockdown reduced the expression levels of these five NANOG target genes (*SLC23A2,*

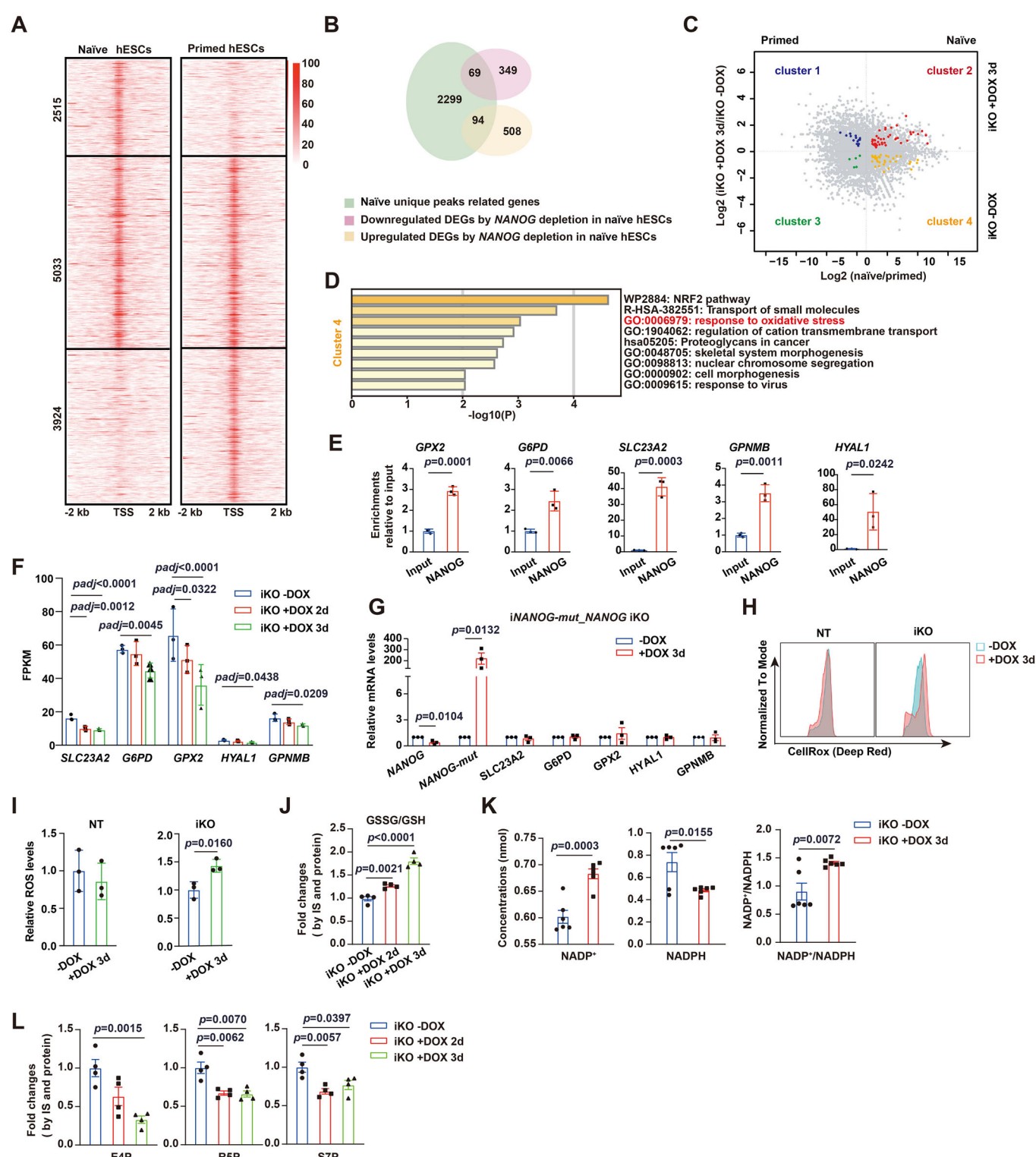

G6PD, GPX2, HYAL1, and GPNMB) in naïve hESCs, with a minor reduction in the GPNMB level (Fig. EV6C). Meanwhile, the protein level of GPX2 decreased drastically after NANOG knockdown by DOX treatment for three days in naïve NANOG iKD hESCs (Fig. EV6D). In addition, in line with our finding in naïve NANOG iKO hESCs (Fig. EV1D), NANOG knockdown elevated cleaved Caspase-

3 levels in naïve NANOG iKD hESCs (Fig. EV6D). Specifically, NANOG-mut overexpression abolished the reduction in the transcript levels of the five target genes caused by NANOG depletion (Fig. 6G). These results lead to the conclusion that NANOG could directly activate antioxidant genes specifically in naïve hESCs.

**Figure 6. NANOG directly activates antioxidant genes to protect naïve hESCs against oxidative stress and cell death.**

(A) The heatmaps showing NANOG binding peaks around TSS in primed and naïve SHhES8 hESCs. (B) The Venn diagram showing the overlap between DEGs induced by *NANOG* depletion and genes occupied by NANOG uniquely in naïve hESCs. (C) Relative expression levels of putative NANOG target genes unique in naïve hESCs between naïve and primed hESCs and in the presence or absence of DOX. The abscissa is log2 (naïve/primed), and the ordinate is log2 (iKO +DOX 3d/iKO -DOX). (D) The bar chart showing functional annotation of genes from cluster 4 described in (C). The hypergeometric test was used for statistical analysis. (E) ChIP-qPCR results showing the NANOG occupancy on the promoter region of oxidative stress- related genes in naïve hESCs. Data are presented as mean ± SD ($n = 3$). (F) FPKM values of five NANOG-regulated oxidative stress- related genes in naïve *NANOG* iKO SHhES8 hESCs, either untreated or treated with DOX for two or three days. Data are presented as mean ± SD ($n = 3$). *SLC23A2*, Row 1, $p = 2.41 \times 10^{-8}$; *GPX2*, Row 1, $p = 6.10 \times 10^{-6}$. (G) RT-qPCR analysis results for relative mRNA levels of the indicated genes in naïve i*NANOG-mut_NANOG* iKO SHhES8 hESCs, either untreated or treated with DOX for three days. Data are presented as mean ± SEM ($n = 3$). (H, I) The representative ROS peak graph (H) and quantitative analysis of relative ROS levels (I) calculated by CellROX staining in the combination with flow cytometric analysis in naïve NT and *NANOG* iKO SHhES8 hESCs, either untreated or treated with DOX. Data are shown as mean ± SD ($n = 3$). (J) The LC-MS analysis results for GSSG/GSH ratios in naïve *NANOG* iKO SHhES8 hESCs, either untreated or treated with DOX for two or three days. Data are presented as mean ± SEM ($n = 4$). Row 1, $p = 7.39 \times 10^{-5}$. IS, internal standard. (K) The LC-MS analysis results for the abundance of NADP$^+$ and NADPH, as well as the ratio of NADP$^+$/NADPH in naïve *NANOG* iKO SHhES8 hESCs, either untreated or treated with DOX for three days. Data are presented as mean ± SEM ($n = 6$). (L) The LC-MS analysis results for relative abundances of three indicated metabolites in the PPP pathway in naïve *NANOG* iKO SHhES8 hESCs, either untreated or treated with DOX for two or three days. Data are presented as mean ± SEM ($n = 4$). IS, internal standard. The unpaired two-tailed Student's *t*-test was used for the statistical analysis in (E–G, I–L). Source data are available online for this figure.

Based on the prominent role of NANOG in regulating antioxidant gene expression, we examined whether NANOG would play a role in the maintenance of redox homeostasis in naïve hESCs. We found that *NANOG* depletion significantly elevated total cellular ROS levels (Fig. 6H,I). Concomitantly, ratios of GSSG/GSH (Fig. 6J) and NADP$^+$/NADPH (Fig. 6K) increased evidently, suggesting the occurrence of oxidative stress after *NANOG* depletion. The elevated ratios of NADP$^+$/NADPH after *NANOG* depletion could be attributed to decreased expression of *G6PD* (Fig. 6F), which is known to play a pivotal role in the pentose phosphate pathway (PPP) for generating NADPH, a crucial molecule for cellular antioxidant defense (Teslaa et al, 2023). In agreement with this explanation, our LC-MS analysis showed decreased abundance of some PPP metabolites (E4P, R5P, and S7P) (Fig. 6L). Hence, NANOG is required to maintain redox homeostasis in naïve hESCs.

We next attempted to identify the key mediator of NANOG in securing redox homeostasis in naïve hESCs. *GPX2* was shown to be highly expressed in the epiblast of human preimplantation blastocysts in vivo, sharing a similar expression pattern with *NANOG* (Fig. 7A). In this study, we found that *GPX2* was preferentially expressed in naïve hESCs, being barely detectable in primed hESCs (Fig. 7B,C), and that GPX2 expression paralleled well with NANOG expression (Fig. 7D,E). GPX2 is a member of the selenium- dependent glutathione peroxidase (GPX) family. These enzymes act to reduce hydroperoxides at the expense of reduced glutathione, representing one tier of the cellular protective system against ROS (Ufer and Wang, 2011). Of members of this protein family, GPX2 is the only protein specifically expressed in blastocysts of human preimplantation embryos (Fig. EV7A). Therefore, we hypothesized that GPX2 could be a functional mediator of NANOG to protect naïve hESCs against oxidative stress. To test this hypothesis, we generated a primed hESC line, which could be induced to express exogenous *GPX2* and delete *NANOG* by DOX simultaneously (iKOE), through inserting an inducible *GPX2* overexpression (OE) cassette into the *ROSA26* allele of our established *NANOG* iKO primed hESCs with the CRISPR- engineered system (Fig. EV7B). An inducible *GPX2* OE alone primed hESC line (iOE) was also established. The efficient simultaneous induction of *NANOG* depletion and *GPX2* OE by DOX in the iKOE line and induction of *GPX2* OE in the iOE line were, respectively, verified in primed hESCs (Fig. EV7C,D). Both lines were then converted to the naïve state using the 5iLA system

and used in the following experiments. Consistent with the results in the preceding sections, the depletion of *NANOG* led to elevated ROS levels (Fig. 7F) and reduced numbers of naïve hESC colonies (Fig. EV7E), enhanced levels of cleaved Caspase-3 (Fig. EV7F), and increased percentages of annexin V$^+$/PI$^+$ cells (Fig. 7G). Strikingly, enforced expression of *GPX2* fully rescued *NANOG* depletion caused elevation of ROS levels (Fig. 7F). Similar effects of *GPX2* OE on *NANOG* depletion- elevated ROS levels were obtained in naïve hESCs cultured under the PXGL condition (Fig. EV7G). In addition, *GPX2* OE rescued *NANOG* depletion- induced cell death partially (Figs. 7G and EV7E,F). Taken together, these findings support the notion that NANOG directly controls a set of antioxidant genes, and that, among these target genes, GPX2 is a key mediator of NANOG in sustaining redox homeostasis and survival of naïve hESCs.

## Discussion

Metabolic processes and metabolites critically regulate chromatin configuration and cell fate, yet how core transcriptional factors maintain gene regulatory networks by modulating metabolism and epigenetic signature in naïve hESCs remains unclear. Here, we focus on NANOG, a core transcription factor highly expressed in naïve hESCs, and propose a model (Fig. 7H), where (1) NANOG activates genes associated with naïve pluripotency directly and represses genes associated with differentiation towards extraembryonic lineages; (2) genome- wide NANOG occupation is highly correlated with active transcriptional marks, especially co-distributing with H3K27ac, maintaining the epigenetic landscape essential for gene expression networks in naïve hESCs; (3) NANOG transcriptionally controls multiple genes acting in different metabolic pathways for acetyl-CoA production, thereby modulating intracellular acetyl-CoA levels and histone acetylation modifications; (4) NANOG is an indispensable regulator for the bivalent metabolic state in naïve hESCs; (5) NANOG protects naïve hESCs against oxidative stress through directly activating antioxidant genes, particularly *GPX2*. Therefore, this study deciphers how NANOG acts as an upstream guardian of human naïve pluripotency through modulating acetyl-CoA synthesis, energy metabolism, epigenetic signature and redox homeostasis. The new target genes identified here provide rational explanations for how

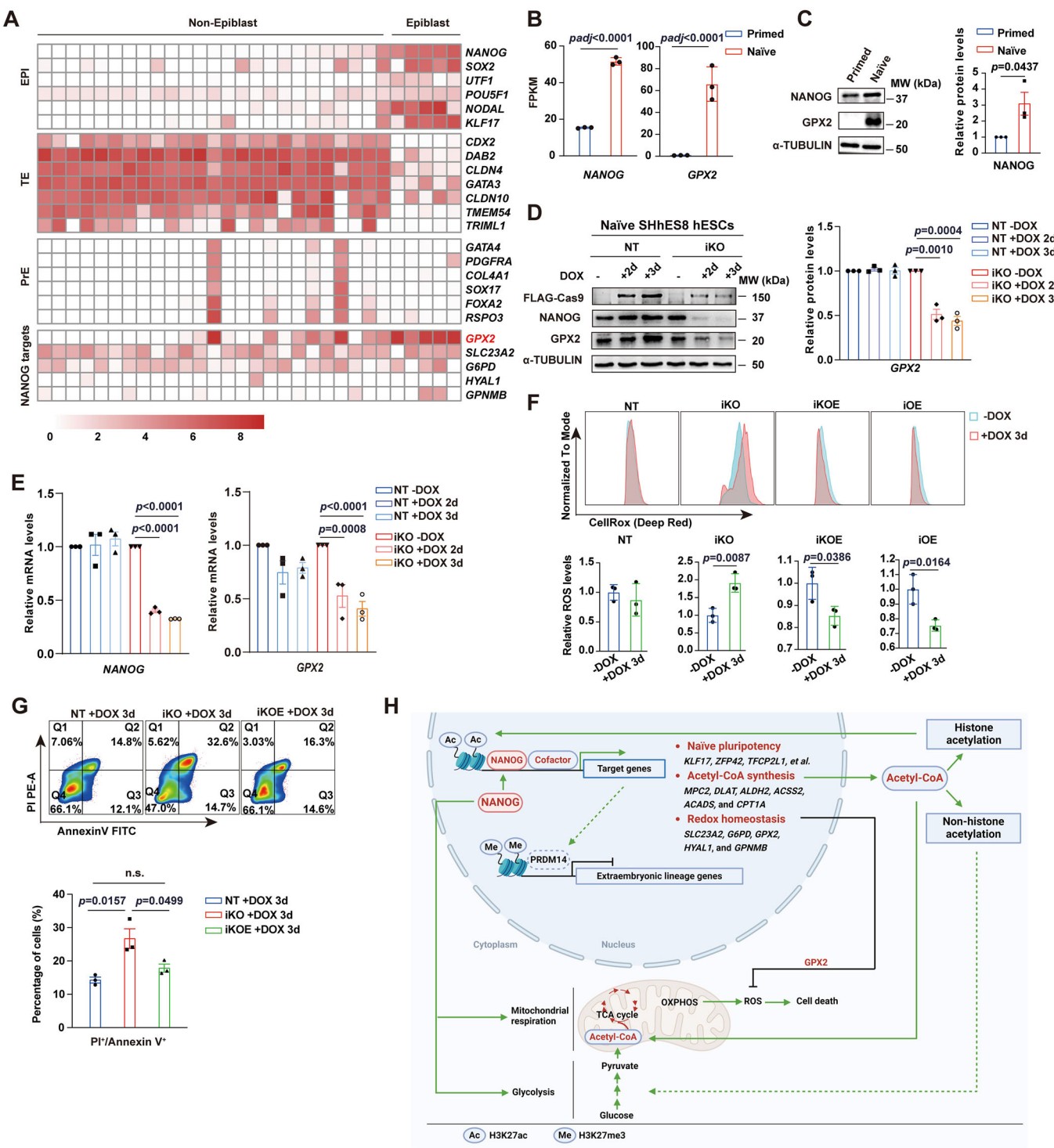

NANOG executes these functions uniquely in naïve hESCs, and provides insights into the regulation of metabolism and redox homeostasis in the early human development.

Previous studies have demonstrated that NANOG is crucial for keeping ESCs in an undifferentiated pluripotency state. In naïve mESCs, Nanog binding sites were predominantly found near genes that are specific to and maintain the naïve pluripotency state (Costa et al, 2013;

Loh et al, 2006). Nanog directly represses *Gata6* expression, maintaining pluripotency and inhibiting primitive endoderm differentiation (Frankenberg et al, 2011; Singh et al, 2007). In addition, to suppress trophectoderm differentiation, Nanog binds and represses *Cdx2* in mESCs (Chen et al, 2009). In primed hESCs, NANOG works with other key transcription factors like OCT4 and SOX2 to prevent differentiation and sustain self-renewal (Boyer et al, 2005; Wang et al, 2012; Xu et al, 2008). Specifically,

**Figure 7. NANOG contributes to the redox homeostasis through modulating *GPX2* expression in naïve hESCs.**

(A) Heatmaps showing mRNA levels of selected genes related to the epiblast, PrE, and TE, as well as putative NANOG targets related to antioxidation in the epiblast cells or non-epiblast cells from human late blastocysts. Data were obtained from the published dataset (GEO: GSE36552). (B) FPKM values of *NANOG* and *GPX2* in primed and naïve SHhES8 hESCs from our RNA sequencing data. Data are shown as mean ± SD ($n = 3$). *NANOG*, $p = 5.95 \times 10^{-97}$; *GPX2*, $p = 1.24 \times 10^{-108}$. (C) The representative western blot analysis result for GPX2 and NANOG protein levels in primed and naïve hESCs (left). Alpha-TUBULIN served as a loading control. The quantitative analysis results of the relative protein levels of NANOG are shown on the right side. Data are presented as mean ± SEM ($n = 3$). (D) The representative western blot analysis result for the steady-state levels of indicated proteins in naïve NT and *NANOG* iKO SHhES8 hESCs, either untreated or treated with DOX for two or three days. Alpha-TUBULIN served as a loading control (left). The quantitative analysis result of relative GPX2 protein levels is shown on the right side. Data are shown as mean ± SEM ($n = 3$). (E) RT-qPCR analysis results for relative mRNA levels of *GPX2* and *NANOG* in naïve NT and *NANOG* iKO hESCs, either untreated or treated with DOX for two or three days. Data are presented as mean ± SEM ($n = 3$). *NANOG*, $p = 8.32 \times 10^{-11}$ (Row 1), $p = 4.69 \times 10^{-6}$ (Row 2); *GPX2*, $p = 2.34 \times 10^{-5}$ (Row 1). (F) The representative ROS peak graph measured by the CellROX staining combined with the flow cytometric analysis in naïve NT, iKO, iKOE, and iOE hESCs, either untreated or treated with DOX for three days (top). Quantitative analysis results of relative ROS levels are shown at the bottom. Data are shown as mean ± SD ($n = 3$). (G) Representative images of the Annexin V/PI flow cytometric analysis for naïve NT, iKO, and iKOE SHhES8 hESCs treated with DOX for three days (top). The quantitative analysis of data are shown at the bottom. Data are shown as mean ± SD ($n = 3$). (H) A model for the function and mechanisms of NANOG in naïve hESCs. Solid lines represent the tested function, while dashed lines represent the hypothesized function. The unpaired two-tailed Student's *t*-test was used for the statistical analysis in (B, C, E–G). Source data are available online for this figure.

NANOG represses differentiation toward the embryonic ectoderm lineage, helping primed hESCs avoid premature specialization (Wang et al, 2012). In this study, we found that NANOG was enriched near the primed pluripotency genes and inhibited the expression of certain marker genes of the three major embryonic germ layers, especially for ectoderm and endoderm in primed hESCs. Compared to mESCs and primed hESCs, the function of NANOG is much less investigated in naïve hESCs. Our study demonstrates that NANOG preferentially occupied active promoters and enhancers in naïve hESCs. We also found that NANOG was less prevalent around promoters of TE and PrE genes. Exceptionally, NANOG was enriched near the PrE gene *GATA6* and TE genes, *CDX2* and *GATA3*. The identification of *GATA3* as a new NANOG target gene here suggests that, compared to mESCs, NANOG could directly repress additional lineage genes to ensure human naïve pluripotency. It is interesting to elucidate how NANOG executes pluripotency state-dependent functions in future studies.

As a transcriptional factor, NANOG controls gene expression primarily through directly binding to its target genes in naïve hESCs, including certain naïve pluripotency markers, extraembryonic lineage markers, and metabolic genes. However, our analyses revealed that *NANOG* depletion- upregulated not only NANOG- occupied lineage- specific genes but also those TE and PrE genes lacking nearby NANOG enrichments, indicating that NANOG also inhibits the expression of lineage markers through epigenetic modifications. The finding is consistent with previous reports in mESCs (Chalut et al, 2012; Costa et al, 2013; Wang et al, 2006). In line with this view, Heurtier et al. dissected how Nanog orchestrates pluripotency at the transcriptional and epigenetic levels in mESCs (Heurtier et al, 2019). They showed that, in the absence of LIF, enhanced expression of endogenous *Nanog* could block differentiation by sustaining H3K27me3 at developmental regulators, particularly *Otx2*, to indirectly repress a large set of genes involved with differentiation (mainly in nervous system development). However, in the presence of LIF, the induction of *Nanog* had minor effects on H3K27me3 deposition at these genes. Herein, we found that *NANOG* depletion in naïve hESCs reduced H3K27me3 enrichments across promoter regions of bivalent genes globally, including bivalent promoters of certain TE and PrE genes, whereas the expression of *OTX2* remained unchanged. These findings underscore that NANOG modulates gene expression programs at an epigenetic level through both conserved and divergent mechanisms between naïve hESCs and mESCs. We are

currently unclear how H3K27me3 at these specific genomic sites is modulated by NANOG. Among putative NANOG target genes identified in this study, PRDM14, a determinant of hESC identity (Nakaki and Saitou, 2014), stood out. It is known to repress developmental genes by recruiting PRC2 for H3K27me3 modification (Chan et al, 2013). PRDM14 was also reported to be associated with polycomb repressive complex 1.3 (PRC1.3) and represses developmental genes, safeguarding naïve reprogramming (Collier et al, 2022). We found that the *PRDM14* transcript level was higher in naïve hESCs than in primed hESCs, and it was downregulated by *NANOG* depletion. It is possible that NANOG could activate PRDM14 to establish repressive histone marks and silence PrE and TE genes. Further investigation is needed to test this possibility.

Intracellular acetyl-CoA levels have been shown to influence cell fate in primed hESCs (Moussaieff et al, 2015). However, the question of whether acetyl-CoA participates in human naïve pluripotency maintenance was not previously addressed. This study not only presents the first experimental evidence for the functional role of acetyl-CoA in naïve hESCs, but also identifies the key regulator for acetyl-CoA homeostasis in naïve hESCs. Our results highlight the role of the metabolite in NANOG- mediated function and the significance of the NANOG-acetyl-CoA-H3K27ac axis for the maintenance of the naïve transcriptional program. Of note, a recent study revealed that the level of H3K27ac modification was significantly higher in naïve hESCs than in primed hESCs (Li et al, 2024). Thus, our discovery of a link between NANOG and H3K27ac modification offers molecular insights into how this distinctive epigenetic feature might be regulated and suggests that modulating H3K27ac modification is likely a major mechanism for NANOG to govern gene expression and human naïve pluripotency. Nevertheless, the broader impact of acetyl-CoA on non-histone protein acetylation in pluripotency maintenance warrants further exploration.

Another contribution of the current study is to discover that NANOG safeguards naïve hESCs from oxidative stress through directly activating a set of antioxidant genes, such as *G6PD* and *GPX2*. Importantly, we identified GPX2 as a key mediator of NANOG in scavenging ROS and maintaining survival of naïve hESCs. Downregulation of both *G6PD* (a key enzyme for the PPP pathway to produce NADPH) (Teslaa et al, 2023) and *GPX2* are likely responsible for the elevated ratios of GSSG/GSH and NADP$^+$/NADPH caused by *NANOG* depletion in naïve hESCs.

Taken together, our findings not only advance our understanding of how gene transcription, metabolism and metabolites, as well as epigenetic landscape and redox homeostasis interplay to sustain naïve pluripotency, but also aid to develop new strategies to optimize the hESC primed-to-naïve transition system and promote future applications of hPSCs in regenerative medicine.

# Methods

### Reagents and tools table

| Reagent/resource | Reference or source | Identifier or catalog number |
|---|---|---|
| **Oligonucleotides and other sequence-based reagents** | | |
| gRNA target | Sequences (5'-3') | |
| AAVS1 | gA: GTCCCTAGTGGCCCCACTGT<br>gB: GACAGAAAAGCCCCATCCTT | N/A |
| ROSA26 | gL: GTCGAGTCGCTTCTCGATTA<br>gR: GGCGATGACGAGATCACGCG | N/A |
| NANOG (for iKO) | TTCTGCTGAGATGCCTCACA | N/A |
| NANOG (for iKD) | CCAGCAGAACGTTAAAATCC | N/A |
| Non-targeting (NT) | GGAGACGGACGTCTCC | N/A |
| **RT-qPCR primers** | | |
| Gene ID | Sequence (5'-3') | |
| ACTB | Fw: CATGTACGTTGCTATCCAGGC<br>Rv: CTCCTTAATGTCACGCACGAT | N/A |
| NANOG | Fw: CCTGTGATTTGTGGGCCTGA<br>Rv: AGTCTCCGTGTGAGGCATCT | N/A |
| NANOG-mut | Fw: AGTGTGGATCCAGCTTGTCC<br>Rv: TATGGGGCATTTCCGCGCTA | N/A |
| OCT4 | Fw: ACATCAAAGCTCTGCAGA AAGAACT<br>Rv: CTGAATACCTTCCCAAATA GAACCC | N/A |
| SOX2 | Fw: TTGTTCGATCCCAACTTTCC<br>Rv: ACATGGATTCTCGGCAGACT | N/A |
| PRDM14 | Fw: ACACGCCTTTCCCGTCCTA<br>Rv: GGGCAGATCGTAGAGAGGCT | N/A |
| KLF17 | Fw: GCTGCCCAGGATAACGAGAAC<br>Rv: ATCTCTGCGCTGTGAGGAAAG | N/A |
| DNMT3L | Fw: ATGAAGTCAAGGCTAACCAGC<br>Rv: CGTCATCGTCGTACAGGAAGAG | N/A |
| SOX17 | Fw: GTGGACCGCACGGAATTTGR<br>Rv: GGAGATTCACACCGGAGTCA | N/A |
| GATA3 | Fw: CCGGTCCAGCACAGAAGGCA<br>Rv: ATAGAGCCCGCAGGCGTTGC | N/A |
| GATA6 | Fw: CCCACAACACAACCTACAGC<br>Rv: GCGAGACTGACGCCTATGTA | N/A |
| HAND1 | Fw: GTGCGTCCTTTAATCCTCTTC<br>Rv: GTGAGAGCAAGCGGAAAAG | N/A |
| STELLA | Fw: GTTACTGGGCGGAGTTCGTA<br>Rv: TGAAGTGGCTTGGTGTCTTG | N/A |
| KLF2 | Fw: CTACACCAAGAGTTCGCATCTG<br>Rv: CCGTGTGCTTTCGGTAGTG | N/A |
| KLF5 | Fw: CCTGGTCCAGCAAGATGTGA<br>Rv: GAACTGGTCTACGACTGAGGC | N/A |
| REX1 | Fw: GGAATGTGGGAAAGCGTTCGT<br>Rv: CCGTGTGGATGCGCACGT | N/A |
| HORMAD1 | Fw: AGCAACGAATCTAGCATGTTGT<br>Rv: TCACCATCCTTAAAACCGGGA | N/A |

| Reagent/resource | Reference or source | Identifier or catalog number |
|---|---|---|
| TFCP2L1 | Fw: CGTTTAAGCAGAACG AGAATGGG<br>Rv: TTTCATAGGACGGCTGG TATTTC | N/A |
| NLRP9 | Fw: TTGAATGACGCATATACTGCTGC<br>Rv: GTCCTTCCATAAGTTTCCCTCTG | N/A |
| KHDC3L | Fw: CCTTTCCTGAGCCTTTCGCC<br>Rv: AGGGCCTTTCCTGAGCCTTTC | N/A |
| SENM5B | Fw: CCGTGGGTCTCTAACTTCACC<br>Rv: GACTCGCACGTAGTTCTGACA | N/A |
| CNTN1 | Fw: CAGCCCTTTCCCGGTTTACAA<br>Rv: TGCTTCTGACCATCCCGTAGT | N/A |
| ZIC2 | Fw: CACCTCCGATAAGCCCTATCT<br>Rv: GGCGTGGACGACTCATAGC | N/A |
| ZIC3 | Fw: CGGCGCACGATCTATCTTCAG<br>Rv: TGGCGGAACAGAAACTCGC | N/A |
| GRPR | Fw: TCCTCCTAATAACGTGTGCTCC<br>Rv: GACACCCCAACAGAGGTAAGC | N/A |
| SLC23A2 | Fw: CTTCACTCTTCCGGTGGTGAT<br>Rv: TTTCCGTAGTGTAGATCGCCA | N/A |
| GPNMB | Fw: AAGATTGCCACTTGATGCCG<br>Rv: TCCCTCATGTAAGCAGAAGGTC | N/A |
| HYAL1 | Fw: CGATATGGCCCAAGGCTTTAG<br>Rv: ACCACATCGAAGACACTGACAT | N/A |
| G6PD | Fw: ACCGCATCGACCACTACCT<br>Rv: TGGGGCCGAAGATCCTGTT | N/A |
| GPX2 | Fw: GGTAGATTTCAATA CGTTCCGGG<br>Rv: TGACAGTTCTCCTG ATGTCCAAA | N/A |
| ACADS | Fw: CGGCAGTTACACACCATCTAC<br>Rv: GCAATGGGAAA CAACTCCTTCTC | N/A |
| ACSS2 | Fw: AAAGGAGCAAC TACCAACATCTG<br>Rv: GCTGAACTGACACACTTGGAC | N/A |
| ALDH2 | Fw: ATGGCAAGCCCTATGTCATCT<br>Rv: CCGTGGTACTTATCAGCCCA | N/A |
| CPT1A | Fw: ATGCGCTACTCCCTGAAAGTG<br>Rv: GTGGCACGACTCATCTTGC | N/A |
| MPC2 | Fw: CTCACCCAGGTAGCGACTCC<br>Rv: GTTCTCGTCCCTGGCTGTTG | N/A |
| DLAT | Fw: CGGAACTCCACGAGTGACC<br>Rv: CCCCGCCATACCCTGTAGT | N/A |
| OGDH | Fw: CATCGACAAATCCAGCGAGAT<br>Rv: TCCTCTCATGGTACATGCCC | N/A |
| SDHB | Fw: ACAGCTCCCCGTATCAAGAAA<br>Rv: GCATGATCTTCGGAAGGTCAA | N/A |
| SDHC | Fw: CAAAGAAGAG ATGGAGCGGTTC<br>Rv: TCCTAGGTCCCACATCTGCAC | N/A |
| PDHB | Fw: AAGAGGCGCTTTCACTGGAC<br>Rv: ACTAACCTTGTATGCCCCATCA | N/A |
| SUCLG1 | Fw: TATGGCACCAAACTCGTTGGA<br>Rv: GAAGCCGTTGCTCCTGTCT | N/A |
| SUCLA2 | Fw: TCTCCGTTCCCAAAGGATATGT<br>Rv: CACCAGCTAAAACCTGTGCC | N/A |
| IDH3G | Fw: AAACAATTCCTCCGTCCGCT<br>Rv: GCCTGAAGACGGACTTGACA | N/A |
| CS | Fw: TGCTTCCTCCACGAATTTGAAA<br>Rv: CCACCATACATCATGTCCACAG | N/A |
| ACO1 | Fw: CGCAGCACAAGA ACATAGAAGT<br>Rv: CATTGCAGCAAAGTCAACCAC | N/A |

| Reagent/resource | Reference or source | Identifier or catalog number |
|---|---|---|
| ACO2 | Fw: CCCTACAGCCTACTGGTGACT<br>Rv: TGTACTCGTTGGGCTCAAAGT | N/A |
| FH | Fw: CGGTCAGGTCTGGGAGAATTG<br>Rv: ACACTGAGTAGGGTTCACCTT | N/A |
| MDH1 | Fw: ACCATGCCAAGGTGAAATTGC<br>Rv: ACAGTCGTGACAAATTCTCCC | N/A |
| MDH2 | Fw: GCCATGATCTGCGTCATTGC<br>Rv: CCGAAGATTTTGTTGGGGTTGT | N/A |
| **ChIP-qPCR primers** | | |
| **Gene ID** | **Sequence (5'-3')** | |
| DLAT-1 | Fw: GCGTCTCTGCGCCTTTTTAG<br>Rv: TGGTTCCCGCAGGTTTACTC | N/A |
| DLAT-2 | Fw: GAGTAAACCTGCGGGAACCA<br>Rv: CCGCCATACCCTGTAGTCAC | N/A |
| MPC2-1 | Fw: TTATCGAGGAGCCGGTGGTA<br>Rv: CGCAGACTTGGTGAGGTGAT | N/A |
| MPC2-2 | Fw: ATCACCTCACCAAGTCTGCG<br>Rv: ACTGAAGGCCCAAAAACCCA | N/A |
| GPX2 | Fw: CATTATTCCAGACACTCCAA<br>Rv: TCAACGAGCTGCAATGCCGCT | N/A |
| G6PD | Fw: TGTGGTCCTTTTCCCCAAAG<br>Rv: GGAGGAGTAGAAACAGTCCA | N/A |
| SLC23A2 | Fw: GATCCCCGTTGAAGATGGAG<br>Rv: TCCCGGGTTCACGCCATTCT | N/A |
| GPNMB | Fw: AAGCACATGAGTTGTAAGAGG<br>Rv: CTGAATTCTCACGGACGC | N/A |
| HYAL1 | Fw: TAGGGAGTGGTGATGA<br>Rv: GAAAGATGTAGACGTGGGAG | N/A |
| **Antibodies/target** | | |
| NANOG | R&D systems | Cat# AF1997 |
| FLAG | Sigma-Aldrich | Cat# F1804 |
| OCT4 | Santa Cruz | Cat# sc-5279 |
| SOX2 | R&D systems | Cat# AF2018 |
| PRDM14 | Abcam | Cat# ab187881 |
| Caspase-3 | Cell Signaling Technology | Cat# 9662S |
| Cleaved Caspase-3 | Cell Signaling Technology | Cat# 9662S |
| GATA6 | R&D systems | Cat# AF1700 |
| GATA3 | Cell Signaling Technology | Cat# 5852 |
| DLAT | Santa Cruz | Cat# sc-271534 |
| ACSS2 | Proteintech | Cat# 16087-1-AP |
| ALDH2 | Proteintech | Cat# 15310-1-AP |
| CPT1A | Proteintech | Cat# 15184-1-AP |
| MPC2 | Proteintech | Cat# 20049-1-AP |
| GPX2 | Abclonal | Cat# A11171 |
| DNMT3L | Abclonal | Cat# A2342 |
| KLF17 | Abclonal | Cat# A13743 |
| REX1 | Thermo Fisher Scientific | Cat# PA5-27567 |
| CD75 | Invitrogen | Cat# 50-0759-42 |
| SUSD2 | BioLegend | Cat# 327406 |
| H2BK5ac | Cell Signaling Technology | Cat# 2574 |
| H3ac | Active motif | Cat# 61937 |
| H3K27ac | Active motif | Cat# 39034 |
| H4ac | Active motif | Cat# 39026 |
| H3K4me1 | Active motif | Cat# 61811 |
| H3K4me3 | Abclonal | Cat#A2357 |
| H3K27me3 | Millipore | Cat# 3018864 |
| H3K36me1 | Abclonal | Cat# A2364 |
| H3K36me2 | Abclonal | Cat# A2365 |
| H3K36me3 | Abclonal | Cat# A2366 |
| H3 | Cell Signaling Technology | Cat# 4620 |
| α-TUBULIN | Sigma | Cat# T9026 |
| β-ACTIN | Proteintech | Cat# 66009-1-Ig |
| **Chemicals, peptides, and recombinant proteins** | | |
| PD0325901 | StemCell Technologies | Cat# 72184 |
| IM12 | Selleck | Cat# S7566 |
| SB590885 | Selleck | Cat# S2220 |
| WH-4-023 | Selleck | Cat# S7565 |
| Y27632 | StemCell Technologies | Cat# 72037 |
| Y27632 | Selleck | Cat# S1049 |
| Human LIF | Millipore | Cat# LIF1050 |
| Human LIF | Peprotech | Cat# 300-05 |
| Activin A | Peprotech | Cat# 120-14E |
| bFGF | Peprotech | Cat# 100-18B |
| XAV939 | Sigma-Aldrich | Cat# X3004 |
| Gö6983 | Sigma-Aldrich | Cat# G1918 |
| Z-VAD | Selleck | Cat# S7023 |
| **Complete culture media and cell dissociation reagent** | | |
| mTeSR1 | StemCell Technologies | Cat# 85850 |
| N2 | Gibco | Cat# 17502048 |
| B27 | Gibco | Cat# 17504044 |
| DMEM/F12 | Gibco | Cat# 21331020 |
| Neurobasal | Gibco | Cat# 21103049 |
| Accutase | StemCell Technologies | Cat# 07920 |
| Dispase | StemCell Technologies | Cat# 07923 |
| 0.25% Trypsin-EDTA | Gibco | Cat# 25200056 |
| **Cell attachment proteins and peptides** | | |
| Matrigel | Corning | Cat# 354277 |
| Geltrex | Thermo Fisher Scientific | Cat# A1413301 |
| **Critical commercial kits** | | |
| Seahorse XF Cell Mito Stress Test Kit | Agilent | Cat# 103015-100 |
| Seahorse XF Glycolysis Stress Test Kit | Agilent | Cat# 103020-100 |
| CellROX™ Deep Red | Thermo Fisher Scientific | Cat# C10422 |
| Annexin V-FITC Apop Kit | Invitrogen | Cat# BMS500FI-300 |
| **Experimental models: cell lines** | | |
| SHhES8 | Zhu et al.(Zhu et al, 2017) | N/A |
| H9 | WiCell | WA09 |
| **Recombinant DNA** | | |
| pX335-AAVS1-gA/gB | Fang et al.(Fang et al, 2019) | N/A |
| pgRNA-CKB | Mandegar et al.(Mandegar et al, 2016) | Addgene # 73501 |
| pAAVS1-PDi-CRISPRn | Mandegar et al.(Mandegar et al, 2016) | Addgene # 73500 |
| pAAVS1-TRE3G-EGFP | Gift from Su-chun Zhang | N/A |
| pX335-ROSA26-L | This paper | N/A |
| pX335-ROSA26-R | This paper | N/A |
| **Software and algorithms** | | |

| Reagent/resource | Reference or source | Identifier or catalog number |
|---|---|---|
| FastQC | Babraham Bioinformatics | https://www.bioinformatics.babraham.ac.uk/projects/fastqc/ |
| Trim Galore | Babraham Bioinformatics | https://www.bioinformatics.babraham.ac.uk/projects/trim_galore/ |
| MarkDuplicates.jar | Hofmeister et al.(Hofmeister et al, 2020) | https://gatk.broadinstitute.org/hc/en-us/articles/360037052812-MarkDuplicates-Picard |
| XenofilteR | Kluin et al.(Kluin et al, 2018) | https://github.com/NKI-GCF/XenofilteR |
| Hisat2 v2.1.0 | Kim et al.(Kim et al, 2019) | https://daehwankimlab.github.io/hisat2/ |
| Htseq-count | Anders et al.(Anders et al, 2015) | https://htseq.readthedocs.io/en/master/htseqcount.html |
| StringTie | Pertea et al.(Pertea et al, 2015) | https://github.com/gpertea/stringtie |
| DESeq2 v1.26.0 | Love et al.(Love et al, 2014) | https://bioconductor.org/packages/release/bioc/html/DESeq2.html |
| pheatmap_1.0.12 | Raivo Kolde | https://cran.r-project.org/web/packages/pheatmap/pheatmap.pdf |
| Metascape | Zhou et al.(Zhou et al, 2019) | https://metascape.org |
| R | The R Foundation for Statistical Computing | https://www.r-project.org |
| Rstudio | Posit Software | https://www.rstudio.com/ |
| MACS | Zhang et al.(Zhang et al, 2008) | https://macs3-project.github.io/MACS/ |
| deepTools v3.3.0 | Ramirez et al.(Ramírez et al, 2014) | https://test-argparse-readoc.readthedocs.io/en/latest/index.html |
| HOMER | Heinz et al.(Heinz et al, 2010) | http://homer.ucsd.edu/homer/ |
| Samtools | Danecek et al.(Danecek et al, 2021) | https://github.com/samtools/samtools |

## Cell culture

Human embryonic stem cell (hESC) line SHhES8 (XX; derived by our lab) (Zhu et al, 2017) was used in all experiments except where it was indicated that hESC line H9 (XX; WA09, WiCell; RRID:CVCL_9773) (Thomson et al, 1998) was used. Cells were routinely tested ensuring to be mycoplasma-free. Normal karyotypes of the cell lines were

authenticated. Primed hESCs were cultured under the feeder-free condition in Matrigel (Corning) pre-coated plates in the mTeSR1 medium (STEMCELL Technologies), with 20% $O_2$ and 5% $CO_2$ at 37 °C. The medium was daily changed. Every 5–6 days, cells were dissociated into smaller clumps by incubating with Dispase (STEM-CELL Technologies) for 3 min at 37 °C. Naïve hPSCs were converted from their primed counterparts (SHhES8 and H9 hESCs) by the 5iLA medium (Theunissen et al, 2014) and cultured on radiation-inactivated mouse embryonic fibroblast (MEF, homemade) (Xu et al, 2009) feeders at either 20 or 5% $O_2$ and 5% $CO_2$, at 37 °C. For the most of experiments, naïve hESCs were cultured in the 5iLA medium, comprising a 1:1 mixture of DMEM/F12 and Neurobasal, 1% N2-supplement, 2% B27-supplement, 2 mM GlutaMAX, 0.1 mM β-mercaptoethanol, 1 × NEAA, 1 × penicillin-streptomycin, 1 µM PD0325901 (STEMCELL Technologies), 1 µM IM12 (Selleck), 0.5 µM SB590885 (Selleck), 1 µM WH-4-023 (Selleck), 10 µM Y27632 (STEMCELL Technologies), 20 ng/mL human LIF (PeproTech), and 20 ng/mL Activin A (PeproTech). For a few experiments, naïve hESCs were cultured in the PXGL medium (Guo et al, 2021), comprising a 1:1 mixture of DMEM/F12 and Neurobasal, 0.5% N2-supplement, 1% B27-supplement, 2 mM GlutaMAX, 0.1 mM β-mercaptoethanol, 1 × NEAA, 1 × penicillin-streptomycin, 1 µM PD0325901 (STEMCELL Technologies), 2 µM XAV939 (Sigma-Aldrich), 2 µM Gö6983 (Sigma-Aldrich), and 10 ng/mL human LIF (PeproTech). Naïve hESCs were passaged every 5–6 days by ratios of either 1:2 or 1:3 after single-cell dissociation with Accutase (STEMCELL Technologies).

## Constructs

RNA was isolated from naïve hESCs and reverse-transcribed into cDNA, which was used as the template to amply the full-length CDS of *GPX2* transcript variant 1 (NM_002083) and the selenocysteine insertion sequence (SECIS) element with the specific primers (GPX2-cDNA-F: 5'-GCGCTTCACCATGGCTTTCA-3'; GPX2-cDNA-R: 5'-CCACACCTGCCCTTT ATTGGT-3'). Then, the purified *GPX2* PCR product was inserted into the AAVS1-TRE3G-EGFP plasmid (a gift from Su-chun Zhang) backbone to generate AAVS1-TRE3G-GPX2 plasmid. To introduce the DOX-inducible *GPX2* overexpression cassette into chromosome 3 (ROSA26 locus), the TRE3G-*GPX2* fragment from the AAVS1-TRE3G-GPX2 plasmid was inserted into the pROSA construct containing the ROSA26 site- specific homologous recombination arms (homemade), generating the pROSA-TRE3G-GPX2 plasmid. Two gRNAs (gRNA- L and R) (Bertero et al, 2016) targeting the ROSA26 locus were cloned into the pX335 plasmid, respectively. The gRNA sequences are listed in the Reagents and tools table. DNA sequences of all constructs were validated by DNA sequencing.

The construction for DOX- inducible *NANOG-mut* overexpression was accomplished by the piggyBAC transposon system (Ding et al, 2005). The full-length coding sequence of *NANOG* was amplified by PCR from the pROSA-Bsd-TRE3G-NANOG vector using primers containing synonymous mutations to evade gRNA targeting. The primers used for amplification included NANOG-gRNA-mut-F (5'-CACTTCCTACCCTCGTAAAG-3') and NANOG-mut-Sac I-R (5'- ATGGGGCATTTCCGCGCTAGACATTTG-CAAGGATGGATAG-3') for the short product, as well as NANOG-mut-Mlu I-F (5'- AGCGCGGAAATGCCCCATACGGA-GACTGTCTCTCCTCTTC-3') and NANOG-mut-Mlu I-R

(5'- ACAGT TACATTGGATCCCTGC -3') for the long product. Subsequently, the short and long products were ligated through homologous recombination in accordance with the manufacturer's protocol (TIANGEN VI201). The purified full-length NANOG product was then inserted into the pUC19-pB-T3G-3×FLAG-EGFP plasmid backbone (a homemade construct), which had been digested with Mlu I and Sac I, resulting in the generation of the pUC19-pB-T3G-3×FLAG-NANOG-mut plasmid.

## Generation of DOX- inducible NANOG knockout and GPX2 overexpression (iKOE) hESC lines

DOX-inducible NANOG knockout (iKO) hESC lines were established by the published CRISPRn technology (Mandegar et al, 2016). Initially, to establish host hESC lines with DOX- inducible spCas9 expression under the control of the TetO (TRE3G) promoter, singularized hESCs were electroporated with the pAAVS1-PDi-CRISPRn plasmid (a gift from Mohammad A. Mandegar), together with the pX335-AAVS1 plasmids containing two gRNAs (Fang et al, 2019). Subsequently, constructs for constitutively expressing gRNAs targeting NANOG or non-targeting (NT) sequences (Mandegar et al, 2016) were electroporated into the host hESC lines. After selection with 10 μg/mL of Blasticidine for a week, single colonies were picked up and identified, resulting in the generation of DOX- inducible NANOG iKO or NT hESC lines. The NANOG depletion induced by DOX (0.1 μg/mL) was validated by RT-qPCR, western blotting, and immunofluorescent staining. The gRNA sequences are listed in the Reagents and tools table.

To establish DOX- inducible GPX2 overexpression (iOE), or NANOG knockout together with GPX2 overexpression (iKOE) hESC lines, singularized NT hESCs and NANOG iKO primed hESCs, respectively, were electroporated with the pROSA-TRE3G-GPX2 plasmid, together with the plasmids of pX335-ROSA26-L and pX335-ROSA26-R. After selection with 200 μg/mL of G418 for a week, single colonies were picked up and identified by western blotting and genomic PCR. DOX at 0.1 μg/mL was used, unless otherwise indicated.

## Generation of a DOX- inducible NANOG knockout and NANOG-mut overexpression (iNANOG-mut_ NANOG iKO) hESC line

To generate the iNANOG-mut_ NANOG iKO hESC line, where simultaneous expression of exogenous NANOG-mut and deletion of endogenous NANOG could be induced by DOX, singularized primed NANOG iKO hESCs were electroporated with the pUC19-pB-T3G-3×FLAG-NANOG-mut plasmid, together with the Act-pBase plasmid (Li et al, 2011). After being selected with 100 μg/mL of G418 for 2 weeks, the successful insertion of an inducible NANOG-mut overexpression cassette into the genome of our established primed NANOG iKO hESCs through the piggyBAC transposon system was verified by western blotting and genomic PCR. DOX at 0.1 μg/mL was used, unless otherwise indicated.

## Generation of the DOX- inducible NANOG knockdown (NANOG iKD) hESC line

The DOX- inducible CRISPRi technology was employed to knockdown NANOG expression (Mandegar et al, 2016). To establish the host hESC line with DOX- inducible dCas9 expression driven by the TetO (TRE3G) promoter, singularized primed hESCs were electroporated with the pAAVS1-NDi-CRISPRi plasmid (a gift from Mohammad A. Mandegar), together with the pX335-AAVS1 plasmid containing two gRNAs (Fang et al, 2019). Subsequently, constructs for constitutively expression of gRNAs targeting NANOG were electroporated into the host hESCs (Mandegar et al, 2016), resulting in the generation of DOX-inducible CRISPRi_NANOG hESC lines, termed as NANOG iKD hESCs. After being selected with 1 μg/mL of puromycin for 2–3 days, single colonies were picked up and identified by western blotting and genomic PCR.

## Western blot analysis

Cell lysates were collected using a protein lysis buffer (50 mM Tris-HCl, pH 7.4, 2% SDS, 5% β-mercaptoethanol, 10% glycerol, 0.01% bromophenol blue) and denatured at 99 °C for 5 min. Protein samples were separated on the 12% SDS-PAGE and then transferred to nitrocellulose (NC) membranes. The membranes were blocked with 3% BSA (Dobi) in Tris-HCl buffered saline containing 0.1% Tween-20 (TBST) at room temperature for 2 h, and then incubated overnight with primary antibodies in 3% BSA at 4 °C. The membranes were treated with horseradish peroxidase (HRP)-conjugated secondary antibodies for 2 h on the following day. After three washes with the TBST, the blotted membranes were exposed to Immobilon Western Chemiluminescent HRP Substrate (Millipore) and analyzed by a Tanon 5200 instrument. The antibodies used for western blotting are listed in the Reagents and tools table. The recommended concentrations were used for all antibodies, and a minimum of three independent experiments were carried out for all western blot analyses.

To quantify relative levels of protein bands, the intensity of bands was measured by the ImageJ, followed by calculating the ratios of the intensity between target protein bands and loading control bands. The ratio of DOX- untreated samples was set as 1, unless otherwise indicated.

## Reverse transcription and real-time quantitative polymerase chain reaction (RT-qPCR)

Cells were lysed with the TRIzol reagent (Life Technologies), and RNA was isolated according to standard RNA isolation procedures. For cDNA synthesis, 2 μg RNA was used with the RT-Ace (TOYOBO). RT-qPCR was performed using the qTOWER384 (AnalytikJena) with TB Green™ Premix Ex Taq™ II (Takara), following the manufacturer's instructions. ACTB served as the internal control. A list of primers used for RT-qPCR is provided in the Reagents and tools table.

## Immunofluorescence staining

Immunofluorescence staining was conducted as described (Ma et al, 2012). Antibodies used are as follows: NANOG (R&D Systems, AF1997), NANOG (Proteintech, 14295-1-AP), GATA6 (R&D Systems, AF1700), GATA3 (Cell Signaling Technology, 5852), DNMT3L (Abclonal, A2342), KLF17 (Abclonal, A13743), REX1 (Thermo Fisher Scientific, PA5-27567), and SSEA4 (R&D Systems, MAB1435). The recommended concentrations were used for all antibodies.

## Oxygen consumption rate (OCR) analysis

MEFs were seeded at a density of 15,000 cells/well in the XF24 Cell Culture Microplate (Agilent) two days prior to seeding naïve hESCs, which were seeded at a density of 30,000 cells/well in the 5iLA medium. After 3 days of seeding, hESCs were cultured for an additional 3 days with or without DOX. One hour prior to measurement, the culture medium was switched to the XF base medium, supplemented with 10 mM D-glucose, 1 mM sodium pyruvate, and 2 mM L-glutamine, all from Agilent. The OCR was assessed using Seahorse XF24 Extracellular Flux Analyzers (Agilent). Inhibitors, including Oligomycin (1.5 mM), FCCP (1 mM) and Rotenone/Antimycin A (0.5 mM), were sequentially administered. The assay was conducted according to the manufacturer's instructions, and OCR values were normalized against total protein amounts quantified using a BCA protein assay kit (Thermo Scientific).

## Extracellular acidification rate (ECAR) analysis

MEFs and naïve hESCs were seeded and cultured as in the OCR analysis. One hour prior to the measurement, the culture medium was switched to the XF base medium supplemented with 2 mM L-glutamine. The ECAR was assessed using Seahorse XF24 Extracellular Flux Analyzers. Substrates and inhibitors, including 10 mM D-glucose, 1 μM oligomycin, and 50 mM 2-deoxy-D-glucose, were sequentially administered. The assay was conducted according to the manufacturer's instructions, and ECAR values were normalized against total protein levels quantified using the BCA protein assay kit.

## Flow cytometry analyses

For CD75/SUSD2 staining, naïve hESCs were singularized by Accutase, collected into 15 mL tubes, and centrifuged at 100×g for 3 min. The supernatant was discarded, and cell pellets were resuspended in 100 μL of the FACS buffer (DPBS with 5% BSA) containing CD75 antibody (Invitrogen, 50-0759-42) and SUSD2 antibody (BioLegend, 327406) at a dilution of 1:100. A negative control was prepared using the FACS buffer alone for suspension. All samples were incubated on ice for 30 min in the dark to stain. After staining, samples were centrifuged at 100×g for 3 min. The supernatant was discarded, and cell pellets were resuspended in the cold FACS buffer. This process was repeated once before analyzing the fluorescence signals using a CytoFLEX LX Flow Cytometer (Beckman). The analysis was performed according to the manufacturer's instructions.

Annexin V/propidium iodide (PI) (Annexin V/PI) staining was conducted using the Annexin V-FITC Apop Kit (Invitrogen, BMS500FI-300). Cells from one well of a six-well plate, including cells in the culture medium and adherent growing cells, were collected, washed once with 1 mL 1×PBS, and resuspended in 100 μL of the 1 × binding buffer. These samples were then incubated with 4 μL of Annexin V for 15 min in the dark, followed by adding 100 μL of 1 × binding buffer and 4 μL of PI. Samples were then analyzed using the CytoFLEX LX Flow Cytometer (Beckman).

For the measurement of intracellular ROS, naïve hESCs, without or with DOX treatment, were dissociated using Accutase, and centrifuged at 130×g for 3 min. The supernatant was discarded, and cell pellets were resuspended in 500 μL of the 5iLA culture medium containing 1 μM CellROX™ Deep Red reagent (Invitrogen) and incubated for 60 min at 37 °C, followed by PBS wash once. Cell samples resuspended in 100 μL PBS were analyzed by flow cytometry following the manufacturer's instructions.

## Metabolic measurements

Naïve hESCs were seeded in 10-cm plates, and each group contained four plates. Three days later, cells were treated without or with DOX for an additional 2 or 3 days. All samples were mechanically collected with 1 mL of dry ice-cold solution of 80% methanol: 20% water, and sonicated to break up the cells to release metabolites, and then centrifuged at 13,500×g for 10 min. The whole process was kept at a low temperature. Supernatant was collected and further evaporated. Samples were diluted with 100 μL of mobile phase A (25% ACN, 5 mM ammonium acetate) and then centrifuged at 20,000×g for 10 min to remove insoluble particles for LC-MS detection of metabolites. Metabolites were analyzed by the LC-MS on an Ultimate 3000 UHPLC liquid chromatography system (Thermo Scientific) equipped with a C18 column (Phenomenex, 00B-4252-B0) and a TSQ Quantiva mass spectrometer (Thermo Scientific). The solvent gradient was the same as in a previous study (Moon et al, 2019). The retention time and the m/z of metabolites were determined by standards as previously reported (Henneman et al, 2011).

For $NADP^+$ and NADPH quantification, naïve hESCs, either without or with DOX treatment for 3 days, were mechanically collected with 1 mL of methanol: acetonitrile: water (2:2:1, v/v) solution. Samples were analyzed by the Applied Protein Technology (Shanghai, China). The method was the same as previously reported (Liu et al, 2022). Briefly, following the SRM/MRM method, metabolite analysis of $NADP^+$ and NADPH was performed on a system consisting of a UHPLC (1290 Infinity LC, Agilent) coupled to a QTRAP mass spectrometer (AB SCIEX 5500). Quantitative information from every sample set was extracted in the Multiquant Software.

## Reference genome

The human reference genome GRCh37 from Ensembl was utilized for sequence alignment. The gene annotation file GRCh37 from Ensembl was applied to all genomic analyses.

## RNA-seq and data analysis

Total RNA was isolated from cells using the TRIzol Reagent (Life Technology) following the manufacturer's protocol. The quantity and integrity of the RNA were assessed using the K5500 (Beijing Kaiao, China) and the Agilent 2200 TapeStation (Agilent Technologies, USA), respectively. Briefly, mRNA was enriched using oligo (dT) according to the instructions of the NEBNext® Poly(A) mRNA Magnetic Isolation Module (NEB, USA), and then fragmented to approximately 200 bp in size. Subsequently, RNA fragments were subjected to first and second strand cDNA synthesis, followed by adapter ligation and enrichment using a low-cycle according to the instructions of the NEBNext® Ultra™ RNA Library Prep Kit for Illumina. The purified library products were evaluated using the Agilent 2200 TapeStation and Qubit (Thermo Fisher Scientific, USA). Sequencing was performed on

libraries using the Illumina (Illumina, USA) with 150 bp paired-end reads at Ribobio Co. Ltd. (Ribobio, China).

The quality control for raw reads was performed using FastQC v0.11.8. Alignment to the human reference genome GRCh37(--trim5 30 --trim3 10) was performed using Hisat2 v2.1.0 (Kim et al, 2019). PCR duplicates were removed using the Java package MarkDuplicates.jar (Hofmeister et al, 2020), and MEF contamination was filtered out with the R-package XenofilteR (Kluin et al, 2018). Gene quantification was performed using both htseq-count (Anders et al, 2015) and StringTie (Pertea et al, 2015). Differentially expressed genes (DEGs) were identified using R package DESeq2 v1.26.0 (Love et al, 2014), with an adjusted $p$ value threshold of $< 0.05$. The GO term enrichment for DEGs was conducted using the Metascape (Zhou et al, 2019) (https://metascape.org). Downstream data visualization, including PCA plots, was generated using the prcomp function in R, and gene expression patterns of selected genes were visualized in heatmaps using the R package pheatmap_1.0.12 (https://cran.r-project.org/web/packages/pheatmap/pheatmap.pdf). Gene expression levels were normalized using the Z-score of RPKM (Reads Per Kilobase per Million mapped reads).

## CUT&Tag assays and data analyses

The CUT&Tag assays were performed according to the manufacturer's instructions provided by the Hyperactive Universal CUT&Tag Assay Kit (Vazyme, Nanjing, China). Each group consisting of $2 \times 10^5$ naïve SHhES8 cells was washed once with 500 μL wash buffer, followed by a 10-min incubation with ConA beads to facilitate cell collection. Cells were then incubated overnight at 4 °C with 1 μg each of the following antibodies: H3K27ac (active motif, 39133), H3K27me3 (Millipore, 3018864), H3K4me1 (active motif, 61633), and H3K4me3 (Abclonal, A2357). On the following day, the corresponding secondary antibody was added and incubated for 1 h at room temperature, followed by three washes with DIG-wash buffer. Cells were then incubated with 0.04 μM pA/G-Tnp for 1 h at room temperature, washed three times with DIG 300 buffer, and then treated with $5 \times$ TTBL to fragment the DNA, which was incubated at 37 °C for 1 h. Following fragmentation, DNA was extracted using proteinase K, buffer LB and DNA extract beads at 55 °C for 10 min. Library construction was completed using the TD202 TruePrep Index Kit V2 for Illumina (Vazyme Biotech).

The data quality control for CUT&Tag assays was performed using FastQC v0.11.8. Alignment was performed using the Hisat2 v2.1.0 (Kim et al, 2019) onto the human reference genome GRCh37(--trim5 30 --trim3 10). PCR duplicates were removed using the Java package MarkDuplicates.jar (Hofmeister et al, 2020), and MEF contamination was filtered using the R-package XenofilteR (Kluin et al, 2018). CUT&Tag peaks were defined using the MACS (Model-based Analysis for ChIP-Seq) (Zhang et al, 2008) with default parameters. Bigwig tracks were generated using deepTools v3.3.0 (Ramírez et al, 2014), normalized to FPKM with a binsize of 10 bp. CUT&Tag signals over genomic regions were plotted using deepTools v3.3.0 (Ramírez et al, 2014).

## ATAC-seq and data analysis

ATAC-seq was conducted using the MagicSeq Tn5 DNA Library Prep Kit for Illumina (Magic-Bio, M3141). Cell pellets were treated with transposase at 37 °C for 30 min, purified using the MinElute PCR Purification Kit (QIAGEN, 28006), and amplified using the 1

$\times$ NEBnext PCR Master Mix (NEB, M0541S) with custom Nextera PCR primers 1 and 2. Following purification with the MinElute PCR Purification Kit (QIAGEN, 28006), libraries were sequenced using the NovaSeq with 150 bp pair-end reads.

The quality control for ATAC-seq data was conducted using FastQC v0.11.8. Adapters were trimmed using Trim_Galore, and raw FASTQ files were processed accordingly (https://www.bioinformatics.babraham.ac.uk/projects/trim_galore/). Alignment was performed using the Hisat2 v2.1.0 (Kim et al, 2019) onto the human reference genome GRCh37, with specified trimming parameters (--trim5 30 --trim3 10). PCR duplicates were removed using the Java package MarkDuplicates.jar (Hofmeister et al, 2020), and MEF contamination was filtered out using the R-package XenofilteR (Kluin et al, 2018). ATAC-seq peaks were identified using the MACS (Zhang et al, 2008) with default parameters. Bigwig tracks were generated with deepTools v3.3.0 (Ramírez et al, 2014), normalized to RPKM with a binsize of 10 bp. Genomic region signals from ATAC-seq signals were visualized using deepTools v3.3.0 (Ramírez et al, 2014).

## ChIP-qPCR and ChIP-seq data analyses

Cells were harvested, fixed with 1% formaldehyde (Thermo) for 10 min at room temperature with rotation, and quenched with 0.2 M glycine at room temperature for 10 min. Cells were lysed using the ChIP lysis buffer containing 10 mM Tris-HCl (pH 8.0), 0.25% Triton X-100, 10 mM EDTA, 100 mM NaCl, and a protease inhibitor cocktail. Genomic DNA was sonicated to obtain short fragments averaging 500 bp in size for ChIP-qPCR or 250 bp in size for ChIP-seq. Fragmented DNA was incubated with NANOG antibody (R&D Systems, AF1997) overnight at 4 °C. The mixture was then incubated with 30 μL protein G Dynabeads for 2 h at room temperature with rotation. DNA was eluted in the elution buffer (50 mM Tris-HCl, pH 8.0, 1 mM EDTA, 1% SDS), and treated with proteinase K overnight at 50 °C. DNA purification was conducted using a QIAquick PCR Purification Kit as instructed. Subsequently, DNA (0.5 ng) was used for each ChIP-qPCR amplification. Results are presented as relative enrichments compared to the input. All primers used in ChIP-qPCR assays are listed in the Reagents and tools table.

The ChIP-seq data quality control was conducted using FastQC v0.11.8. Adapters were trimmed using Trim_Galore. Raw FASTQ files were processed with the Trim Galore (https://www.bioinformatics.babraham.ac.uk/projects/trim_galore/). The HISAT2 v2.1.0 (Kim et al, 2019) was used for alignment onto the human reference genome GRCh37 with specified trimming parameters (--trim5 30 --trim3 10). PCR duplicates were removed by the Java package MarkDuplicates.jar (Hofmeister et al, 2020), and MEF contamination was filtered out using the R-package XenofilteR (Kluin et al, 2018). ChIP-seq peaks were identified using the MACS (Zhang et al, 2008) with default parameters, and signals were visualized using deepTools v3.3.0 (Ramírez et al, 2014), normalized to RPKM with a binsize of 10 bp.

## Motif enrichment analysis of transcription factors

The motif enrichment analysis of transcription factors was performed using the HOMER (Hypergeometric Optimization of Motif EnRichment) (Heinz et al, 2010) (http://homer.salk.edu/

homer/chipseq), and peaks were annotated with the annotate-Peaks.pl function in the Homer.

## Statistical analysis

The unpaired two-tailed Student's *t*-test was used to estimate statistical significance for differences between two groups for data satisfying with the assumptions of normal distribution. Otherwise, the Wilcoxon signed-rank test was used.

## Graphics

The synopsis image and the image in Fig. 7H were created with BioRender.com (Agreement number: WX28XJ9O7Z).

# Data availability

The datasets produced in this study are available in the following databases: All sequencing data have been deposited in the Sequence Read Archive (SRA) of NCBI: PRJNA1258868, PRJNA1258370, and PRJNA1255744. All data of metabolites has been deposited in the MetaboLights: MTBLS12577 and MTBLS12578.

The source data of this paper are collected in the following database record: biostudies:S-SCDT-10_1038-S44319-025-00629-9.

# Peer review information

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

## Acknowledgements

We thank Dr. Junke Zheng for help with the OCR and ECAR assays, Dr. Jing Zhang for help with the ROS measurements, Dr. Xuemei Tong for helpful discussions about glucose metabolism and NADPH generation, and ErBo Xu for English editing of the manuscript. We thank the Core Facility of Basic Medical Sciences, Shanghai Jiao Tong University School of Medicine, for the technical support. The study was supported by grants from the National Natural Science Foundation of China (32170796 and 32370847), the Ministry of Science and Technology of the People's Republic of China (2021YFA1100400 and 2016YFA0100100), and the Innovative Research Team of High-level Local Universities in Shanghai (SHSMU-ZLCX20210201).

## Author contributions

**Min Shao**: Conceptualization; Data curation; Software; Formal analysis; Investigation; Visualization; Writing—review and editing. **Han Wang**: Conceptualization; Data curation; Formal analysis; Investigation; Visualization; Methodology; Writing—review and editing. **Yujie Liu**: Validation; Investigation; Methodology. **Yongqiang Wang**: Formal analysis; Investigation; Methodology. **Hanzhi Zhao**: Investigation; Methodology. **Junjie Gu**: Resources; Methodology. **Ning Zhong**: Validation; Investigation. **Yifan Zhou**: Resources; Methodology. **Huiyong Yin**: Resources; Supervision; Methodology. **Ying Jin**: Conceptualization; Resources; Supervision; Funding acquisition; Writing—review and editing. **Bing Liao**: Conceptualization; Resources; Supervision; Funding acquisition; Writing—original draft; Project administration; Writing—review and editing.

Source data underlying figure panels in this paper may have individual authorship assigned. Where available, figure panel/source data authorship is listed in the following database record: biostudies:S-SCDT-10_1038-S44319-025-00629-9.

## Disclosure and competing interests statement

The authors declare no competing interests.

# Expanded View Figures

**Figure EV1.** *NANOG* **deficiency impairs self-renewal and transcriptional programs of hESCs.**

(A–C) Characterization of naïve hESCs by the flow cytometric analysis of the percentage of CD75 and SUSD2 double-positive cells (A), RT-qPCR analyses of relative mRNA levels of naïve (top) and primed (bottom) pluripotency markers (B), and representative immunofluorescence staining images for NANOG as well as naïve (DNMT3L, KLF17, and REX1) and primed (SSEA4) pluripotency markers in SHhES8 hESCs (C). Scale bar, 25 μm. Data are presented as mean ± SEM ($n = 3$) (A, B). $p = 1.09 \times 10^{-8}$ (A). *KLF5*, $p = 1.56 \times 10^{-5}$; *DNMT3L*, $p = 5.76 \times 10^{-5}$; *KHDC3L*, $p = 1.33 \times 10^{-5}$ (B). (D) The representative western blot analysis result showing protein levels of NANOG and cleaved Caspase-3 in naïve (top) and primed (bottom) NT or *NANOG* iKO SHhES8 hESCs, either untreated or treated with DOX for the indicated time lengths. Alpha-TUBULIN served as a loading control. (E) Representative phase contrast images displaying cell colonies of primed NT (left) and *NANOG* iKO (right) SHhES8 hESCs, either untreated or treated with DOX for two or three days. (F) The line chart depicting live cell counts of primed NT (top) or *NANOG* iKO (bottom) SHhES8 hESCs, either untreated or treated with DOX over the indicated days. Data are presented as mean ± SEM ($n = 3$). (G, H) Volcano plots depicting upregulated or downregulated DEGs in naïve (G) and primed (H) *NANOG* iKO SHhES8 hESCs induced by *NANOG* depletion. Upregulated genes (red dots) were defined by an adjusted *p* value (*p*adj) <0.05 and log2 (FoldChange) >0; Downregulated genes (blue dots) are shown with *p*adj <0.05 and log2 (FoldChange) <0. Comparisons were made between DOX-treated cells and their untreated counterparts. NS, not significantly changed. DESeq2 was used for the statistical analysis of DEGs. Three biological replicates were used for each condition ($n = 3$). (I) Heatmaps displaying normalized RNA-seq levels of selected markers for three germ layers and primed pluripotency state in primed *NANOG* iKO SHhES8 hESCs, either untreated or treated with DOX for two or three days. The color represents Z-scores. Three biological replicates were used for each condition ($n = 3$). The unpaired two-tailed Student's *t*-test was used for the statistical analysis in (A, B, F).

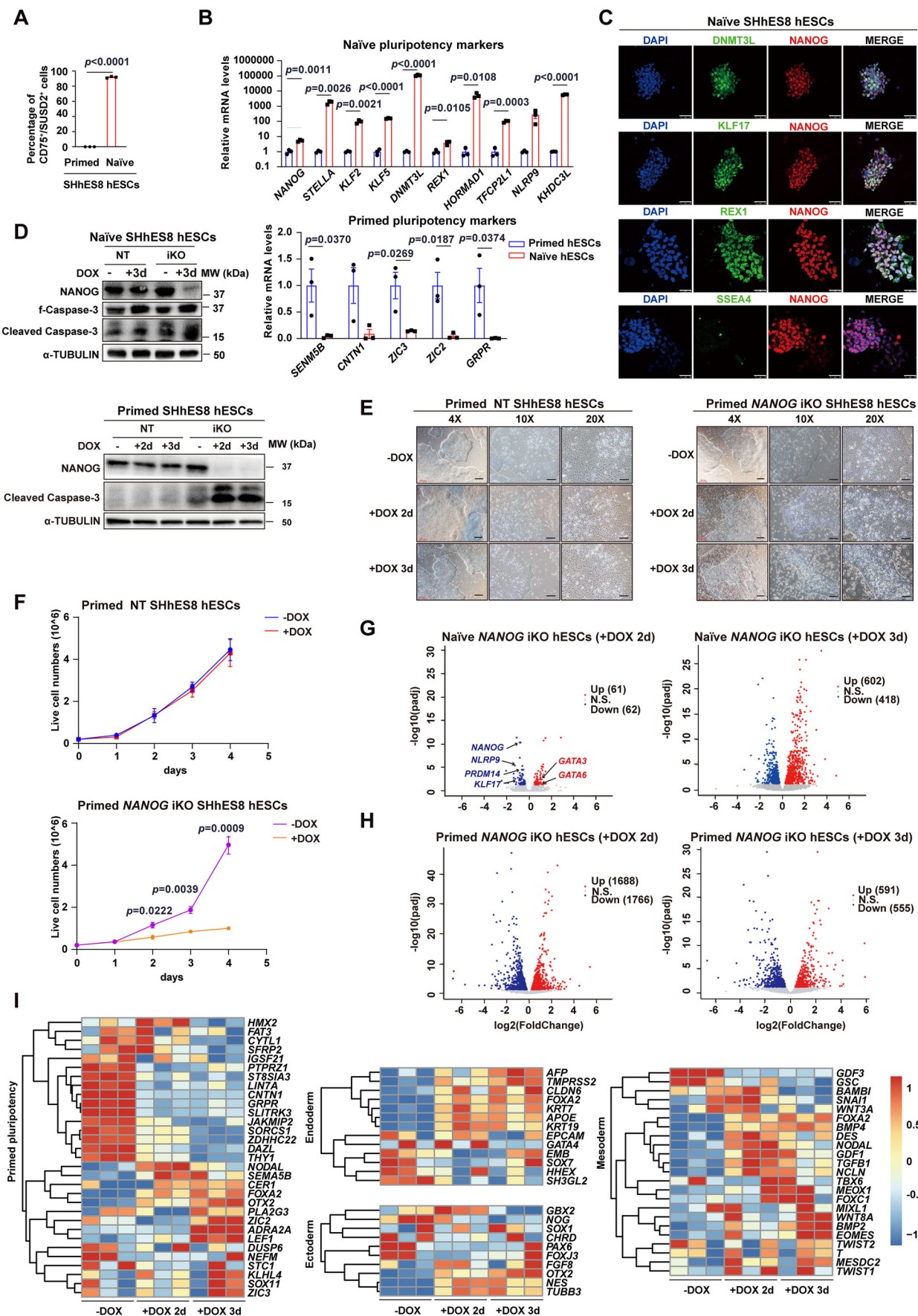

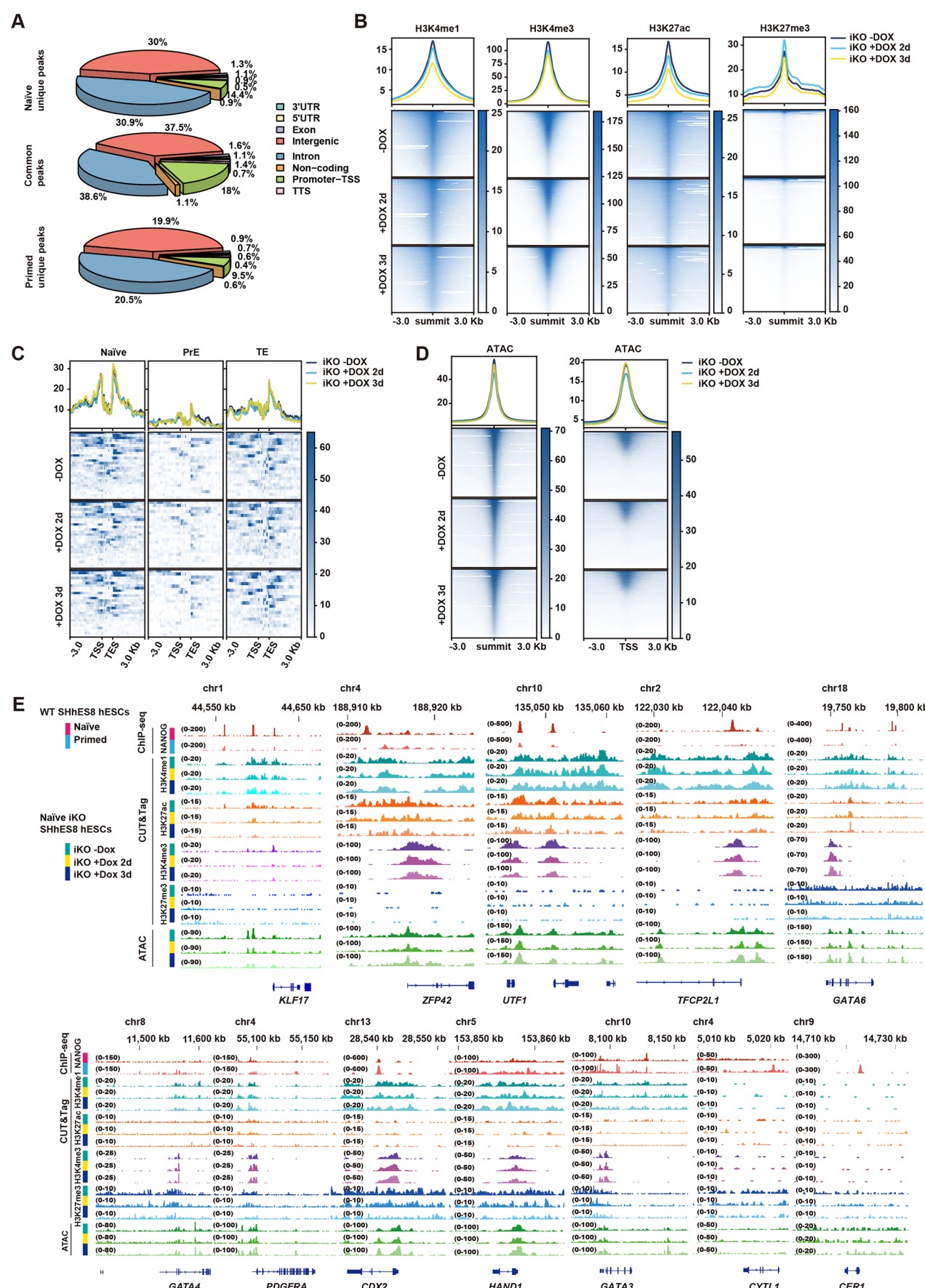

◀ **Figure EV2. *NANOG* depletion results in distinct epigenetic changes for marker genes of different lineages.**

(A) Pie charts displaying the genomic distribution of the three groups of NANOG binding peaks. (B) Heatmaps depicting changes in the abundance (normalized by total reads) of histone modifications (H3K27ac, H3K27me3, H3K4me1, and H3K4me3) across the whole genome in naïve *NANOG* iKO SHhES8 hESCs, either untreated or treated with DOX for two or three days, as analyzed by the deepTools. (C, D) Heatmaps illustrating the abundance (normalized by total reads) of ATAC-seq signals within a 3 kb range upstream and downstream of marker genes associated with naïve pluripotency, TE, or PrE (C), and of ATAC-seq signals across the whole genome (left) or TSS regions (right) (D) in naïve *NANOG* iKO SHhES8 hESCs, either untreated or treated with DOX for three days, as analyzed by the deepTools. (E) Genome browser snapshots of NANOG occupancy at the vicinity of naïve pluripotency markers (*KLF17, ZFP42, UTF1*, and *TFCP2L1*), PrE markers (*GATA6, GATA4*, and *PDGFRA*), TE markers (*GATA3, CDX2*, and *HAND1*), and primed pluripotency markers (*CYTL1* and *CER1*) in primed and naïve SHhES8 hESCs, as well as histone modifications and chromatin openness in naïve *NANOG* iKO SHhES8 hESCs, either untreated or treated with DOX for two or three days.

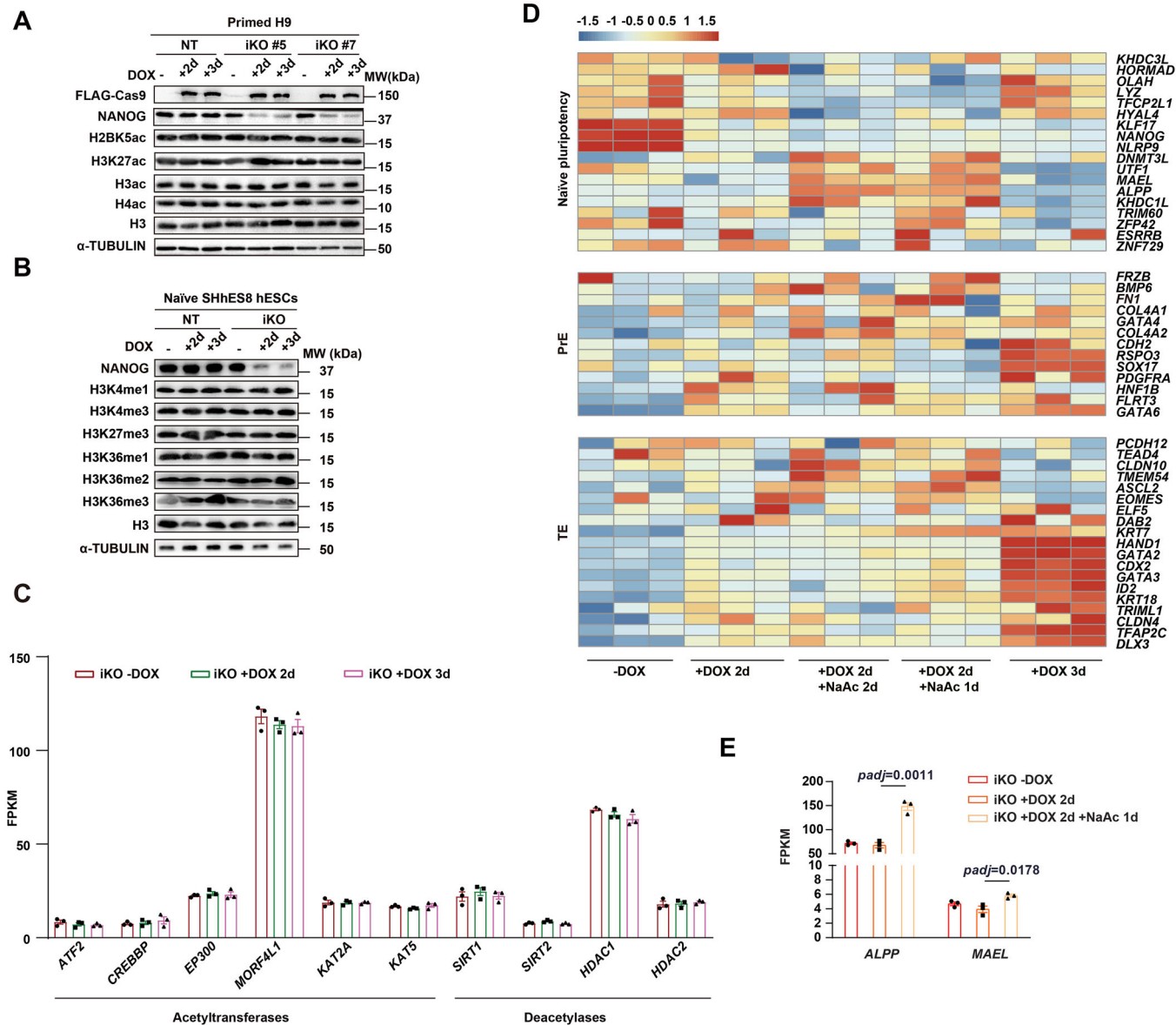

**Figure EV3. NANOG maintains histone acetylation modifications via controlling acetyl-CoA production in naïve hESCs.**

(A) The representative western blot analysis result for levels of NANOG proteins and histone acetylation modifications in primed NT and *NANOG* iKO H9 hESCs, either untreated or treated with DOX for two or three days. Alpha-TUBULIN and histone 3 served as loading controls. Single clonal culture of *NANOG* iKO #5 and #7 was used. (B) The representative western blot analysis result for levels of NANOG and histone methylation modifications in naïve NT and *NANOG* iKO SHhES8 hESCs, either untreated or treated with DOX for two or three days. Alpha-TUBULIN and histone 3 served as loading controls. (C) The FPKM values from our RNA sequencing data for a set of genes related to histone acetylation modifications in naïve *NANOG* iKO hESCs, either untreated or treated with DOX for two or three days. Data are presented as mean ± SEM (*n* = 3). The unpaired two-tailed student's *t*-test was used for statistical analysis. (D) Heatmaps showing normalized mRNA levels, measured by our RNA-seq assay, of selected naïve pluripotency, PrE and TE marker genes in naïve *NANOG* iKO SHhES8 hESCs cultured under indicated conditions. The color represents Z-scores. Three biological replicates were used for each culture condition. (E) The FPKM values from our RNA sequencing data for two naïve pluripotency marker genes, *ALPP* and *MAEL*, either untreated or treated with DOX or DOX and NaAc in naïve *NANOG* iKO hESCs. Data are presented as mean ± SEM (*n* = 3). DESeq2 was used for the statistical analysis.

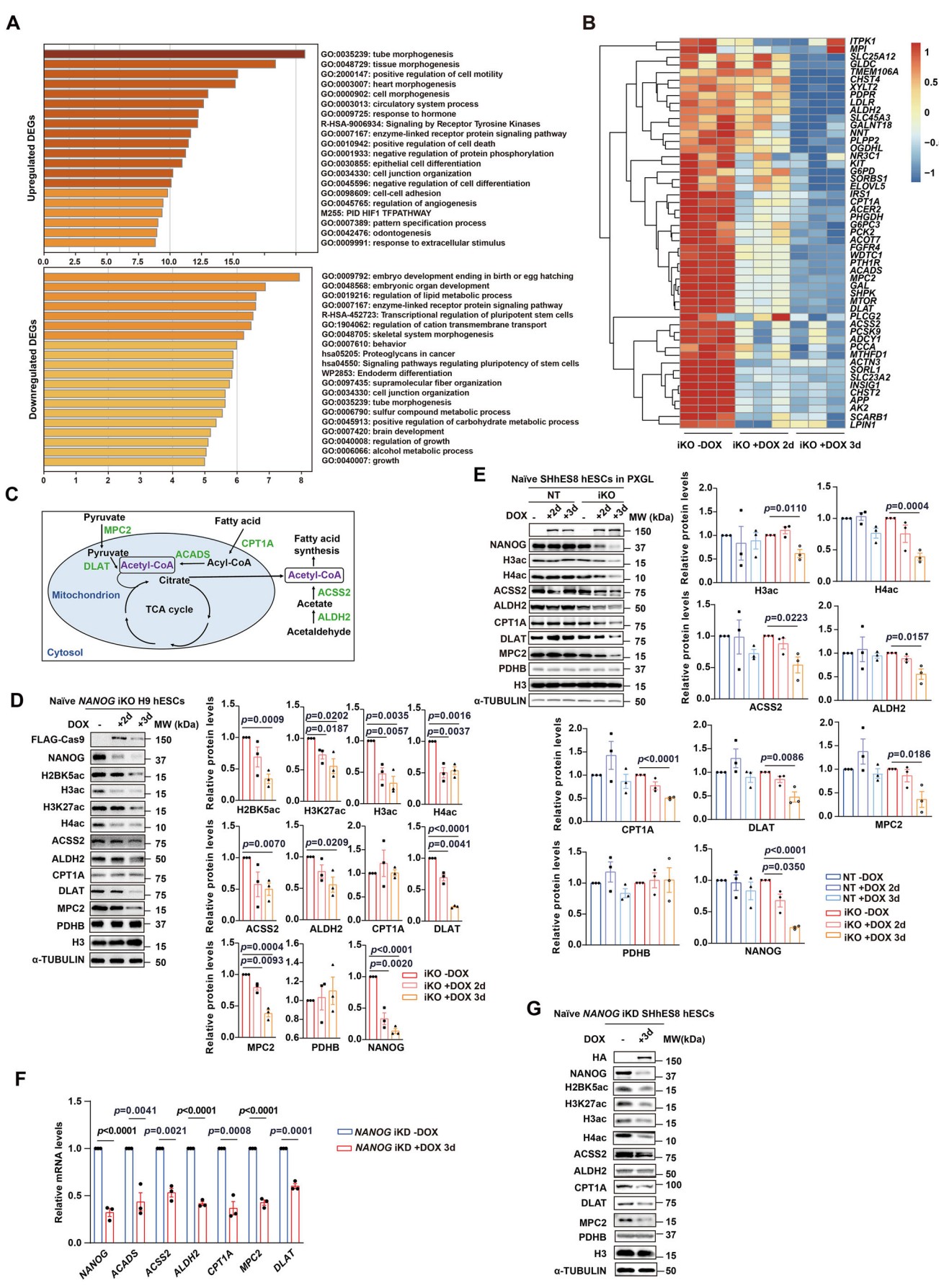

**Figure EV4. NANOG activates a set of genes involved in acetyl-CoA synthesis in naïve hESCs.**

(A) The GO analysis of 473 NANOG- occupied and *NANOG* depletion- upregulated (top) as well as 362 NANOG-occupied and *NANOG* depletion- downregulated (bottom) DEGs in naïve hESCs. The top 20 enriched terms are presented. (B) The heatmap depicting normalized mRNA levels of metabolism- related genes in enriched pathways shown in Fig. 4B, measured by our RNA-seq assays. The color represents Z-scores. Three biological replicates were used for each condition ($n = 3$).
(C) A schematic diagram of metabolic pathways of acetyl-CoA synthesis and putative NANOG targets associated with the pathway. (D) The representative western blot analysis result for protein levels of NANOG, histone acetylation modifications, and proteins related to acetyl-CoA synthesis in naïve H9 *NANOG* iKO hESCs cultured under the 5iLA condition, either untreated or treated with DOX for two or three days. Alpha-TUBULIN and histone 3 served as loading controls (left). The quantitative analysis results of relative protein levels are shown on the right side. Data are presented as mean ± SEM ($n = 3$). DLAT, Row 1, $p = 2.93 \times 10^{-7}$; NANOG, Row 1, $p = 2.60 \times 10^{-5}$.
(E) The representative western blot analysis result for levels of proteins indicated in naïve NT and *NANOG* iKO SHhES8 hESCs cultured under the PXGL condition, either untreated or treated with DOX for two or three days. Alpha-TUBULIN and histone 3 served as loading controls (left). The quantitative analysis results of relative protein levels are shown on the right side. Data are presented as mean ± SEM ($n = 3$). CPT1A, $p = 1.08 \times 10^{-5}$; NANOG, Row 1, $p = 1.06 \times 10^{-6}$. (F) RT-qPCR analysis results for relative mRNA levels of the indicated genes in naïve *NANOG* iKD SHhES8 hESCs, either untreated or treated with DOX for 3 days. Data are presented as mean ± SEM ($n = 3$). *NANOG*, $p = 9.39 \times 10^{-5}$; *ALDH2*, $p = 9.35 \times 10^{-6}$; *MPC2*, $p = 3.62 \times 10^{-5}$. (G) The representative western blot analysis result for levels of indicated proteins in naïve *NANOG* iKD SHhES8 hESCs, either untreated or treated with DOX for 3 days. dCas9 was tagged by HA. Alpha-TUBULIN and histone 3 served as loading controls. The unpaired two-tailed student's *t*-test was used for the statistical analysis in (D–F).

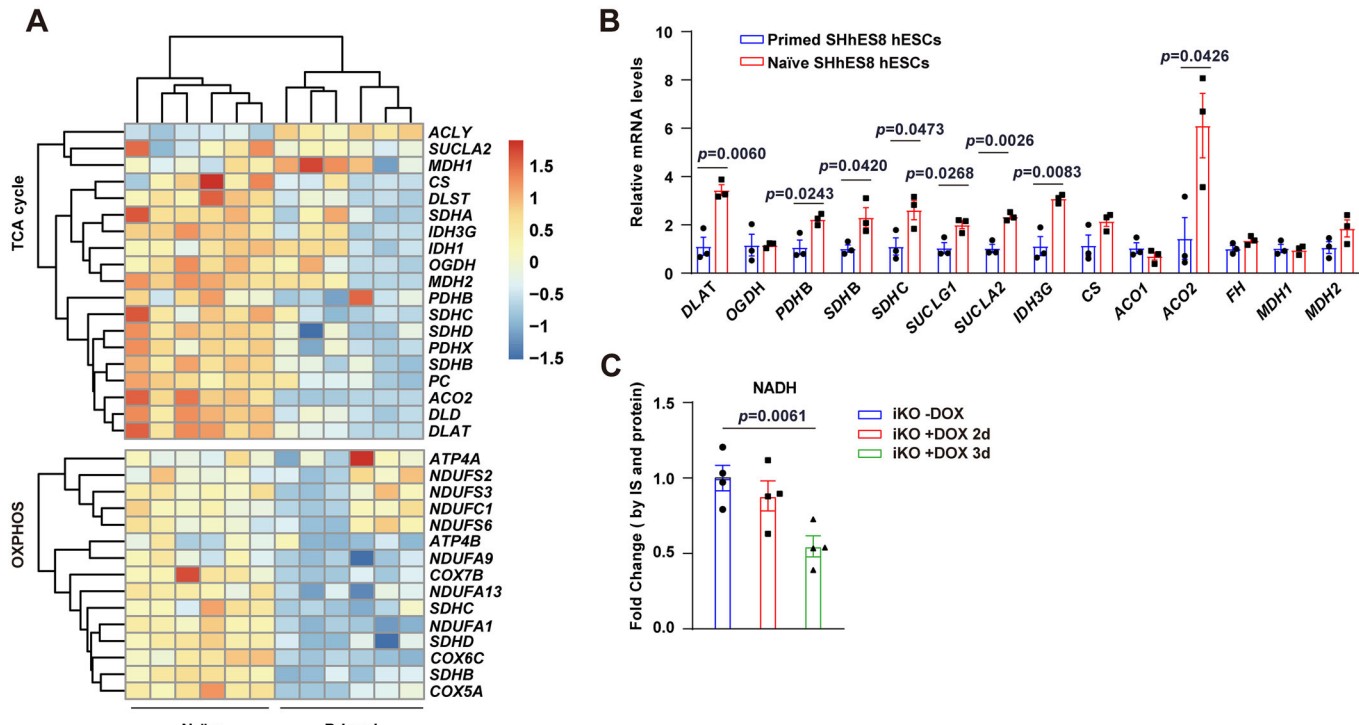

**Figure EV5. Expression profiles of metabolism-associated genes and abundance of NADH.**

(**A**) Heatmaps showing mRNA levels of genes related to the TCA cycle and OXPHOS in naïve and primed hESCs. Public RNA-seq datasets (GEO: GSE169678) and our RNA-seq datasets (SRA: PRJNA1258868) were used. The color represents Z-scores. Six biological replicates were used for each state of hESCs. (**B**) Results of RT-qPCR assays for the comparison in relative mRNA levels of TCA cycle- related genes between primed and naïve hESCs. Data are presented as mean ± SEM (*n* = 3). (**C**) Bar charts showing the relative abundance of NADH in naïve *NANOG* iKO SHhES8 hESCs, either untreated or treated with DOX for two or three days, as measured by the LC-MS analysis. Data are shown as mean ± SEM (*n* = 4). The unpaired two-tailed student's *t*-test was used for the statistical analysis in (**B, C**).

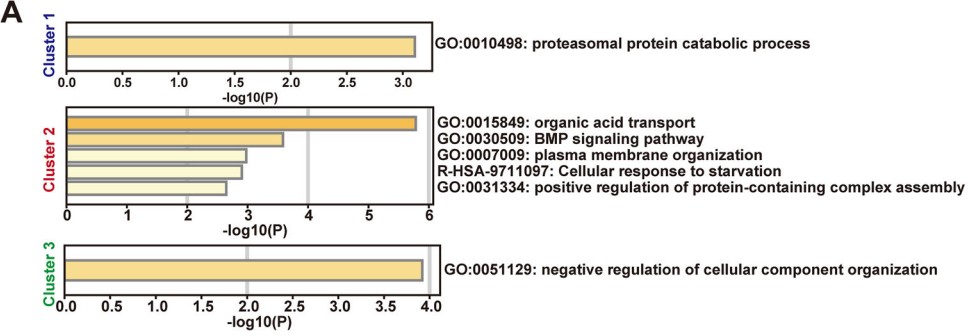

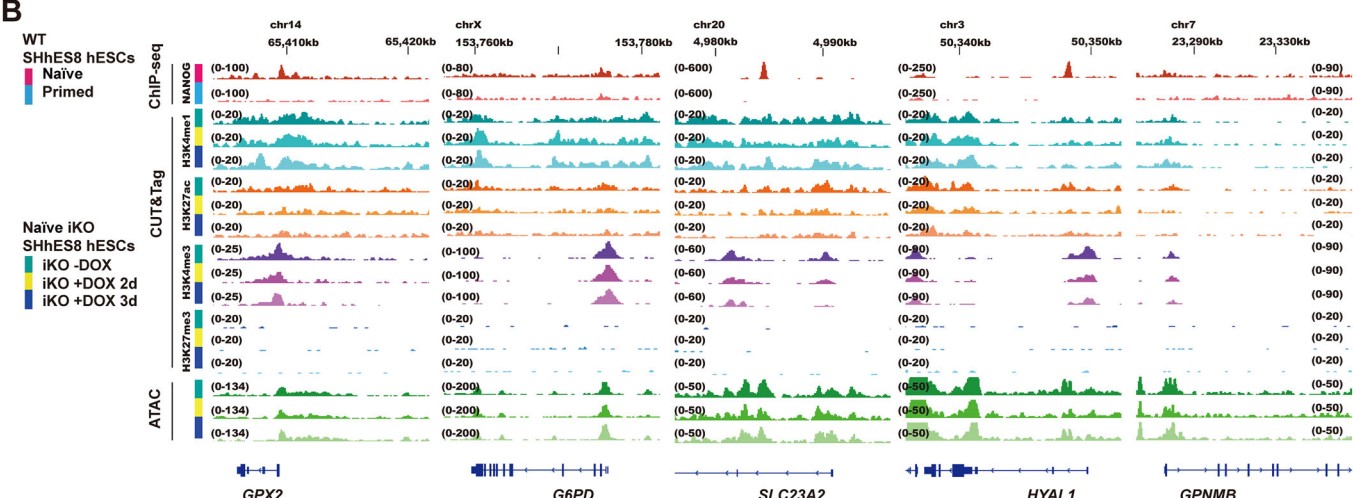

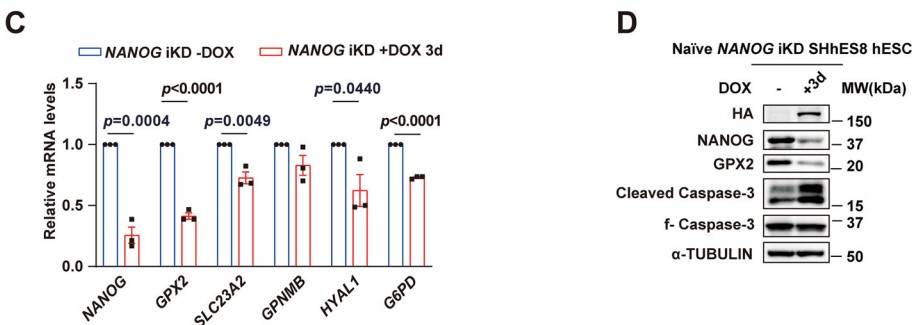

**Figure EV6. NANOG directly regulates certain oxidative stress- related genes in naïve hESCs.**

(A) Bar charts showing functional annotation of genes from the three clusters described in Fig. 6C. The hypergeometric test was used for statistical analysis. (B) Genome browser snapshots of NANOG occupancy at the vicinity of *GPX2*, *G6PD*, *SLC23A2*, *HYAL1*, and *GPNMB* in primed and naïve SHhES8 hESCs, as well as histone modifications and chromatin openness in naïve *NANOG* iKO SHhES8 hESCs, either untreated or treated with DOX for two or three days. (C) RT-qPCR analysis results for relative mRNA levels of indicated genes in naïve *NANOG* iKD SHhES8 hESCs, either untreated or treated with DOX for three days. Data are presented as mean ± SEM (n = 3). *GPX2*, $p = 2.48 \times 10^{-5}$; *G6PD*, $p = 2.31 \times 10^{-6}$. The unpaired two-tailed student's *t*-test was used for statistical analysis. (D) The representative western blot analysis result for levels of indicated proteins in naïve *NANOG* iKD SHhES8 hESCs, either untreated or treated with DOX for three days. dCas9 was tagged by HA. Alpha-TUBULIN served as a loading control.

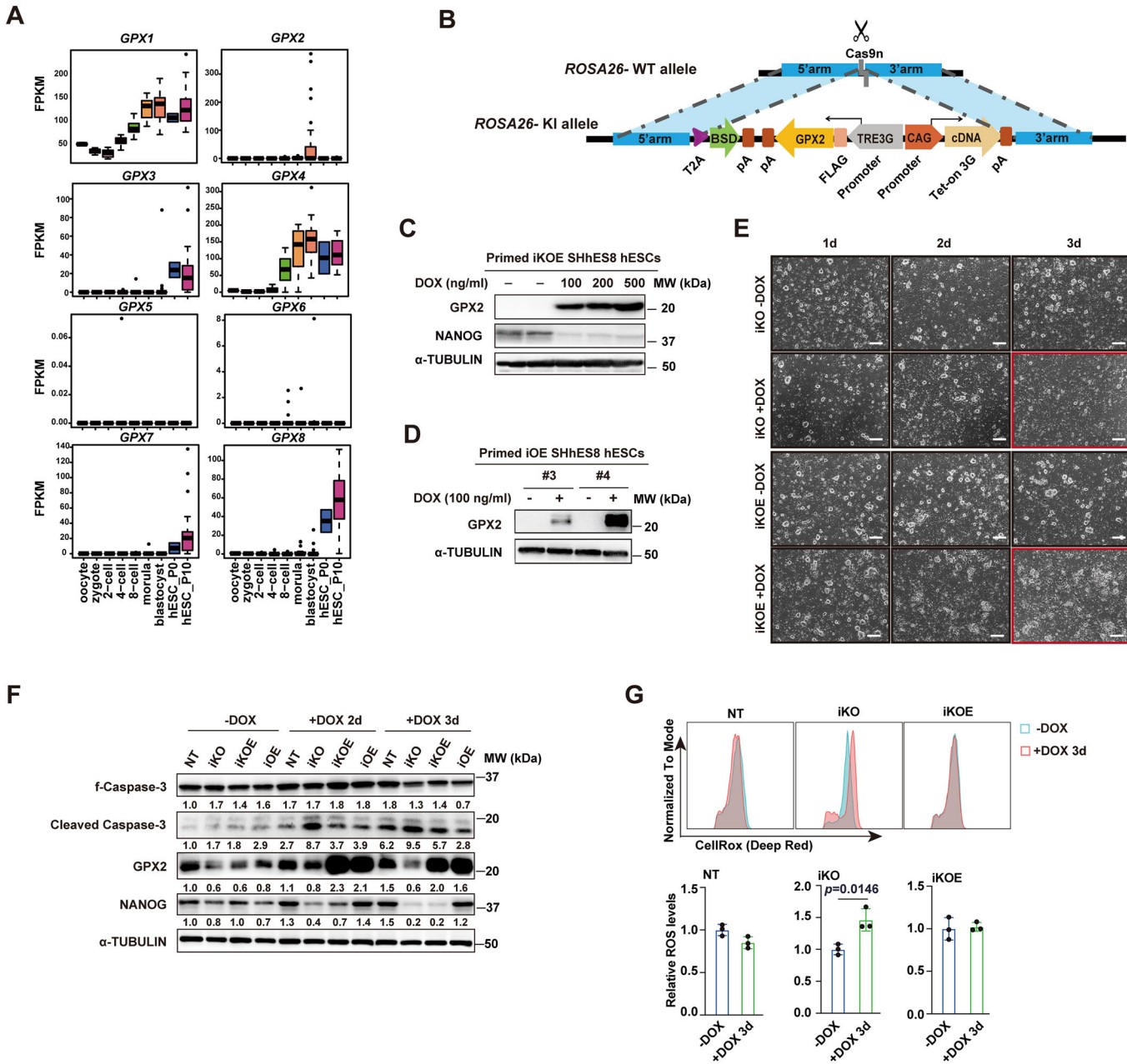

**Figure EV7.** *GPX2* expression profiles and its rescue effect on *NANOG* depletion- induced oxidative stress and cell death in naïve hESCs.

(A) The expression profile of glutathione peroxidase family members during human early embryonic development and in cultured hESCs. Data were obtained from the published dataset (GEO: GSE36552). (B) The schematic diagram showing the insertion of the cassette for DOX inducible overexpression of *GPX2* into the ROSA26 locus of hESCs. (C, D) The representative western blot analysis results of GPX2 and NANOG protein levels in primed iKOE (C) and GPX2 in iOE (D) SHhES8 hESCs, either untreated or treated with DOX for three days. Alpha-TUBULIN served as a loading control. (E) Phase contrast images of naïve *NANOG* iKO and iKOE SHhES8 hESCs, either untreated or treated with DOX. Scale bar, 200 μm. (F) The representative western blot analysis result for levels of indicated proteins in naïve NT, iKO, iKOE, and iOE SHhES8 hESCs, either untreated or treated with DOX for two or three days. Alpha-TUBULIN served as a loading control. The number under each row indicates the relative protein abundance measured by ImageJ. (G) The representative ROS peak graph calculated by the CellROX staining combined with flow cytometric analysis in naïve NT, iKO, and iKOE hESCs cultured in the PXGL medium, treated without or with DOX (top). The quantitative analysis of relative ROS levels from three independent experiments are shown at the bottom. Data are shown as mean ± SD ($n = 3$). The unpaired two-tailed Student's $t$-test was used for statistical analysis.

 