## [Peer Review File · EMBO Reports]

NANOG governs cell metabolism and redox homeostasis in human naïve embryonic stem cells

Ying Jin, Bing Liao, Min Shao, Han Wang, Yujie Liu, Yongqiang Wang, Hanzhi Zhao, Junjie Gu, Ning Zhong, Yifan Zhou, and Huiyong Yin

Corresponding author(s): Ying Jin (yjin@sibs.ac.cn) , Ying Jin (yjin@sibs.ac.cn), Bing Liao (liaoqing@shsmu.edu.cn)

Review Timeline:

Transfer Date:	12th Feb 25
Editorial Decision:	18th Feb 25
Revision Received:	22nd Jul 25
Editorial Decision:	19th Sep 25
Revision Received:	16th Oct 25
Accepted:	24th Oct 25

Editor: Achim Breiling / Esther Schnapp

Transaction Report: This manuscript was transferred to EMBO reports following peer review at The EMBO Journal.

Dear Dr. Jin,

Thank you for transferring your manuscript to EMBO reports. I now went through the manuscript, the referee reports from The EMBO Journal (attached again below) and your revision plan (provisional point-by-point response).

Judging from your revision plan, it seems that you will be able to adequately address the referee concerns during a major revision. I thus invite you to revise your manuscript accordingly with the understanding that all referee concerns must be addressed in the revised manuscript and/or in a final detailed point-by-point response, as indicated in your revision plan.

Revised manuscripts should be submitted within three months of a request for revision. Please contact me to discuss the revision (also by video chat) if you have further questions or comments regarding the revision, or should you need additional time.

Acceptance of your manuscript will depend on a positive outcome of another round of review at EMBO reports, using the same referees.

1) a .docx formatted version of the final manuscript text (including legends for main figures, EV figures and tables), but without the figures included. Please make sure that changes are highlighted to be clearly visible. Figure legends should be compiled at the end of the manuscript text.

2) individual production quality figure files as .eps, .tif, .jpg (one file per figure), of main figures and EV figures. Please upload these as separate, individual files upon re-submission. Please make sure that all figure panels are called out separately and sequentially in the manuscript text

For more details please refer to our guide to authors:

See also our guide for figure preparation:

Moreover, please consult our guidelines for figure legend preparation:

4) a complete author checklist, which you can download from our author guidelines (<https://www.embopress.org/page/journal/14693178/authorguide>). Please insert page numbers in the checklist to indicate where the requested information can be found in the manuscript. The completed author checklist will also be part of the RPF.

5) that primary datasets produced in this study (e.g. RNA-seq, ChIP-seq and array data) are deposited in an appropriate public

database. This is now mandatory (like the COI statement). If no primary datasets have been deposited in any database, please state this in this section (e.g. 'No primary datasets have been generated and deposited').

The accession numbers and database should be listed in a formal "Data Availability " section (placed after Materials & Methods) that follows the model below. Please note that the Data Availability Section is restricted to new primary data that are part of this study.

Data availability

8) Regarding data quantification and statistics, please make sure that the number "n" for how many independent experiments were performed, their nature (biological versus technical replicates), the bars and error bars (e.g. SEM, SD) and the test used to calculate p-values is indicated in the respective figure legends (also for potential EV figures and all those in the final Appendix). Please also check that all the p-values are explained in the legend, and that these fit to those shown in the figure. Please provide statistical testing where applicable. Please avoid the phrase 'independent experiment' but clearly state if these were biological or technical replicates. Please also indicate (e.g. with n.s.) if testing was performed, but the differences are not significant. In case n=2, please show the data as separate datapoints without error bars and statistics.

See also:

<http://www.embopress.org/page/journal/14693178/authorguide#statisticalanalysis>

Please add to each legend (main, EV figures, Appendix, where applicable) a 'Data Information' section explaining the statistics used or providing information regarding replicates and scales. See:

9) Please add scale bars of similar style and thickness to microscopic images, using clearly visible black or white bars (depending on the background). Please place these in the lower right corner of the images themselves. Please do not write on or near the bars in the image but define the size in the respective figure legend.

10) Please note our reference format:

11) We updated our journal's competing interests policy in January 2022 and request authors to consider both actual and perceived competing interests. Please review the policy <https://www.embopress.org/competing-interests> and add a statement declaring your competing interests. Please name that section 'Disclosure and Competing Interests Statement' and add it after the author contributions section.

12) Please restrict the keywords to 5 and order the manuscript sections like this using these names:

Title page - Abstract - Keywords - Introduction - Results - Discussion - Methods - Data availability section (DAS) - Acknowledgements (including the funding information) - Disclosure and Competing Interests Statement - References - Figure legends - Expanded View Figure legends

13) Please have your revised manuscript carefully proofread by a native speaker.

14) Please make sure that all the funding information is also entered into the online submission system and is complete and similar to the one in the manuscript text file (in the Acknowledgements).

15) We now use CRediT to specify the contributions of each author in the journal submission system. CRediT replaces the author contribution section. Please use the free text box to provide more detailed descriptions. Thus, please do NOT provide your final manuscript text file with an author contributions section. See also guide to authors: <https://www.embopress.org/page/journal/14693178/authorguide#authorshipguidelines>

16) All materials and methods used need to be described in the main text using our 'Structured Methods' format, which is required for all research articles. According to this format, the Methods section should include a Reagents and Tools Table (listing key reagents, experimental models, software, and relevant equipment and including their sources and relevant identifiers), uploaded as separate file, followed by a Methods section in which we encourage the authors to describe their methods using a step-by-step protocol format with bullet points, to facilitate the adoption of the methodologies across labs. More information on how to adhere to this format as well as downloadable templates (.doc or .xls) for the Reagents and Tools Table can be found in our author guidelines (section 'Structured Methods'):

Please add the gRNA sequence and the primer information (Tables S1-S3) to the Reagents and Tools Table.

I look forward to seeing a revised form of your manuscript when it is ready.

Please let me know if you have questions or comments regarding the revision.

Kind regards,

Achim

Referee #1:

The authors study the role of NANOG, one of the core transcription factors for pluripotency, in naïve human embryonic stem cells (hESC). One of the ways NANOG modulates the epigenetic landscape is by regulating histone acetylation through maintaining the levels of intracellular acetyl-CoA. It was shown that NANOG is essential for maintenance of bivalent metabolic state, glycolysis and OXPHOS, typically observed in naïve hESCs. The authors identified a novel target of NANOG, GPX2, which plays essential role in maintaining redox homeostasis by reducing ROS levels to enhance survival of naïve hESCs. Overall, the authors extensively studied the role of NANOG in maintaining pluripotency by looking into transcription, epigenetic signatures, and metabolism.

Overall comments:

Overall, this paper is well written with supporting evidence for their claims. The paper has a logical flow and rationale behind

each experiment are well explained. The authors performed multiple different experiments to support the claims, looking into transcription, epigenetic landscape, and metabolism. This paper provided novel insights into the multifaceted role of NANOG in maintaining pluripotency.

Major comments:

1. While the primary focus of the paper is to investigate the role of NANOG in naïve pluripotent stem cells. The authors could discuss their perspectives of NANOG's roles in naïve and primed states.
2. The discussion section is unwieldy, and the writing can be improved. In particular, the second paragraph of the discussion seems to suggest that the authors have identified many novel insights in gene and chromatin regulation in hESC. The authors should acknowledge what is known in the literature and cite those papers.
3. Inconsistency between fig. 1a and 1g: the reduction of OCT4 and SOX2 protein levels slightly decreases with NANOG KO in fig. 1a but the reduction is much more prominent in fig. 1g.
4. In fig. 1h, there seems to be an outlier in iKO +DOX 3d sample where it is making the mRNA expression of lineage marker genes higher. As such, GATA6 mRNA expression is not reflecting the protein levels when looking at iKO +DOX 2d vs iKO +DOX 3d.
5. In fig. 3i, protein bands for ACCS2 and DLAT do not look reduced after NANOG KO although quantification suggests otherwise. Similarly in fig. S3g, the reduction in protein levels of ACCS2 and ALDH2 does not look convincing on the western blot image.
6. Authors also mentioned that there is reduced expression of CPT1A at transcript levels but not at protein levels. Any explanation on why that might be the case? If protein does not reduce, it may not be fair to assume there is any downstream effects.
7. For fig. 3j, addition of NaAc rescued the reduction of gene expression of ACADS and CPT1A after NANOG KO. What about the other genes (ACSS2, ALDH2, DLAT, and MPC2)? Since it is RNAseq data, data on other genes should be easily retrievable.
8. In fig. S3a, there does not seem to be a good reduction of NANOG protein levels in both iKO #5 and iKO #7 especially in 3d samples.
9. It would be good to provide possible explanations on the increased levels of aspartate and glutamine (fig. 4c).
10. In fig. 6b, there is almost negligible protein levels of NANOG in primed hESCs but in fig. S1e, we can see that primed hESCs express a good amount of NANOG protein. Please explain the discrepancy.

Minor comments:

1. Consider changing the colours for the heatmap as most would assume red is downregulated.
2. Typo error on page 25: "... support a role of NAONG in the control of..."

Referee #2:

Using a doxycycline inducible system, in this paper Shao et al. show that NANOG controls viability, gene expression, activating and repressive histone modifications, TF binding and metabolic processes - such as glycolysis and mitochondrial respiration - in naïve human ESCs. The authors highlight the mutual dependence of these different aspects of Nanog activity. The data presented provide examples of how NANOG activates expression of specific genes, through classical mechanism of transcriptional regulation, while also affecting more globally the deposition of activating histone acetylation marks, by controlling the metabolism of hESC and, consequently, the cellular levels of acetyl-CoA. Finally the authors discuss how NANOG might activate genes involved in the protection against ROS, which might explain the poor viability of NANOG-depleted cells. I am not an expert in metabolism, and therefore will not comment in detail on the aspects of this work pertinent to that field. However, I point out that, while the paper does not report any novel general finding on the metabolism of naïve and primed hESCs, the authors highlights a potential novel role of NANOG in directly controlling energy metabolism. The conclusions of this work related to the mechanisms through which NANOG regulates transcription and ESC identity are also not entirely novel. However, the control of the expression of metabolic genes by NANOG is assessed in detail, which bring valuable new data, and experiments are in general based on clear, complete, and well executed experiments.

The paper suffers from two related technical limitations that can be easily addressed:

First, all experiments are performed using a single mean of knocking-down Nanog expression, which seem to work with variable efficiency, and may introduce stress related to a protracted expression of Cas9 in the cells. Alternative means of controlling NANOG levels should be included in the analysis.

Second, it is difficult to disentangle the indirect effects due to the loss of naïve pluripotency/differentiation triggered by Nanog depletion, from those specifically attributable to a direct transcriptional control by this TF. Note that Oct4 mRNA and protein levels are reduced at day 3 of depletion. The authors do present evidence of Nanog binding in the proximity of putative target genes, which argues in favor of direct effects. However repeating some analyses after manipulating NANOG protein levels more rapidly would solidify the claims. The authors should consider rescuing Nanog expression in stable KO hESC lines, and analysing the transcriptional effects of an acute loss/gain of function. This could be achieved by doxycycline inducible systems, as done for other factors in the last figures of the paper, or more convincingly by expressing degron-tagged NANOG protein in KO cells. Similarly, it is not clear whether the effects of Nanog depletion on the metabolic state of hESC are direct. Repeating a selection of the analyses presented in figures 3-6 using orthogonal gain/loss of function systems should be considered.

Besides these general comments, here are some specific remarks:

Figure 2f: The authors present metaplots to show that the levels of H3K27ac are reduced more drastically in proximity to naïve pluripotency genes, than near other classes of genes. However, this claim is not fully supported by the current analyses. First, the presence of single regions showing high levels of variation can profoundly impact metaplot representations. The authors should instead provide heatmaps of the levels of H3K27Ac around the body of naïve pluripotency or other classes of genes. Most importantly, it is difficult to compare the reduction near pluripotency genes, to the global reduction of histone acetylation documented in figure 3. A boxplot representation of the changes in acetylation signal at the promoters or accessible regions proximal to pluripotency vs other gene categories, and all genes, should be provided. In alternative, the enrichment of regions showing reduced acetylation in proximity of different gene classes should be computed.

Also related to Figure 2f: the authors present evidence that NANOG supports H3K27me3 deposition at the promoters of bivalent genes. This observation is very interesting and deserves further investigation. Also, the authors should discuss how their observation relate to what already described in mouse ESCs (see Heurtier et al. "The molecular logic of Nanog-induced self-renewal in mouse embryonic stem cells"; Nature Communications, 2019).

Discussing Fig. 3k the authors claim: "The transcriptome analysis showed that the addition of NaAc to DOX- treated cells (for both 1 and 2 days) rescued NANOG depletion- induced downregulation of naïve pluripotency genes, including DNMT3L, UTF1, MAEL, ALPP, and KHDC1L, whereas it exerted little impact on NANOG depletion-induced upregulation of PrE and TE marker genes". However, the panel of naïve pluripotency genes presented in the figure does not include key markers and pluripotency TFs, and the effects are variable. Is expression of genes like Dppa3, Dppa5, Klf5, Rex1, Tbx3,..., consistently rescued, and to what extent? For instance, expression of Klf17 and Tfcp2l1 seems negatively affected by NaAc. Moreover, the choice of a heatmap representation, without a properly annotated legend, makes it difficult to understand the data. Please provide a complete analysis of fold change in expression of key pluripotency genes, in a bar plot of similar format, as done in other figure panels.

As a general note, the figures are extremely busy and font size very small. Some of the panels, for instance some of the histone mark and TF binding profiles at individual loci, could be moved to supplementary.

Referee #3:

In this manuscript, the authors develop a DOX-inducible NANOG-KO iPSC line and document their epigenetic landscape. They then focus on NANOG-dependent transcriptional reprogramming that impacts cell metabolism and describe how NANOG possibly regulates acetyl-CoA and antioxidants levels.

The work is obviously interesting for stem cell basic research, and has some merit. The metabolic angle is interesting as the nature and consequence of iPSC-metabolic ambivalence is unknown. However, it lacks focus and depth. In its current status, the manuscript is organized in three major sections (epigenetic mapping, acetyl-CoA reprogramming; antioxidant response) that are poorly glued together. The first section is very descriptive with little insight and unclear novelty. The other two parts - related to metabolic reprogramming - are poorly executed and conclusions not well supported.

In addition, authors often cherry pick examples of regulated genes but fail to provide a more comprehensive description of NANOG-regulated genes/loci. Some of the effects they define as "evident" are unclear in fact.

Another issue with the general architecture of the work is that is sometimes difficult to compare effects in naïve and primed cells (for examples, quantifications are often normalized against two different controls).

In the first part, authors execute multiple CUT&tag (and ATAC-seq) to map chromatin conformation in naïve vs primed iPSC before and after NANOG ablation (DOX addition). This is overlapped with mapping of NANOG binding sites (ChIP-seq). The work is extensive and provides a useful description of NANOG binding sites. The major conclusion is the NANOG preferentially sits at euchromatin loci, which is not a particularly insightful conclusion. Moreover, authors fail to robustly quantify this effect although an interesting statement is that NANOG is "poorly" associated to epigenetically bivalent (poised) loci. The authors should provide more data to support this conclusion, which is interesting indeed.

The relationship between NANOG binding and gene expression is unclear. Authors get to that much later in the text (Fig 5) but do not specifically make any comment. However, authors conclude this first section stating that "These epigenetic features align well with transcriptional changes induced by NANOG deficiency in naïve hESCs". I don't think this is supported by their data.

Because NANOG is strongly associated to genomic loci marked by H3K27ac, authors decided to look at how NANOG impacts acetyl-CoA and histone acetylation. I don't think the reasoning is sound. However, NANOG ablation does promote strong reductions in both global levels of histone acetylation and acetyl-CoA levels. This is obviously a strong and novel finding. Of note, tracks in Fig 2g do not show any obvious change in histone acetylation (while authors do claim that in the text).

That said, authors fail to provide a solid explanation for how that happens. They highlight 6 genes somehow "related" to acetyl-CoA production but to me they are more vaguely implicated in catabolic metabolism. Many of those are expected to impact

mitochondrial acetyl-CoA, which is not available for histone acetylation. Only exception is ACSS2, but its regulation by NANOG is not really clear from the western blots (or tracks) shown here. The blot showing the effect of acetate supplementation (Fig 3c) is extremely poor.

Authors later expand the analysis to mitochondrial function. This is superficial, but it does seem that NANOG depletion impairs mitochondria. However, DOX alone can disrupt mitochondrial function. Authors are invited to always use a non-DOX supplemented control. The final point authors want to make with this figure (Fig 4) is unclear.

Is "cellular NADH" total NAD(H) or reduced NAD specifically? Authors should specify that and draw conclusions accordingly. The final part on oxidative metabolism is somewhat the most solid because there's a direct link with transcription and authors use rescue construct (GPX2-OE) to pinpoint the mechanism. The work is still a bit superficial, bouncing between the PPP pathway and GPX2, but has some novelty. More work is needed to define the culprit of NANOG-dependent remodeling of antioxidant response. Did authors checked whether NANOG impact Nrf2 activation, for example?

Several typos were noted.

The point-by-point response letter for the manuscript entitled “NANOG governs cell metabolism and redox homeostasis to secure human naïve pluripotency” (manuscript # EMBOR-2025-61331V2)

Referee #1:

Summary of the paper:

The authors study the role of NANOG, one of the core transcription factors for pluripotency, in naïve human embryonic stem cells (hESC). One of the ways NANOG modulates the epigenetic landscape is by regulating histone acetylation through maintaining the levels of intracellular acetyl-CoA. It was shown that NANOG is essential for maintenance of bivalent metabolic state, glycolysis and OXPHOS, typically observed in naïve hESCs. The authors identified a novel target of NANOG, GPX2, which plays essential role in maintaining redox homeostasis by reducing ROS levels to enhance survival of naïve hESCs. Overall, the authors extensively studied the role of NANOG in maintaining pluripotency by looking into transcription, epigenetic signatures, and metabolism.

Response: Thanks for the summary.

Overall comments:

Overall, this paper is well written with supporting evidence for their claims. The paper has a logical flow and rationale behind each experiment are well explained. The authors performed multiple different experiments to support the claims, looking into transcription, epigenetic landscape, and metabolism. This paper provided novel insights into the multifaceted role of NANOG in maintaining pluripotency.

Response: Thanks for the positive comments.

Major comments:

Comment 1. While the primary focus of the paper is to investigate the role of

NANOG in naïve pluripotent stem cells. The authors could discuss their perspectives of NANOG's roles in naïve and primed states.

Response: To address this concern, we added the following statement at the end of the first result section on page 8, “NANOG activates naïve pluripotency genes and prevents extraembryonic lineage differentiation in naïve hESCs, while it sustains expression of primed pluripotency genes and suppresses differentiation into embryonic lineages in primed hESCs. Despite of these differences, NANOG is essential for maintaining cell survival and the undifferentiated state for both naïve and primed hESCs.” In addition, we further introduced and discussed previous and current findings related to NANOG/Nanog in mESCs, primed hESC, and naïve hESCs in the second paragraph of the discussion section.

Comment 2. The discussion section is unwieldy, and the writing can be improved. In particular, the second paragraph of the discussion seems to suggest that the authors have identified many novel insights in gene and chromatin regulation in hESC. The authors should acknowledge what is known in the literature and cite those papers.

Response: Thanks for pointing out this defect in our discussion section. In the revised manuscript, we have substantially edited the discussion section. Particularly, the second and third paragraphs were largely modified. We acknowledged previous findings related to roles and mechanisms of NANOG in controlling pluripotency of both human and mouse pluripotent stem cells in the literature and cited these papers.

Comment 3. Inconsistency between fig. 1a and 1g: the reduction of OCT4 and SOX2 protein levels slightly decreases with NANOG KO in fig. 1a but the reduction is much more prominent in fig. 1g.

Response: Thanks for your careful examination of our data. As you know, western blot assays are usually used to examine relative changes in protein

levels. The whole cell lysate samples used in Fig. 1a and Fig.1g were obtained from different batches of naïve hESCs, where Cas9 induction was made independently. Thus, the efficiency of *NANOG* depletion as well as changes in OCT4 and SOX2 protein levels might not be exactly the same. Although the extent of reduction in OCT4 and SOX2 protein levels were different between Fig. 1a and Fig 1g, the result that *NANOG* depletion reduced protein levels of these two factors is obvious. Thus, this kind of variation between different experiments does not affect our conclusion that *NANOG* depletion reduced protein levels of OCT4 and SOX2 (results are shown in revised Fig. 1A and Fig. 1F). Moreover, all of our western blot results was obtained from at least three independent experiments, and representative findings from consistent results are presented in the manuscript.

Comment 4. In fig. 1h, there seems to be an outlier in iKO +DOX 3d sample where it is making the mRNA expression of lineage marker genes higher. As such, GATA6 mRNA expression is not reflecting the protein levels when looking at iKO +DOX 2d vs iKO +DOX 3d.

Response: Thanks again for your careful examination of our data. As you mentioned, there was indeed one of three iKO +DOX 3d samples somewhat different from the other two (Fig.1h in original manuscript; Fig. 1G in the revised manuscript). However, the mean value from this RT-qPCR result was consistent with our RNA-seq data (Fig. 1f in our original manuscript; Fig. 1E in the revised manuscript). The latter also clearly showed that *NANOG* depletion led to the upregulation of lineage marker genes, and the upregulation was more evident at day 3 than day 2 of DOX treatment. A similar trend of upregulation of GATA6 protein levels was observed (Fig. 1g in original manuscript; Fig. 1F in the revised manuscript). GATA6 protein levels increased markedly by *NANOG* depletion, with a higher level at day 3 than day 2. As pointed out by the referee, the difference in the protein level of GATA6 between iKO +DOX 2d and iKO +DOX 3d seemed not as evident as in the mRNA level,

this could be attributed to the variation of samples from different batches. Nevertheless, the upregulation of GATA6 is consistent upon *NANOG* deletion in naïve hESCs at both transcriptional and protein levels, which is further supported by our immunofluorescent staining (Fig.1i in original manuscript; Fig. 1H in the revised manuscript). Thus, the result of the activation of GATA6 by *NANOG* depletion is not due to the presence of this outlier. Our conclusion is supported by results of orthogonal approaches, including RT-qPCR, western blotting, immunostaining and RNA-sequencing.

Comment 5. In fig. 3i, protein bands for ACCS2 and DLAT do not look reduced after *NANOG* KO although quantification suggests otherwise. Similarly in fig. S3g, the reduction in protein levels of ACCS2 and ALDH2 does not look convincing on the western blot image.

Response: I understand why the reviewer had this concern. Let me explain. Our quantification analysis was made by the grayscale intensity for protein bands on the blots. The band intensities of our interested proteins were then normalized to α -Tubulin intensity in the same sample. Therefore, whether the protein level changes not only measured by the intensity of this particular protein band, but also determined by the band intensity of the internal control (α -Tubulin intensity) in the same sample and manipulated simultaneously with our interested proteins. During the revision of this manuscript, we conducted more experiments and further validated the reduction in the protein abundance of ACCS2, DLAT, and ALDH2 induced by *NANOG* depletion in naïve hESCs. The new result is shown in Fig. 4F of the revised manuscript, and also shown below. The decrease in the level of these proteins after *NANOG* depletion is credible.

Figure 4F. The representative western blot analysis result for proteins examined in naïve NT or *NANOG* iKO hESCs, either untreated or treated with DOX for two or three days. Alpha-TUBULIN served as a loading control (the left side). The quantitative analysis results of relative protein levels are shown in the right side. Data are presented as mean \pm SEM (n=4). A two-tailed Student's *t* test was used for statistical analysis.

Comment 6. Authors also mentioned that there is reduced expression of CPT1A at transcript levels but not at protein levels. Any explanation on why that might be the case? If protein does not reduce, it may not be fair to assume there is any downstream effects.

Response: Thanks for raising this question. We found that NANOG can directly bind to the regulatory region of *CPT1A*, and that *NANOG* depletion decreased the *CPT1A* transcript level, accompanied by reduced H3K27ac modification at the *CPT1A* locus. This suggests that CPT1A could be a transcriptional target of NANOG. At a protein level, *NANOG* depletion also reduced CPT1A protein levels in SHhES8 hESCs (Fig. 4F and EV4E in revised manuscript), consistent with its transcript change in *NANOG* depleted cells. However, we did not find the reduction in the CPT1A protein level when *NANOG* was depleted in H9 hESCs (Fig. EV4D). Therefore, we pointed out in the revised manuscript on page 16 that “The decrease in the CPT1A protein level was variable between hESC lines (Fig. 4F; Fig. EV4D, E), although its downregulation by *NANOG* depletion was invariably detected at the

transcriptional level (Fig. 4E). Thus, the contribution of CPT1A to NANOG-mediated regulation of cellular acetyl-CoA levels needs further investigation.”. Currently, we do not know why the two hESC lines responded to *NANOG* depletion differently regarding to the CPT1A protein level.

Comment 7. For fig. 3j, addition of NaAc rescued the reduction of gene expression of *ACADS* and *CPT1A* after *NANOG* KO. What about the other genes (*ACSS2*, *ALDH2*, *DLAT*, and *MPC2*)? Since it is RNAseq data, data on other genes should be easily retrievable.

Response: In response to the question of the referee, we have carefully checked our RNA-seq data, and found that NaAc treatment elevated the expression levels of *ACADS*, *CPT1A* and *ACSS2*, but not the other three genes (*ALDH2*, *DLAT*, and *MPC2*). However, the increase in the expression level was statistically significant only for *ACSS2* by the Padj value. In our original Fig. 3J, the statistical significance was judged according to the p value. By the p value, the elevation in the expression of *ACADS* and *CPT1A* was statistically significant. As RNA-seq data should be analyzed by the Padj value, we feel that the non-significant changes should not be included in the manuscript. The omission of this result from our manuscript does not affect our conclusion that *NANOG* directly controls the transcription of these 6 genes. The analysis result is shown below for referee’s information.

I would like to explain further. The effect of *NANOG* depletion on the expression of these six gene was stronger at day 3 than at day 2. However, the supplement of NaAc lasted only for the first 2 days, as cells could not endure this treatment longer than 2 days. This restricted us to fully explore the rescue effect of NaAc supplement.

Figure for referee with unpublished data and its description has been removed upon request by the authors.

Comment 8. In fig. S3a, there does not seem to be a good reduction of NANOG protein levels in both iKO #5 and iKO #7 especially in 3d samples.

Response: Yes, NANOG protein levels recovered somewhat in day 3 samples, although NANOG protein levels in both day 2 and day 3 samples were lower than those in NT cells. Based on our experience in culturing these hESCs, we have noticed that H9 hESCs behave differently from SHhES8 hESCs, being more vulnerable to culture disturbances than SHhES8 hESCs. For SHhES8 hESCs, NANOG depletion- induced cell death occurs most severely at day 3, with lower NANOG protein levels at day 3 than day 2. However, for H9 hESCs, massive cell death takes place at day 2, with lower NANOG protein levels at day 2 than day 3. For H9 hESCs, the day 3 sample might contain less NANOG depleted cells than in the day 2 sample. In fact, the difference in NANOG protein levels between day 2 and day 3 of DOX treatment could be influenced by the cell growth state and biological activity of DOX used in each independent experiment, which all affect outcomes of DOX treatment. This fact also explains why we need at least 3 independent experiments to make a conclusion. The result shown in the original Fig. S3a is only one representative result. In the revised manuscript, we have provided a more representative western blot result, which shows that DOX treatment leads to evident reductions in NANOG protein levels at day 2 and 3 in both iKO #5 and iKO #7

H9 cells, albeit with a slight increase in NANOG protein levels at day 3, compared to day 2 in iKO #5 H9 cells. The related result is shown below and also in revised Fig. EV3A.

Figure EV3A. The representative western blot analysis result for levels of NANOG proteins and histone acetylation modifications in primed NT and *NANOG* iKO H9 hESCs, either untreated or treated with DOX for two or three days. Alpha-TUBULIN and histone 3 served as loading controls. Single clonal culture of *NANOG* iKO #5 and #7 was used.

Comment 9. It would be good to provide possible explanations on the increased levels of aspartate and glutamine (fig. 4c).

Response: To be honest, we currently don't have a good explanation to this question. This was an unexpected result. We deemed that it is an interesting finding, so included this in the manuscript (Fig. 5C in the revised manuscript). It could be removed if the referee considers it inappropriate to have an unexplained result in the manuscript.

Based on the data generated in our manuscript, we would like to discuss possible explanations here: (1) as the expression of enzymes related to aspartate synthesis (such as GOT1 and GOT2) was not significantly altered by *NANOG* depletion, it is unlikely that enhanced synthesis of aspartate was the main cause. Considering that *NANOG* KO led to the reduction in acetyl-CoA levels, we speculate that oxaloacetate (OAA), a precursor of aspartate, may not be fully consumed through the TCA cycle when acetyl-CoA is insufficient,

and is instead converted to aspartate by aspartate aminotransferase (GOT1 and GOT2). This might lead to the increase in aspartate abundance; (2) *NANOG* depletion reduced acetyl-CoA and citrate levels, cells might increase glutamine uptake to compensate for the insufficient material supply for TCA cycle, which was indicated by the slight increase of α -KG, succinate, fumarate, and malate at day 3 of DOX induction compared to day 2 counterparts (Fig. 5A in the revised manuscript). Thus, the decrease in acetyl-CoA caused by *NANOG* depletion may drive the accumulation of aspartate and glutamine through metabolic shunting and feedback regulation. This phenomenon reflects the adaptive capability of naïve hESCs under metabolic stress. Of course, these explanations need further investigation to validate.

Comment 10. In fig. 6b, there is almost negligible protein levels of *NANOG* in primed hESCs but in fig. S1e, we can see that primed hESCs express a good amount of *NANOG* protein. Please explain the discrepancy.

Response: Thanks for asking this question. As a core transcriptional factor of PSCs, *NANOG* is highly expressed in both primed and naïve hESCs. However, we and others have found that *NANOG* protein levels are relatively higher in naïve hESCs compared to their primed counterparts. The protein levels measured by western blotting are relative and determined by many factors, including the protein amount loaded to gels and exposure time length. Again, as I explained in the preceding question, the intensity of the protein band is also normalized to that of an internal control in the same sample. Therefore, the comparison of protein levels between two samples should only be made in the same experiments. Results from our original Fig. 6b and Fig. 1Se were from two different experiment settings. Fig. 6b shows the relative protein levels of *NANOG* between primed and naïve hESCs, whereas Fig. S1e shows the comparison in *NANOG* protein levels between control cells (without DOX) and DOX- treated cells at the primed state. In Fig. 6b, if we would expose the blot for a longer time, the good amount of *NANOG* could be seen in primed hESCs,

however, the intensity of NANOG in naïve hESCs would be too strong to be quantified. Nevertheless, we have to admit that the result shown in the original Fig. 6b is not an appropriate one, as the level of α -TUBULIN was obviously lower in primed hESC sample than that in naïve hESCs, which leads to negligible protein levels of NANOG on the surface in primed hESCs. In the revised manuscript, we have conducted experiments with an appropriate exposure time and comparable amount of loading samples for hESCs at the two states. As shown in Fig. 7C of the revised manuscript, relative levels of NANOG proteins can be easily compared between primed and naïve hESCs, and our statistical analysis further supports that NANOG protein levels are higher in naïve hESCs than that in primed hESCs. Thus, our conclusion remains the same that the NANOG protein level is higher in naïve hESCs than that in their primed counterparts. The result is also shown below.

Figure 7C. The representative western blot analysis result for protein levels of GPX2 and NANOG in primed and naïve hESCs (the left side). Alpha-TUBULIN served as a loading control. The quantitative analysis results of relative protein levels of NANOG are shown at the right side. Data are presented as mean \pm SEM (n=3). A two-tailed Student's *t* test was used for statistical analysis.

Minor comments:

Comment 1. Consider changing the colours for the heatmap as most would assume red is downregulated.

Response: Thank for your suggestion. However, we have used the red color to indicate upregulation or higher expression levels throughout the manuscript. We would like to stay with it. I think that either way is fine, as long as the

meaning of the color is indicated in the figure or figure legend. I hope that it is acceptable.

Comment 2. Typo error on page 25: "... support a role of NAONG in the control of..."

Response: Thanks for the correction. It has been corrected in the revised manuscript.

Referee #2:

Using a doxycycline inducible system, in this paper Shao et al. show that NANOG controls viability, gene expression, activating and repressive histone modifications, TF binding and metabolic processes - such as glycolysis and mitochondrial respiration - in naïve human ESCs. The authors highlight the mutual dependence of these different aspects of Nanog activity. The data presented provide examples of how NANOG activates expression of specific genes, through classical mechanism of transcriptional regulation, while also affecting more globally the deposition of activating histone acetylation marks, by controlling the metabolism of hESC and, consequently, the cellular levels of acetyl-CoA. Finally the authors discuss how NANOG might activate genes involved in the protection against ROS, which might explain the poor viability of NANOG-depleted cells. I am not an expert in metabolism, and therefore will not comment in detail on the aspects of this work pertinent to that field. However, I point out that, while the paper does not report any novel general finding on the metabolism of naïve and primed hESCs, the authors highlight a potential novel role of NANOG in directly controlling energy metabolism. The conclusions of this work related to the mechanisms through which NANOG regulates transcription and ESC identity are also not entirely novel. However, the control of the expression of metabolic genes by NANOG is assessed in detail, which bring valuable new data, and experiments are in general based on clear, complete, and well executed experiments.

The paper suffers from two related technical limitations that can be easily addressed:

First, all experiments are performed using a single mean of knocking-down Nanog expression, which seem to work with variable efficiency, and may introduce stress related to a protracted expression of Cas9 in the cells. Alternative means of controlling NANOG levels should be included in the analysis.

Second, it is difficult to disentangle the indirect effects due to the loss of naïve pluripotency/differentiation triggered by Nanog depletion, from those specifically attributable to a direct transcriptional control by this TF. Note that Oct4 mRNA and protein levels are reduced at day 3 of depletion. The authors do present evidence of Nanog binding in the proximity of putative target genes, which argues in favor of direct effects. However repeating some analyses after manipulating NANOG protein levels more rapidly would solidify the claims. The authors should consider rescuing Nanog expression in stable KO hESC lines, and analysing the transcriptional effects of an acute loss/gain of function. This could be achieved by doxycycline inducible systems, as done for other factors in the last figures of the paper, or more convincingly by expressing degron-tagged NANOG protein in KO cells. Similarly, it is not clear whether the effects of Nanog depletion on the metabolic state of hESC are direct. Repeating a selection of the analyses presented in figures 3-6 using orthogonal gain/loss of function systems should be considered.

Response: Thanks for the summary and positive comments to our study, and the suggestion is very constructive. In order to address the two technical limitations raised by the referee, we have conducted the following experiments:

- 1) In the revised manuscript, we have utilized the CRISPR-dCas9 technique as an alternative mean to knock down *NANOG* expression, and related results are shown in revised Figure EV4F, G and Figure EV6C, D. In addition, we made our best effort to establish stable NANOG-dTAG hESC line, hoping to control NANOG protein levels rapidly and reversibly. To date, we have successfully generated the construct for inserting the dTAG sequence to the *NANOG* loci. Unfortunately, we have only obtained hESC colonies with insertion of the dTAG sequence into one of the *NANOG* loci (Response Letter Figure 2). The hESCs with the insertion of dTAG sequence into both *NANOG* loci have not been obtained. Considering the uncertainty for the time needed to obtain such hESC colonies, and the fact that we have successfully

expressed exogenous *NANOG* in *NANOG* depleted hESCs (details provided below), we would like to submit the revised manuscript now without data in *NANOG*-dTAG hESCs. On the other hand, to address the concern of the referee about protracted expression of Cas9 in our *NANOG* iKO cells and distinguish effects of *NANOG* depletion from cellular stress induced by protracted expression of Cas9, we have included the NT control naïve hESCs, which also carry inducible Cas9 expression cassette in related experiments. Utilization of this control cell line could exclude the possibility that phenotypes observed in *NANOG* depleted cells are due to cellular stress induced by protracted expression of Cas9.

Figure EV4. (F) RT-qPCR analysis results for relative mRNA levels of the indicated genes in naïve *NANOG* iKD SHhES8 hESCs, either untreated or treated with DOX for three days. Data are presented as mean \pm SEM (n=3). **(G)** The representative western blot analysis result for levels of indicated proteins in naïve *NANOG* iKD SHhES8 hESCs, either untreated or treated with DOX for three days. Alpha-TUBULIN and histone 3 served as loading controls.

Figure EV6. (C) RT-qPCR analysis results for relative mRNA levels of the indicated genes in naïve *NANOG* iKD SHhES8 hESCs, either untreated or treated with DOX for three days. Data are presented as mean \pm SEM (n=3). **(D)** The representative western blot analysis result for levels of indicated proteins in naïve *NANOG* iKD SHhES8 hESCs, either untreated or treated with DOX for three days. Alpha-TUBULIN served as a loading control.

Figure for referee with unpublished data and its description has been removed upon request by the authors.

2) Regarding disentangling the indirect effects due to the loss of naïve pluripotency/differentiation triggered by *NANOG* depletion, from those specifically attributable to a direct transcriptional control by *NANOG*, our *NANOG* ChIP-seq experiment described in our original manuscript is a commonly used and effective mean to map the binding profile of a transcriptional factor genome-wide. In combination with analysis of differentially expressed genes induced by *NANOG* depletion, we have

identified genes directly regulated by NANOG. Therefore, we, at least partially, disentangled the direct and indirect effects induced by *NANOG* depletion. Furthermore, in response to the helpful suggestion of the referee, we have established *NANOG-mut* iOE_*NANOG* iKO hESCs. In these hESCs, DOX treatment can induce the depletion of endogenous *NANOG* and overexpression of exogenous *NANOG* carrying synonymous mutations to avoid targeting by sgRNA. With these hESCs, we are able to show that *NANOG-mut* overexpression abrogates *NANOG* depletion- induced reduction in histone acetylation (Fig. 3B in the revised manuscript), down regulation of 6 genes related to acetyl-CoA production (Fig. 4G, 4H in the revised manuscript), and reduced expression of 5 genes related to oxidative stress (Fig. 6G in the revised manuscript). These data strongly support the conclusion that the transcriptional and epigenetic changes observed in *NANOG*- depleted cells are specifically caused by *NANOG* deficiency, although they are not sufficient to distinguish the direct and indirect effects of *NANOG*.

The related results are also provided below.

Figure 3B. A representative western blot analysis result for protein levels of *NANOG* and histone acetylation modifications in naïve *iNANOG-mut_NANOG* iKO SHhES8 hESCs, either untreated or treated with DOX for three days. H3 served as a loading control.

Figure 4. (G) The representative western blot analysis result for levels of proteins examined in naïve *iNANOG-mut_NANOG* iKO SHhES8 hESCs, either untreated or treated with DOX for three days. Alpha-TUBULIN served as a loading control. **(H)** RT-qPCR analysis results for relative mRNA levels of indicated genes in naïve *iNANOG-mut_NANOG* iKO SHhES8 hESCs, either untreated or treated with DOX for three days. Data are presented as mean \pm SEM (n=3).

Figure 6G. RT-qPCR analysis results for relative mRNA levels of the indicated genes in naïve *iNANOG-mut_NANOG* iKO SHhES8 hESCs, either untreated or treated with DOX for three days. Data are presented as mean \pm SEM (n=3).

Besides these general comments, here are some specific remarks:

Specific Comment1: Figure 2f: The authors present metaplots to show that the levels of H3K27ac are reduced more drastically in proximity to naïve pluripotency genes, than near other classes of genes. However, this claim is not fully supported by the current analyses. First, the presence of single regions showing high levels of variation can profoundly impact metaplot

representations. The authors should instead provide heatmaps of the levels of H3K27Ac around the body of naïve pluripotency or other classes of genes. Most importantly, it is difficult to compare the reduction near pluripotency genes, to the global reduction of histone acetylation documented in figure 3A boxplot representation of the changes in acetylation signal at the promoters or accessible regions proximal to pluripotency vs other gene categories, and all genes, should be provided. In alternative, the enrichment of regions showing reduced acetylation in proximity of different gene classes should be computed.

Response: Thank you very much for your constructive advice. In the revised manuscript, we have provided heatmaps of the levels of H3K27ac as well as H3K4me1, H3K4me3, and H3K27me3, around the body of naïve pluripotency- and two extraembryonic lineages- associated genes (Fig. 2F in the revised manuscript and below). In addition, by computing the H3K27ac modification density, we have further showed that the reduction of H3K27ac occurs after *NANOG* deletion in proximal regions of naïve pluripotency genes, but not of TE and PrE gene classes. The result is showing below and in revised Fig. 2G.

Figure 2F. The deepTools was used to calculate the signal intensities normalized by the total reads of histone modifications (H3K4me1, H3K4me3, H3K27ac, and H3K27me3) within the 3 kb range upstream and downstream of marker genes associated with naïve pluripotency, TE, or PrE in naïve *NANOG* iKO SHhES8 hESCs, either untreated or treated with DOX for three days.

Figure 2G. Violin plots showing the mean signal intensities of H3K27ac modification within the 2 kb range upstream and downstream of marker genes associated with naïve pluripotency, TE, or PrE in naïve *NANOG* iKO SHhES8

hESCs, either untreated or treated with DOX for three days. Naïve, $p = 0.0532$; TE, $p = 0.4561$; PrE, $p = 0.4806$.

Specific comment 2: Also related to Figure 2f: the authors present evidence that NANOG supports H3K27me3 deposition at the promoters of bivalent genes. This observation is very interesting and deserves further investigation. Also, the authors should discuss how their observation relate to what already described in mouse ESCs (see Heurtier et al. "The molecular logic of Nanog-induced self-renewal in mouse embryonic stem cells"; Nature Communications, 2019).

Response: Thanks for your suggestions and mention of the 2019 NC paper, which is indeed closely related to the current study, although it was conducted in mESCs. In the revised manuscript, we cited this NC paper and discussed how our observation that NANOG supports H3K27me3 deposition at the promoters of bivalent genes relates to findings reported by Heurtier et al in the third paragraph of the revised Discussion section. In addition, we have analyzed H3K27me3 and H3K4me3 profiles for all bivalent genes in *NANOG* depleted cells, and found that *NANOG* depletion led to decreases in H3K27me3 enrichments in the promoter regions of bivalent genes, with H3K4me3 enrichment being unchanged. The result is shown below and in Fig. 2H of revised manuscript.

Figure 2H. Heatmaps depicting changes in the intensity (normalized by total reads) of histone modifications (H3K4me3 and H3K27me3) on the bivalent genes in naïve *NANOG* iKO SHhES8 hESCs, either untreated or treated with DOX for three days, analyzed by the deepTools.

Specific comment 3. Discussing Fig. 3k the authors claim: "The transcriptome analysis showed that the addition of NaAc to DOX- treated cells (for both 1 and 2 days) rescued *NANOG* depletion- induced downregulation of naïve pluripotency genes, including *DNMT3L*, *UTF1*, *MAEL*, *ALPP*, and *KHDC1L*, whereas it exerted little impact on *NANOG* depletion-induced upregulation of PrE and TE marker genes". However, the panel of naïve pluripotency genes presented in the figure does not include key markers and pluripotency TFs, and the effects are variable. Is expression of genes like *Dppa3*, *Dppa5*, *Klfs*, *Rex1*, *Tbx3*,..., consistently rescued, and to what extent? For instance, expression of *Klf17* and *Tfcp2l1* seems negatively affected by NaAc. Moreover, the choice of a heatmap representation, without a properly annotated legend, makes it difficult to understand the data. Please provide a complete analysis of fold change in expression of key pluripotency genes, in a bar plot of similar format, as done in other figure panels.

Response: Thanks for pointing this issue out. First, naïve pluripotency markers are not all the same between mouse ESCs and naïve hESCs. Genes included in original Fig. 3k are typical markers for human naïve pluripotency according to the published literatures. In this figure, we intended to show the naïve pluripotency markers whose expressions might be modulated by NaAc. Mechanistically, *NANOG* maintains naïve pluripotency gene expression primarily through directly controlling their transcription. The contribution of epigenetic controls is limited. The number of genes regulated by *NANOG* through specific H3K27ac modification is further reduced. To illustrate the role of intracellular acetyl-CoA for the function of *NANOG* in gene expression, we utilized NaAc supplementation, which has been often used to mimic functions

of acetyl-CoA. However, it needs to be converted to acetyl-CoA inside cells by ACSS2. In fact, *NANOG* depletion downregulated *ACSS2*, thus the efficiency of conversion would be lower in *NANOG* depleted cells than in normal cells. Therefore, it is not expected to have many downregulated genes rescued by NaAc. In this study, we did not detect significant alterations in the expression of *DPPA3*, *DPPA5*, *KLFs* (*KLF2/4/5*), after *NANOG* depletion or NaAc supplement. However, *ALPP* and *MAEL*, two human naïve pluripotency genes, were consistently observed to be significantly upregulated by NaAc, compared to DOX treatment alone. A statistical analysis of their expression levels in a bar plot format is provided in the revised Fig. EV3E and also below. As to reduced expression of *KLF17* and *TFCP2L1* seen with NaAc supplement, we do not have a good explanation. It could be a feedback response to NaAc supplement. Moreover, we provide expression analysis of marker genes for PrE and TE lineages in addition to markers of naïve pluripotency, showing that NaAc does not influence the expression of these extraembryonic lineage genes. Furthermore, a properly annotated legend for the heatmap is also provide in revised Fig. EV3D.

Figure EV3E. The FPKM values from our RNA sequencing data for two naïve pluripotency marker genes, *ALPP* and *MAEL*, either untreated or treated with DOX or DOX and NaAc in naïve *NANOG* iKO hESCs. Data are presented as mean \pm SEM ($n=3$). A two-tailed Student's *t* test was used for statistical analysis.

Specific comment 4. As a general note, the figures are extremely busy and

font size very small. Some of the panels, for instance some of the histone mark and TF binding profiles at individual loci, could be moved to supplementary.

Response: Thanks for pointing out this issue. In the revised manuscript, we have improved the clarity and representation of figures to ensure readability.

Referee #3:

Comment 1: In this manuscript, the authors develop a DOX-inducible NANOG-KO iPSC line and document their epigenetic landscape. They then focus on NANOG-dependent transcriptional reprogramming that impacts cell metabolism and describe how NANOG possibly regulates acetyl-CoA and antioxidants levels.

The work is obviously interesting for stem cell basic research, and has some merit. The metabolic angle is interesting as the nature and consequence of iPSC-metabolic ambivalence is unknown. However, it lacks focus and depth. In its current status, the manuscript is organized in three major sections (epigenetic mapping, acetyl-CoA reprogramming; antioxidant response) that are poorly glued together. The first section is very descriptive with little insight and unclear novelty. The other two parts - related to metabolic reprogramming - are poorly executed and conclusions not well supported.

Response: Regarding to the three major sections of our manuscript probed by the referee, we would like to make some explanations. (1) In the first section, we wanted to describe the phenotype of *NANOG* depletion in naïve hESCs, with a comparison to primed hESCs, followed by seeking for key target genes of *NANOG* using multi-omics analysis to explain phenotypes of *NANOG* depletion: naïve pluripotency loss and cell death. To our knowledge, this study is the first systematic research of *NANOG* targets in naïve hESCs; (2) In the second section, we provided experimental evidence that acetyl-CoA production and global histone acetylation can partially explain the decrease of transcription level of certain human naïve pluripotency marker genes. However, the major force governing naïve pluripotency marker gene expression by *NANOG* is through its direct occupancy on the regulatory regions of its target genes. In the revised manuscript, we used the *NANOG* overexpression rescue model to further validate our conclusion; (3) In the third section, we specifically focused on *GPX2*, which is a new target gene of *NANOG* identified in this study, to provide molecular basis for *NANOG* maintaining redox homeostasis

and supporting survival of naïve hESCs. We deem that these sections are logically organized. We agree with the referee, some parts of our study lack depth. However, under current culture conditions for naïve hESCs, it is much more difficult to investigate molecular mechanisms of pluripotency regulation, particularly the regulation of metabolism, in naïve hESCs than in primed hESCs, due to the need of feeder cells for naïve hESC growth and low efficiency in transfection.

Comment 2: In addition, authors often cherry pick examples of regulated genes but fail to provide a more comprehensive description of NANOG-regulated genes/loci. Some of the effects they define as "evident" are unclear in fact.

Response: Thanks for your valuable comments. We fully agree that a comprehensive characterization of NANOG-regulated genes/loci is critical to support our conclusions. Based on our comprehensive transcriptional profiling and genome-wide mapping of NANOG binding, we identified 6 genes related to acetyl-CoA production and 5 genes related to redox homeostasis, which can be applied to explain the reduction of global histone acetylation and the oxidative stress- induced cell death after *NANOG* depletion, respectively. To consolidate these findings in the revised manuscript, we used two more cell models: *NANOG* iKD hESCs (an alternative model for *NANOG* iKO), and *iNANOG-mut_NANOG* iKO hESCs (a rescue model for *NANOG* iKO). *NANOG* depletion- induced decreases in histone acetylation (Fig. EV4G in the revised manuscript), acetyl-CoA production- related 6 genes (Fig. EV4F, G in the revised manuscript), and redox- related 5 genes (Fig. EV6C, D in the revised manuscript) were consistently detected in naïve *NANOG* iKD hESCs and rescued by *NANOG-mut* overexpression in naïve *iNANOG-mut_NANOG* iKO hESCs (Fig. 3B, Fig. 4G, H, and Fig. 6G in the revised manuscript), reinforcing the specificity of NANOG- mediated functions.

Figure EV4. (F) RT-qPCR analysis results for relative mRNA levels of the indicated genes in naïve *NANOG* iKD SHhES8 hESCs, either HA untreated or treated with DOX for three days. Data are presented as mean \pm SEM (n=3). **(G)** The representative western blot analysis result for levels of indicated proteins in naïve *NANOG* iKD SHhES8 hESCs, either untreated or treated with DOX for three days. Alpha-TUBULIN and histone 3 served as loading controls.

Figure EV6. (C) RT-qPCR analysis results for relative mRNA levels of the indicated genes in naïve *NANOG* iKD SHhES8 hESCs, either untreated or treated with DOX for three days. Data are presented as mean \pm SEM (n=3). **(D)** The representative western blot analysis result for levels of indicated proteins in naïve *NANOG* iKD SHhES8 hESCs, either untreated or treated with DOX for three days. Alpha-TUBULIN served as a loading control.

Figure 3B. A representative western blot analysis result for protein levels of NANOG and histone acetylation modifications in naïve *iNANOG-mut_NANOG* iKO SHhES8 hESCs, either untreated or treated with DOX for three days. H3 served as a loading control.

Figure 4. (G) The representative western blot analysis result for levels of proteins examined in naïve *iNANOG-mut_NANOG* iKO SHhES8 hESCs, either untreated or treated with DOX for three days. Alpha-TUBULIN served as a loading control. **(H)** RT-qPCR analysis results for relative mRNA levels of indicated genes in naïve *iNANOG-mut_NANOG* iKO SHhES8 hESCs, either untreated or treated with DOX for three days. Data are presented as mean \pm SEM (n=3).

Figure 6G. RT-qPCR analysis results for relative mRNA levels of the indicated genes in naïve *iNANOG-mut_NANOG* iKO SHhES8 hESCs, either untreated or treated with DOX for three days. Data are presented as mean \pm SEM (n=3).

Comment 3: Another issue with the general architecture of the work is that is sometimes difficult to compare effects in naïve and primed cells (for examples, quantifications are often normalized against two different controls).

Response: Thanks for this comment. In most cases, we compared between DOX- treated samples and untreated samples, as these samples were generated from the same cell line. This analysis strategy was used to ensure that only one treatment was different between the two samples. It is challenging to compare the effects of *NANOG* deficiency in naïve and primed hESCs by normalizing against a single control. However, in specific contexts such as gene expression level comparisons, we directly analyzed naïve and primed samples (Fig. 7B, C; Fig. EV1A, B in the revised manuscript).

Comment 4: In the first part, authors execute multiple CUT&tag (and ATAC-seq) to map chromatin conformation in naïve vs primed iPSC before and after *NANOG* ablation (DOX addition). This is overlapped with mapping of *NANOG* binding sites (ChIP-seq). The work is extensive and provides a useful description of *NANOG* binding sites. The major conclusion is the *NANOG* preferentially sits at euchromatin loci, which is not a particularly insightful conclusion. Moreover, authors fail to robustly quantify this effect although an interesting statement is that *NANOG* is "poorly" associated to epigenetically bivalent (poised) loci. The authors should provide more data to support this conclusion, which is interesting indeed.

Response: Thanks for point out this issue. In the revised manuscript, we have analyzed our multiple omics data using analysis methods suggested by the referee #2, providing more quantitative data to support our conclusions. (1) By

computing the H3K27ac modification density, we have further showed that the reduction in H3K27ac occurs after *NANOG* deletion in proximal regions of naïve pluripotency genes, but not of TE and PrE gene classes (Fig. 2G in the revised manuscript); (2) We have analyzed H3K27me3 and H3K4me3 profiles for all bivalent genes in *NANOG* depleted cells, and found that *NANOG* depletion led to decreases in H3K27me3 enrichments in the promoter regions of bivalent genes, without changes in H3K4me3 enrichments (Fig. 2H in the revised manuscript). Thus, *NANOG* may contribute to maintaining the bivalent state by promoting H3K27me3 modifications at these loci.

Figure 2G. Violin plots showing the mean signal intensities of H3K27ac modification within the 2 kb range upstream and downstream of marker genes associated with naïve pluripotency, TE, or PrE in naïve *NANOG* iKO SHhES8 hESCs, either untreated or treated with DOX for three days. Naïve, $p = 0.0532$; TE, $p = 0.4561$; PrE, $p = 0.4806$.

Figure 2H. Heatmaps depicting changes in the intensity (normalized by total reads) of histone modifications (H3K4me3 and H3K27me3) on the bivalent genes in naïve *NANOG* iKO SHhES8 hESCs, either untreated or treated with DOX for three days, analyzed by the deepTools.

Comment 5: The relationship between NANOG binding and gene expression is unclear. Authors get to that much later in the text (Fig 5) but do not specifically make any comment. However, author conclude this first section stating that "These epigenetic features align well with transcriptional changes induced by NANOG deficiency in naïve hESCs". I don't think this is supported by their data.

Response: To address this issue, the upregulated and downregulated DEGs occupied by NANOG are provided in the revised manuscript, as shown in revised Fig. 4A, B and Fig. EV4A, B. Moreover, the original statement of "These epigenetic features align well with transcriptional changes induced by NANOG deficiency in naïve hESCs" has been changed to the more specific statement of "Therefore, reductions in the H3K27me3 level at PrE and TE genes as well as decreases in H3K4me1 and H3K27ac levels near human naïve pluripotency genes after *NANOG* depletion aligned well with upregulation of extraembryonic lineage genes and downregulation of naïve pluripotency genes, respectively.". This conclusion is supported by our data shown in the revised Fig. 2F, Fig.1E and Fig. EV2E.

Comment 6: Because NANOG is strongly associated to genomic loci marked by H3K27ac, authors decided to look at how NANOG impacts acetyl-CoA and histone acetylation. I don't think the reasoning is sound. However, NANOG ablation does promote strong reductions in both global levels of histone acetylation and acetyl-CoA levels. This is obviously a strong and novel finding. Of note, tracks in Fig 2g do not show any obvious change in histone acetylation (while authors do claim that in the text).

Response: Regarding to the question of why we decided to look at how NANOG impacts acetyl-CoA and histone acetylation, in the original manuscript, we stated that “Given the centrality of acetyl-CoA as a substrate for histone acetylation, and undetectable changes in transcript levels of enzymes associated with histone acetylation or deacetylation after *NANOG* depletion (original Fig. S3c), intracellular acetyl-CoA concentrations were measured using the LC-MS analysis.” . If this statement is not clear enough. I can explain again: we found that *NANOG* ablation led to significantly reduced H3K27ac modification globally (Fig. 2E in the revised manuscript). However, it exerted little impact on the transcription levels of enzymes associated with histone acetylation or deacetylation after *NANOG* depletion (Fig. EV3C in the revised manuscript). Hence, we hypothesized that there might be a decrease in the substrates for histone acetylation modification. Therefore, we examined cellular acetyl-CoA levels. For tracks in original Fig. 2g (Fig. EV2E in the revised manuscript), the reduction in H3K27ac modification on naïve pluripotency markers (*KLF17*, *ZFP42*, and *UTF1*) is very obvious, as shown below. In contrast, *NANOG* ablation exerted little impact on H3K27ac modification on lineage markers (*GATA3* and *GATA6*). The referee did not appreciate these changes might be due to the busy figure. We have moved these figures to supplemental files and rearranged the figures.

Figure EV2E. Genome browser snapshots of NANOG occupancy at the vicinity of naïve pluripotency markers (*KLF17*, *ZFP42*, and *UTF1*) in primed and naïve SHhES8 hESCs, as well as histone modifications and chromatin openness in naïve *NANOG* iKO SHhES8 hESCs, either untreated or treated with DOX for two or three days. The red box highlights the H3K27ac modification.

Comment 7: That said, authors fail to provide a solid explanation for how that happens. They highlight 6 genes somehow "related" to acetyl-CoA production but to me they are more vaguely implicated in catabolic metabolism. Many of those are expected to impact mitochondrial acetyl-CoA, which is not available for histone acetylation. Only exception is *ACSS2*, but its regulation by NANOG is not really clear from the western blots (or tracks) shown here. The blot showing the effect of acetate supplementation (Fig 3c) is extremely poor.

Authors later expand the analysis to mitochondrial function. This is superficial, but it does seem that NANOG depletion impairs mitochondria. However, DOX alone can disrupt mitochondrial function. Authors are invited to always use a non-DOX supplemented control. The final point authors want to make with this figure (Fig 4) is unclear.

Is "cellular NADH" total NAD(H) or reduced NAD specifically? Authors should specify that and draw conclusions accordingly.

Response: We agree with the reviewer that downregulation of these 6 genes might not entirely explain the decrease in global histone acetylation caused by *NANOG* depletion. However, all these 6 genes contribute to total intracellular acetyl-CoA levels. In fact, the reduction in mitochondrial acetyl-CoA can reduce citrate generation (Fig. 5A in the revised manuscript), followed by the decrease in its transfer to the cytoplasm. In the cytoplasm, citrate can be converted to acetyl-CoA used for histone acetylation modification. Thus, mitochondrial acetyl-CoA levels can indirectly affect histone acetylation. Our conclusion about the effect of *NANOG* on *ACSS2* protein levels and the effect of NaAc supplement on histone acetylation were made according to results obtained from multiple independent experiments, and our quantitative statistical analysis reveals a significant reduction in the *ACSS2* protein levels by *NANOG* depletion. In the revised manuscript, we have provided improved western blotting images to clarify the changes in *ACSS2* protein levels (Fig. 4F in the revised manuscript and below) and histone acetylation levels (Fig. 3D in the revised manuscript and below).

Figure 4F. The representative western blot analysis result for proteins examined in naïve NT or *NANOG* iKO hESCs, either untreated or treated with DOX for two or three days. Alpha-TUBULIN served as a loading control (the left side). The quantitative analysis results of relative protein levels are shown in the right side. Data are presented as mean \pm SEM (n=4). A two-tailed

Student's *t* test was used for statistical analysis.

Figure 3D. A representative western blot analysis result for levels of NANOG and histone acetylation modifications on H3, H4, H2BK5, and H3K27 in naïve NT and *NANOG* iKO SHhES8 hESCs, either untreated or treated with DOX or DOX and NaAc. Alpha-TUBULIN and H3 served as loading controls.

Regarding to the DOX effect, our experimental design included the control (NT) cell line, in which *NANOG* was not depleted by DOX treatment. DOX treatment depleted *NANOG* and reduced the mitochondrial respiration in iKO naïve hESCs, but DOX supplementation exerted little impact on the mitochondrial respiration in NT counterparts (Fig. 5D in the revised manuscript). The finding precludes the possibility that disturbed mitochondrial respiration was caused by the DOX treatment. Moreover, we did have a non-DOX supplemental control for iKO naïve hESCs in our original manuscript.

The final point we wanted to make with Fig 4 (Fig. 5 in the revised manuscript) was stated at the end of this section in our original manuscript, “Collectively, these findings reveal, for the first time, a role of *NANOG* acting as an indispensable regulator of the bivalent metabolic state in naïve hESCs.”

In Fig. S4c (Fig. EV5C in the revised manuscript), the total reduced NADH levels were measured by the mass spectrum. In response to referee’s question, we have clarified that the level of reduced NADH was lower in *NANOG* depleted cells than in control cells (page 18 in the revised manuscript).

Comment 8: The final part on oxidative metabolism is somewhat the most solid because there's a direct link with transcription and authors use rescue construct (GPX2-OE) to pinpoint the mechanism. The work is still a bit superficial, bouncing between the PPP pathway and GPX2, but has some novelty. More work is needed to define the culprit of NANOG-dependent remodeling of antioxidant response. Did authors check whether NANOG impact Nrf2 activation, for example?

Response: Using multi-omics analysis, we enriched the term of oxidative stress process, which included *SLC23A2*, *G6PD*, *GPX2*, *HYAL1*, and *GPNMB*. Since naïve hESCs possess bivalent metabolic feature, higher OXPHOS may lead to higher ROS levels in naïve hESCs compared to their primed counterparts. This metabolic feature demands a robust system to maintain redox homeostasis. GPX2 is highly expressed in naïve hESCs, but almost absent in primed hESCs. Induced *GPX2* overexpression can partially rescue the increase in ROS abundance and cell death induced by *NANOG* depletion, indicating that GPX2 is an important player in NANOG-mediated antioxidant response. However, we cannot exclude other possibilities, such as the NRF2 pathway. According to our multi-omics datasets, the NRF2 pathway was not enriched. In addition, the transcriptional levels of *NRF2* (*NFEL2*) and *KEAP1* as well as some of its target genes, such as *p62*, *GCLC*, *GCLM*, and *SLC7A11*, remained unchanged after *NANOG* depletion. However, with limited time and financial restriction, I am afraid that we could not expand this study further. We hope that our findings could open a new window for future study to elucidate the mechanism of the unique antioxidant system in naïve pluripotent stem cells.

Comment 9: Several typos were noted.

Response: We have made careful inspections to avoid the typos in the revised manuscript.

Dear Dr. Jin

Thank you for the submission of your revised manuscript. I have taken over its handling as my colleague Achim is currently out of office. We have now received the enclosed reports from the referees and I am happy to say that all support its publication now. Only referee 2 has a few more minor suggestions that I would like you to address before we can proceed with the official acceptance of your manuscript.

A few editorial requests will also need to be addressed:

- The author credits need to be removed from the ms file. All credits need to be entered during online ms submission.
- Please answer all questions on the statistics in the author checklist and send us a new, completed checklist.
- Fig. S3i is not a correct callout, please correct (it is probably an EV figure).
- The Reagents and Tools table needs to be removed from the ms and uploaded as a separate file.

Figure Legends - Comments

- Please note that the exact p values are not provided in the legends of figures 2E, 3A, 4D, E, F; 6F, J; 7B, E; EV4 D-F; EV6 C. Please provide exact values as reasonable.
- Please indicate the statistical test used for data analysis in the legends of figures 2A, E, G; 3A, C; 4D, E, F, H; 5A, B, C, E; 6D, E, F, G, I, J, K, L; 7B, C, E, F, G; EV1 B, F, G, H; EV3 C, E; EV4 D, E, F; EV5 B, C; EV6 A, C; EV7 G.
- Please note that the box plots need to be defined in terms of minima, maxima, centre, bounds of box and whiskers, and percentile in the legends of figures 2E, G
- Please note that information related to n is missing in the legends of figures 2E, G; EV1 G, H; EV4 E
- Please note that the error bars are not defined in the legend of figure EV4 E.

The synopsis image has too much information on it, which is not readable at the final image size. Please send us a new, simpler synopsis image at the final size of 550 pixels wide and 200-600 pixels high.

As referee 2 indicates please correct the title and abstract if causality is implied but not shown. I have not gone through your ms and data, but would like to suggest something along the following title and abstract:

NANOG governs cell metabolism and redox homeostasis in human naïve embryonic stem cells

Naïve human embryonic stem cells (hESCs) possess some advantages over their primed counterparts, displaying distinctive metabolic and epigenetic properties. However, the master regulator governing these features remains unrecognized. Here, we systemically investigate functions of the core transcription factor NANOG in naïve hESCs. Acting as an upstream key regulator, NANOG directly activates genes associated with naïve pluripotency, acetyl-CoA synthesis and anti-oxidation in a naïve pluripotency state- dependent manner, and represses extraembryonic lineage genes through [Is causality shown here? IF not, please re-write] shaping epigenetic landscapes unique in naïve hESCs. NANOG modulates transcription of multiple genes in various pathways of acetyl-CoA synthesis, maintains intracellular acetyl-CoA level and characteristic epigenetic landscapes, particularly the high level of histone acetylation, in naïve hESCs. NANOG is indispensable for the high activity of both OXPHOS and glycolysis, a bivalent metabolic state typical in naïve hESCs. Furthermore, we identify GPX2 as a mediator of NANOG in sustaining redox balance and survival of naïve hESCs. Together, this study reveals previously unrecognized roles of NANOG in orchestrating transcriptional, metabolic and epigenetic signatures that all contribute to secure human naïve pluripotency.

Please adapt as you see fits.

Esther Schnapp, PhD
Senior Editor

EMBO reports

Referee #1:

In this revised version of their manuscript Shao et al. have addressed all my main concerns. In particular, alternative means of manipulating NANOG levels have been added, including the use of inducible rescue hESC lines. It would have been preferable to use such inducible systems to draw stronger conclusions regarding the direct nature of some of the effect described. Nonetheless, the new data included provides sufficient support to the claims made by the authors.

Referee #2:

The manuscript is significantly improved compared to its previous version. I commend all authors for their efforts to systematically address reviewers' concerns.

I do think the manuscript is suitable for publication in EMBO Reports.

I would however suggest to change the title, which appear too strong. Authors clearly demonstrate that NANOG depletion affects metabolism, on top of pluripotent. But evidence remain correlative and a mechanistic link between the two is not demonstrated. I understand some histone acetylation is impacted at critical loci, but no evidence is provided that is caused by down regulation of ACSS2, CPT1A, DLAT or any other NANOG-regulated gene.

The discussion is 6.5-pages long: some trimming will probably be needed.

Referee #3:

The authors have addressed the reviewer's comments.

A point-by-point response letter to comments raised by the referees of manuscript (EMBOR-2025-61331V2)

Referee #1:

Comment: In this revised version of their manuscript Shao et al. have addressed all my main concerns. In particular, alternative means of manipulating NANOG levels have been added, including the use of inducible rescue hESC lines. It would have been preferable to use such inducible systems to draw stronger conclusions regarding the direct nature of some of the effect described. Nonetheless, the new data included provides sufficient support to the claims made by the authors.

Response: Thanks for your positive comment.

Referee #2:

Comment 1. I would however suggest to change the title, which appear too strong. Authors clearly demonstrate that NANOG depletion affects metabolism, on top of pluripotent. But evidence remain correlative and a mechanistic link between the two is not demonstrated. I understand some histone acetylation is impacted at critical loci, but no evidence is provided that is caused by down regulation of ACSS2, CPT1A, DLAT or any other NANOG-regulated gene.

Response 1: Thanks for the suggestion. I have changed the title into “**NANOG governs cell metabolism and redox homeostasis in human naïve embryonic stem cells**”, and made changes in the revised abstract as suggested by the referee.

Comment 2. The discussion is 6.5-pages long: some trimming will probably be needed.

Response 2: Thanks for the suggestion. We have significantly shortened the discussion in the revised manuscript.

Referee #3:

Comment: The authors have addressed the reviewer's comments.

Response: Thanks for the support.

Dr. Ying Jin
Shanghai Institute of Nutrition and Health, CAS
CAS Key Laboratory of Tissue Microenvironment and Tumor
320 Yueyang Road
Shanghai, Shanghai 200032
China

Dear Dr. Jin,

I am very pleased to accept your manuscript for publication in the next available issue of EMBO reports. Thank you for your contribution to our journal.

Yours sincerely,
